Resource

# Reproducible single-cell annotation of programs underlying T cell subsets, activation states and functions

**Dylan Kotliar** [1,2,3,4,5,12], **Michelle Curtis**[1,2,3,4,12], **Ryan Agnew**[1,2,3,4],
**Kathryn Weinand** [1,2,3,4,6], **Aparna Nathan**[1,2,3,4,6], **Yuriy Baglaenko**[1,7,8],
**Kamil Slowikowski**[4,9], **Yu Zhao** [1,2,3,4], **Pardis C. Sabeti** [4,10,11], **Deepak A. Rao** [2] &
**Soumya Raychaudhuri** [1,2,3,4,6] ✉

T cells recognize antigens and induce specialized gene expression programs (GEPs), enabling functions like proliferation, cytotoxicity and cytokine production. Traditionally, different T cell classes are thought to exhibit mutually exclusive responses, including $T_H1$, $T_H2$ and $T_H17$ programs. However, single-cell RNA sequencing has revealed a continuum of T cell states without clearly distinct subsets, necessitating new analytical frameworks. Here, we introduce T-CellAnnoTator (TCAT), a pipeline that improves T cell characterization by simultaneously quantifying predefined GEPs capturing activation states and cellular subsets. Analyzing 1,700,000 T cells from 700 individuals spanning 38 tissues and five disease contexts, we identify 46 reproducible GEPs reflecting core T cell functions including proliferation, cytotoxicity, exhaustion and effector states. We experimentally demonstrate new activation programs and apply TCAT to characterize activation GEPs that predict immune checkpoint inhibitor response across multiple tumor types. Our software package starCAT generalizes this framework, enabling reproducible annotation in other cell types and tissues.

T cells play critical roles in cancer, infection and autoimmune disease, driving widespread interest in characterizing their states using single-cell RNA sequencing (scRNA-seq)[1–3]. Clustering, the predominant analysis approach, has key limitations for interpreting T cell profiles. Transcriptomes reflect expression of multiple GEPs—co-regulated gene modules reflecting distinct biologic functions such as defining cell type, activation states, life cycle processes or external stimuli responses[4]. T cell GEPs vary continuously[5], combine additively within individual

cells[6] and exhibit stimulus-dependent plasticity[7]. However, clustering discretizes cells, obscuring the coexpressed GEPs. For instance, proliferating T cells from multiple subsets may cluster together, masking their subsets. This may explain why clustering typically fails to delineate many canonical T cell subsets[1,8], even with integration of surface protein markers via CITE-seq[9,10].

Component-based models like nonnegative matrix factorization (NMF), hierarchical Poisson factorization and SPECTRA overcome

[1]Center for Data Sciences, Brigham and Women's Hospital and Harvard Medical School, Boston, MA, USA. [2]Division of Rheumatology, Inflammation, and Immunity, Department of Medicine, Brigham and Women's Hospital and Harvard Medical School, Boston, MA, USA. [3]Division of Genetics, Department of Medicine, Brigham and Women's Hospital and Harvard Medical School, Boston, MA, USA. [4]Broad Institute of MIT and Harvard, Cambridge, MA, USA. [5]Harvard-MIT Division of Health Sciences and Technology, Harvard Medical School, Boston, MA, USA. [6]Department of Biomedical Informatics, Harvard Medical School, Boston, MA, USA. [7]Center for Autoimmune Genetics and Etiology and Division of Human Genetics, Cincinnati Children's Hospital Medical Center, Cincinnati, OH, USA. [8]Department of Pediatrics, University of Cincinnati, College of Medicine, Cincinnati, OH, USA. [9]Center for Immunology and Inflammatory Diseases, Department of Medicine, Massachusetts General Hospital, Boston, MA, USA. [10]Department of Organismic and Evolutionary Biology, FAS Center for Systems Biology, Harvard University, Cambridge, MA, USA. [11]Howard Hughes Medical Institute, Chevy Chase, MD, USA. [12]These authors contributed equally: Dylan Kotliar, Michelle Curtis. ✉e-mail: soumya@broadinstitute.org

some limitations of clustering[8,11–14]. These methods model GEPs as gene expression vectors and transcriptomes as weighted mixtures of GEPs. Unlike principal component analysis (PCA), NMF components correspond to biologically interpretable GEPs reflecting cell types and functional states that additively contribute to a transcriptome[12]. Component-based approaches yield GEP vectors that serve as a fixed coordinate system for comparing GEP activities across datasets. This is similar to scoring gene-set activities[15] but with variable gene weights and simultaneous modeling of multiple GEPs. This prevents confounding of related signals and enables comparison of relative GEP activities. Previous analyses of T cells using component-based models have already recognized GEPs associated with T cell activation[8] and exhaustion[13] but were limited in dataset size and only addressed a small number of biological contexts. Furthermore, it is not well established how well such GEPs generalize across datasets.

Here, we present star-CellAnnoTator (starCAT), a framework to score cells based on a fixed, multi-dataset catalog of GEPs. 'star' is a wildcard placeholder based on the asterisk (*) used in programming, indicating applicability across tissues and cell types. Our specific instantiation for T cells is thus written T-CellAnnoTator (TCAT). We derive a comprehensive T cell GEP catalog by applying consensus nonnegative matrix factorization (cNMF)[12] to seven scRNA-seq datasets comprising 1.7 million T cells from 38 human tissues[1,2,10,16–19]. Combining GEPs across datasets yields 46 consensus gene expression programs (cGEPs) capturing T cell subsets, activation states and functions (Fig. 1a). We demonstrate TCAT's utility for inferring subset and antigen-specific activation (ASA) states and identifying cGEPs predictive of immunotherapy response across multiple tumor types.

## Results

### Annotating cells with predefined GEPs

We first augmented the published cNMF algorithm to enhance GEP discovery (Fig. 1a). cNMF mitigates randomness in NMF by repeating NMF and combining outputs into robust estimates, generating GEP spectra (gene weights) and per-cell activities ('usages') reflecting the relative contributions of GEPs to each cell. To improve cross-dataset GEP reproducibility, we corrected batch effects which can cause cNMF to learn redundant dataset-specific GEPs. Standard batch-correction methods are incompatible with cNMF as they introduce negative values or modify low-dimensional embeddings rather than gene-level data. Therefore, we adapted Harmony[20] to provide batch-corrected nonnegative gene-level data. Additionally, we modified cNMF to incorporate surface protein measurements in GEP spectra for CITE-seq datasets to enhance GEP interpretability (Methods).

Next, we developed starCAT to infer the usages of GEPs learned in a reference dataset in new 'query' datasets. Unlike cNMF, which learns GEP spectra and usages simultaneously, starCAT quantifies the activity of predefined GEPs within each cell, using nonnegative least squares, similarly to NMFproject[11]. starCAT then leverages the GEP usages to predict additional cell features, including lineage, T cell antigen receptor (TCR) activation and cell cycle phase (Fig. 1a). This can provide several advantages over running cNMF or similar approaches de novo: it ensures a consistent cell state representation for comparison across datasets, can quantify rarely used GEPs that may be hard to identify de novo in small query datasets and markedly reduces run time.

We benchmarked starCAT's performance through simulations where the reference and query datasets contained only partially overlapping GEPs (Methods). Simulations included two 100,000-cell references and a 20,000-cell query, where each cell expressed one subset-defining and one or more non-subset GEPs. Cells in the reference datasets included additional GEPs or lacked certain GEPs relative to the query datasets and only shared 90% of genes in common (Extended Data Fig. 1a). We then learned GEPs from each reference with cNMF and predicted their usage in the query using starCAT.

starCAT accurately inferred the usage of GEPs overlapping between the reference and query (Pearson $R > 0.7$) and predicted low usage of extra GEPs in the reference that were not in the query (Extended Data Fig. 1b–d). We observed similar prediction accuracies when predicting a simulated query dataset with half or fewer overlapping GEPs between the reference and query (Supplementary Fig. 1). starCAT outperformed direct application of cNMF to the query for overlapping GEPs, despite having extra or missing GEPs in the references. We suspected this was due to the larger size of the references and confirmed that starCAT maintained its performance across smaller query datasets while cNMF's performance declined (Extended Data Fig. 1e).

### cGEPs for T cell annotation

We next developed a catalog of T cell GEPs to use for TCAT. We analyzed T cells from seven datasets spanning blood and tissues from healthy individuals and those with coronavirus disease 2019 (COVID-19), cancer, rheumatoid arthritis or osteoarthritis (Supplementary Table 1 and Extended Data Fig. 1f). We chose datasets to reflect phenotypic breadth, large sample sizes (>70,000 T cells) and, where possible, inclusion of CITE-seq data to aid GEP interpretation. We included two COVID-19 peripheral blood mononuclear cell (PBMC) datasets, two healthy PBMC datasets and two tissue datasets to assess cross-dataset GEP reproducibility. After quality control, 1.7 million cells remained from 905 samples from 695 individuals. We applied cNMF to each batch-corrected dataset independently (Supplementary Fig. 2 and Methods).

GEPs were reproducible across datasets. To quantify this, we clustered GEPs found in different datasets and defined a cGEP as the average of each cluster (Methods). Nine cGEPs were supported by all seven datasets (mean Pearson $R = 0.81$, $P < 1 \times 10^{-50}$ all pairs) and 49 by two or more datasets (mean $R = 0.74$, $P < 1 \times 10^{-50}$ all pairs; Fig. 1b and Extended Data Fig. 1g). Across datasets, 68.4–96.8% of GEPs clustered with at least one GEP from another dataset, indicating high reproducibility. Gene expression principal components showed substantially less concordance across datasets (Extended Data Fig. 1h).

We curated a catalog of 46 T cell cGEPs—27–36 more than prior analyses[11,13,14]—including 11 discovered only in blood, 7 only in tissue, and 28 in both (Supplementary Table 1 and Fig. 1c). We excluded 49 of 52 singleton GEPs as likely dataset-specific artifacts but retained three reflecting biologically justified signals. Specifically, the rheumatoid arthritis dataset contributed a T peripheral helper ($T_{PH}$) GEP, previously identified in inflamed synovium[21] (markers included PD-1 protein, *LAG3* and *CXCL13* RNA; Supplementary Table 2), while the pan-cancer dataset contributed an exhaustion GEP (*HAVCR2*, *ENTPD1*, *LAG3*) and a T follicular helper ($T_{FH}$) GEP (PD-1 protein, *CXCR5* and *CXCL13* RNA) distinct from a second $T_{FH}$ GEP discovered in the non-cancer tissue datasets. We also identified six cGEPs corresponding to non-T cell populations, including erythrocytes (*HBA2*, *HBA1*) and plasmablasts (*JCHAIN*, *IGKC*), likely reflecting residual doublets. We retained these in the catalog to help flag doublets.

We annotated cGEPs by examining their top weighted genes (Fig. 1d, Extended Data Fig. 2a and Supplementary Fig. 3 and Supplementary Table 2). For example, *FOXP3* and *GATA3* marked the regulatory T ($T_{reg}$) and type 2 helper T ($T_H2$)-resting cGEPs. *GATA3*, *IL4*, *IL5* and *IL17A*, *RORC* and *IL26* marked the $T_H2$-activated and interleukin-17-producing helper T ($T_H17$)-activated cGEPs, respectively. Some cGEPs were also annotated based on gene-set enrichment analysis (Supplementary Note and Supplementary Table 3).

We further labeled cGEPs through association with surface marker-based gating of canonical T cell subsets in a COVID-19 PBMC CITE-seq reference (COMBAT)[17] (Extended Data Fig. 2b,c and Methods). Multivariate logistic regression revealed strong associations between specific cGEPs and the $T_{reg}$, γδT, mucosal-associated invariant T (MAIT) cell, CD4/CD8 naive, CD8 effector memory (CD8 EM),

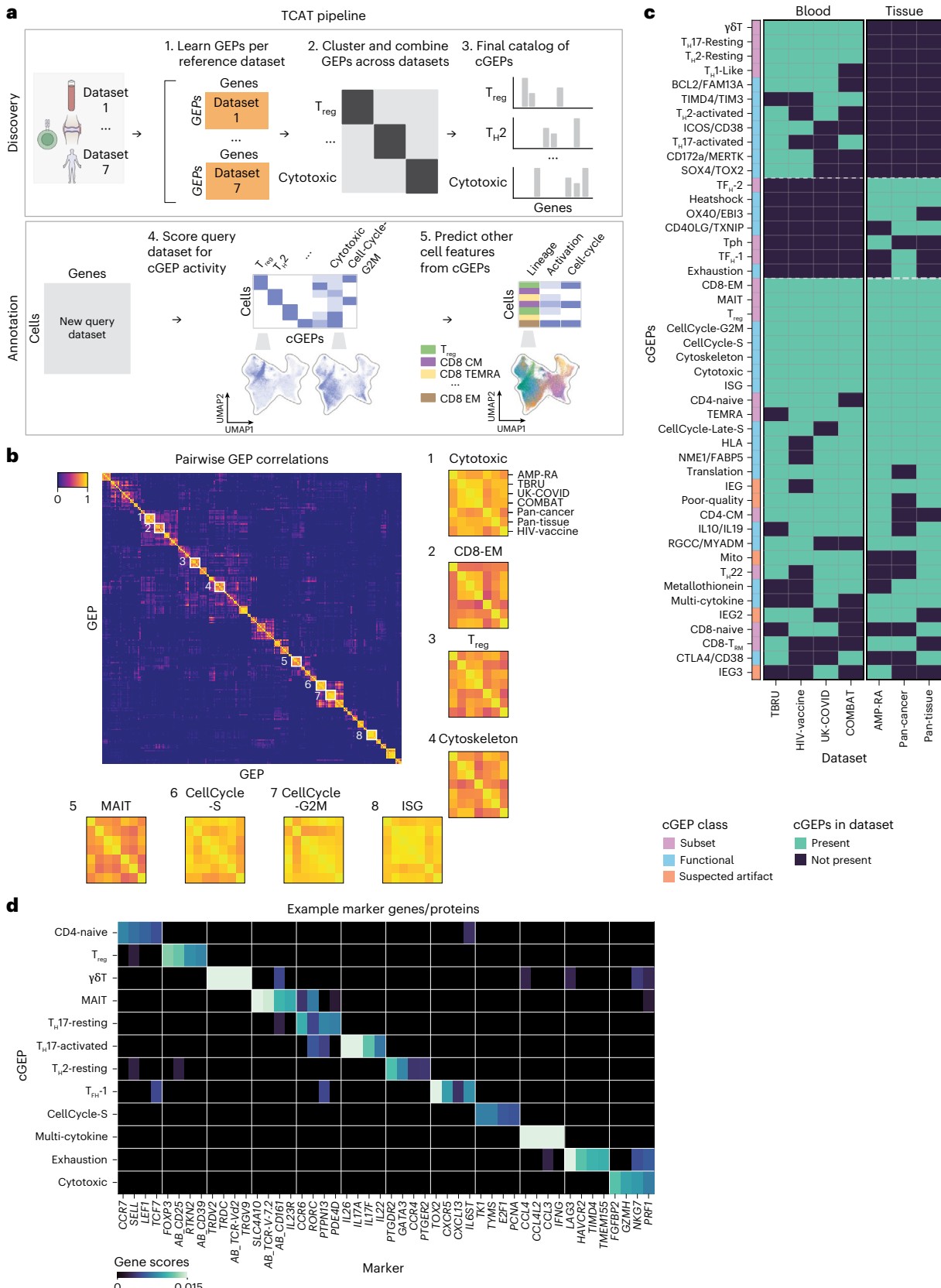

**Fig. 1 | Overview of starCAT. a**, starCAT first identifies GEPs in multiple datasets and aggregates them into cGEPs. It then uses the cGEPs to annotate new query datasets and compute additional scores and classifiers. **b**, Pairwise correlations of GEPs discovered across reference datasets with insets for cGEPs derived from all seven references. Inset row and column orders are the same for all cGEPs. **c**, Heat map of cGEPs (rows) and which datasets the comprising GEPs were found in (columns). Green boxes indicate a GEP was found in a dataset. Colored bar indicates the cGEP's assigned class. cGEPs corresponding to non-T cell lineages are excluded. **d**, Marker genes for selected example cGEPs in z-score units with the minimum value fixed at 0. The AB_ prefix indicates a surface protein.

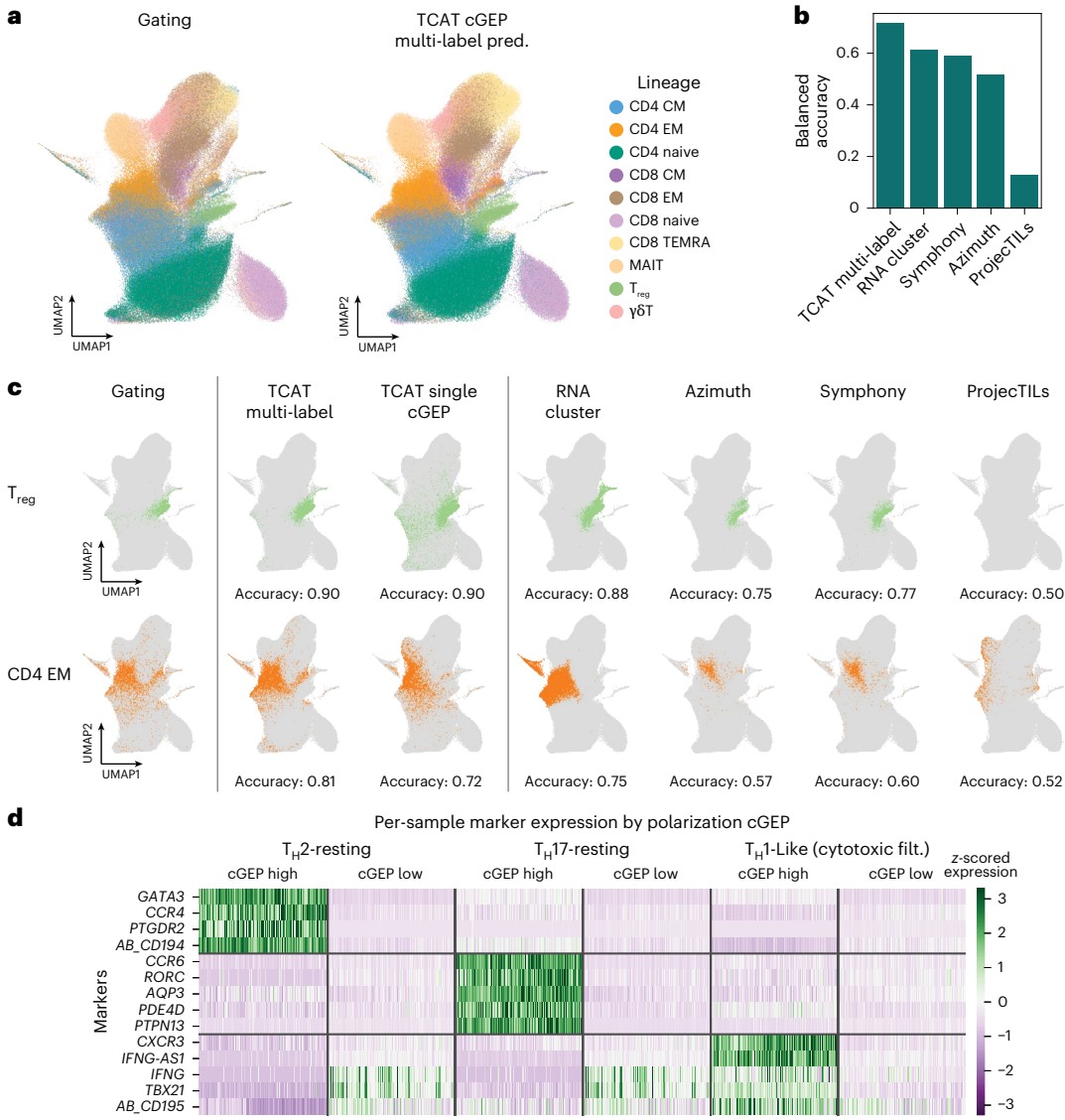

**Fig. 2 | Benchmarking TCAT on a query dataset. a**, UMAP of the flu-vaccine dataset colored by manual gating (Extended Data Fig. 3a) and TCAT multinomial label prediction. **b**, Cross-method comparison of balanced accuracy for manually gated subset prediction. **c**, Same UMAP as **a** but demonstrating prediction of manually gated T$_{reg}$ and CD8 EM populations with the most associated individual cGEP (usage > 0.025), the multi-label classifier based on multiple cGEPs, Leiden clustering with a resolution of 1.0 and three reference mapping algorithms. **d**, Heat map of pseudobulk expression in cGEP-high (usage > 0.1) and cGEP-low (usage < 0.1) cells, per sample. Pseudobulk profiles were normalized by library size and rows were z-scored.

CD4 central memory (CD4 CM) and terminally differentiated effector memory (TEMRA) subsets ($P$ value < $1 \times 10^{-200}$, coefficient > 0.35). The CD4 EM subset was associated with T$_H$17-resting and T$_H$1-like cGEPs as expected ($P < 4.1 \times 10^{-189}$, coefficient > 0.22). In total, we identified 17 subset-associated cGEPs.

We also identified likely technical artifact cGEPs (Supplementary Table 4). A mitochondria cGEP marked by mitochondrially transcribed genes correlated with per-cell mitochondrial transcript fraction (average $R = 0.81$ across datasets), a common quality-control metric in scRNA-seq[22,23]. Another cGEP, labeled 'poor-quality', was marked by *MALAT1*, a long noncoding RNA linked to poor cell viability[24]. Its usage correlated with mitochondrial transcript fraction (mean $R = 0.25$ across datasets), inversely with the fraction of protein-coding transcripts per cell (mean $R = -0.50$) and positively with the percentage of intergenic reads per cell (mean $R = 0.74$; Extended Data Fig. 2d–f). Thus, it may be driven by contaminating DNA or nascent RNA. We also flagged immediate early gene cGEPs as potentially technical in nature (Supplementary Note and Supplementary Fig. 4).

## Benchmarking TCAT on an independent query dataset

Next, we benchmarked TCAT for predicting discrete T cell subsets in a query CITE-seq dataset (labeled 'flu-vaccine'), containing 336,739 T cells from PBMCs of 24 COVID-19-recovered and 17 healthy individuals following influenza vaccine[25]. We defined ten conventional T cell subsets via manual surface protein gating to serve as prediction targets (Extended Data Fig. 3a). While subsets largely separated on a gene expression uniform manifold approximation and projection (UMAP), memory populations overlapped substantially, possibly due to shared functional GEPs (Fig. 2a). We hypothesized that predictions based on TCAT, which disentangles subset and functional cGEPs, would outperform methods unable to distinguish these signals.

Indeed, TCAT enabled more accurate subset prediction than RNA-based clustering or the discrete reference mapping tools Azimuth[10], Symphony[26] and ProjecTILs[3] (Methods). Subset assignment by thresholding the most associated cGEP performed comparably to reference mapping and clustering across nine resolutions for predicting one lineage at a time (Supplementary Fig. 5a). For simultaneous

multi-label prediction, we trained a multinomial logistic classifier on the COMBAT reference and measured performance with balanced accuracy (which weights classes of different sizes equally) in the flu-vaccine query (Methods and Fig. 2a). This greatly outperformed all tested reference mapping methods and clustering (balanced accuracy—TCAT, 0.72; Clustering, 0.61; Symphony, 0.58; Azimuth, 0.52; ProjecTILs, 0.13; Fig. 2b,c and Extended Data Fig. 3b).

We compared the performance of this multi-label classifier when trained using TCAT's cGEP catalog versus previously published GEP catalogs from NMF analyses of T cells in autoimmune diseases[11] and tumors[14] (Methods). TCAT's catalog yielded better prediction accuracy for all lineages (Extended Data Fig. 3c and Supplementary Fig. 5b). These analyses show that TCAT can predict peripheral T cell subsets without manual annotation and with accuracy surpassing clustering and leading reference mappers.

We also found that usage of the CellCycle-S, CellCycle-G2M and mitochondrial cGEPs correlated well with common, gene-set-based estimates of these programs, including published proliferation gene sets[27] ($R > 0.75$; Extended Data Fig. 3d and Methods).

Next, we validated prediction of T cell polarization against canonical marker expression. We discretized cells based on the $T_H1$-like, $T_H2$-resting and $T_H17$-resting cGEPs and computed per-sample pseudobulk profiles of high (usage > 0.1) and low (usage < 0.1) cells. $T_H2$-resting-high cells had significantly higher expression of $T_H2$ markers (GATA3, CCR4, PTGDR2) and analogously for $T_H17$-resting-high cells and $T_H17$ markers (CCR6, RORC, AQP3; $P < 1 \times 10^{-35}$ all, paired $t$-test; Fig. 2d). $T_H1$-like-high cells had increased expression of $T_H1$ markers (CXCR3, IFNG-AS1, CD195 protein; $P < 1 \times 10^{-35}$ all), although IFNG and TBX21 were also expressed in $T_H1$-like-low cells (Extended Data Fig. 3e), potentially due to their expression in cytotoxic T cells[28,29]. Excluding cytotoxic-high cells illustrated significantly higher IFNG and TBX21 in $T_H1$-like-high cells ($P = 8.2 \times 10^{-13}$, $P = 9.6 \times 10^{-47}$).

### cGEPs capture multi-GEP T cell identities

Next, we illustrate how TCAT reveals cellular heterogeneity obscured by clustering in the COMBAT COVID-19 dataset. First, we examined cell cycle effects since they often mask subsets[30]. Regressing out cell cycle programs[31] does not always work well because it may remove correlated signals like activation.

While the published analysis of CD4 memory cells identified multiple proliferating subclusters, these did not correspond directly to subsets, except for one—CD4.TEFF.prolif.MKI67lo—that was enriched for the myeloid doublet cGEP (Fig. 3a,b) and reflects a likely myeloid doublet population (Supplementary Fig. 6a). By contrast, TCAT readily identified distinct proliferating subsets based on coexpression of cell cycle and subset cGEPs (Fig. 3c,d).

This enabled us to quantify subset proliferation rates. Most subsets had increased cell cycle usage in COVID-19 compared to healthy cells (Extended Data Fig. 4a). Proliferation rates were correlated between the COVID-19 datasets ($R = 0.80$, $P = 0.00021$ in COVID-19, $R = 0.56$, $P = 0.025$ in healthy). The most proliferative population expressed the $T_{PH}$ cGEP and likely corresponds to $T_{PH}$ cells recently identified in COVID-19 (ref. 32).

Analogous to the cell cycle, we found that poor-quality, cytotoxic and interferon-stimulated gene (ISG) cGEPs could also dominate clusters, obscuring subsets (Fig. 3b–d and Supplementary Fig. 6b). For example, ISGs drove the CD4.TEM.IFN.resp and CD4.Th.IFN.resp clusters, which contained cells using multiple subset cGEPs (Extended Data Fig. 4b). For example, CD4.Th.IFN.resp contained many cells that expressed the CD4-naive cGEP and expected naive subset markers, suggesting they were misclustered with memory cells due to the ISG signal (Supplementary Note and Supplementary Fig. 6c–e).

Clusters with high cytotoxic cGEP usage contained cells with high usage of many subset cGEPs including CD8 EM, TEMRA and γδT (Extended Data Fig. 4c). Cells coexpressing cytotoxic and subset cGEPs coexpressed the expected cytotoxicity and subset marker genes (Extended Data Fig. 4d). This illustrates how TCAT can reveal cytotoxic T cell heterogeneity.

TCAT revealed polarization via the $T_H1$-like, $T_H2$-resting and $T_H17$-resting cGEPs (Fig. 3c) while the published clustering lacked a $T_H2$ cluster, and only annotated $T_H1/T_H17$ subsets with a high resolution yielding 243 total subclusters. We observed the expected enrichment between $T_H1$ and $T_H17$ annotated subclusters and cells expressing the $T_H1$-like and $T_H17$-resting cGEPs, respectively (Fisher's exact test $P < 1 \times 10^{-100}$).

However, TCAT also identified polarization outside the canonical CD4 memory subsets (Fig. 3e). Across manually gated populations (Extended Data Fig. 2b), $T_H2$-resting was most enriched in CD8 CM (15.7%) and CD4 CM (12.8%) subsets, while $T_H1$-like was enriched in CD8 CM (15.7%), CD4 EM (14.7%), CD8 EM (14.4%) and MAIT populations (12.3%). By contrast, the $T_{reg}$ cGEP was most enriched in the expected $T_{reg}$ subset (88.1%) and $T_H17$-resting in the expected CD4 EM (22.1%) and CD4 CM (10.7%) populations. Subset polarization proportions across subsets correlated strongly between the COMBAT and flu-vaccine datasets ($R > 0.9$, $P < 5.5 \times 10^{-5}$ all; Extended Data Fig. 4e). Furthermore, cells expressed the expected surface markers for their polarization, irrespective of CD4/CD8 lineage (Extended Data Fig. 4f), illustrating how TCAT can reveal polarized CD8$^+$ T cell populations[33].

### cGEPs associated with TCR-dependent activation

Next, we identified cGEPs induced by antigen-specific TCR activation using an activation-induced marker (AIM) assay followed by scRNA-seq (AIM-seq; Fig. 4a–d). We stimulated PBMCs from five healthy donors for 24 h using either a pool of 176 peptide antigens from common pathogens (CEFX, JPT)[34] plus anti-CD28/CD49d co-stimulation, or co-stimulation only (mock). Then, we sorted activated and non-activated T cells from the peptide stimulation using activation markers (OX40 and PD-L1 for the CD4 population[35] and CD137 for the CD8 population[36]). We labeled the resulting populations with hashtag antibodies and pooled them for CITE-seq and TCR-seq, totaling 42,370 cells (12,743 AIM-positive, 15,369 AIM-negative, 14,258 mock; Methods and Supplementary Fig. 7a). As expected, peptide stimulation substantially increased the percentage of AIM-positive cells (Fig. 4b and Extended Data Fig. 5a,b).

The data confirmed expected features of AIM-positive cells. First, they had increased expression of additional activation markers (CD54, CD25, CD71, CD69, $P < 1 \times 10^{-200}$; Extended Data Fig. 5c–e). They were also depleted of naive T cells (CD4: $P = 0.027$, CD8: $P = 8.6 \times 10^{-4}$) and enriched for $T_{reg}$ cells and CD4 CMs and EMs ($P = 0.00064$, 0.0044 and 0.054; Extended Data Fig. 5f), consistent with expected memory to pathogens in the pool. However, we could still detect some naive T cell activation (11.8% and 1.4% of AIM-positive cells were CD4 and CD8 naive, respectively). Clonal expansion (defined as 2+ cells sharing a CDR3 beta sequence) was significantly more common in AIM-positive than AIM-negative CD4 memory cells ($P = 2.1 \times 10^{-7}$ CD4, $P = 0.14$ CD8; Supplementary Fig. 7b) and clones were significantly more likely to share AIM status than expected by chance (77% agreement versus 51% expected; binomial $P < 1 \times 10^{-200}$).

Next, we identified cGEPs that increased following ASA using sample-level pseudobulk association tests (Fig. 4e, Extended Data Fig. 5g and Methods). Twenty-four cGEPs increased in AIM-positive relative to AIM-negative cells (FDR-adjusted $Q < 0.05$). Of these, we labeled the ISG and metallothionein cGEPs as milieu regulated as they increased in both AIM-negative and AIM-positive cells relative to mock ($P < 1.5 \times 10^{-3}$ all). We suspect these cGEPs are upregulated by interferon and extracellular ion concentration sensing[37], independent of TCR engagement.

Five subset cGEPs were higher in AIM-positive cells ($T_H17$-resting, $T_{reg}$, $T_{PH}$, $T_H22$, $T_{FH}$-2) while three were higher in AIM-negative cells (CD8-naive, CD4-naive and $T_H1$-like), likely reflecting baseline differences in the number of peptide-reactive cells in these populations

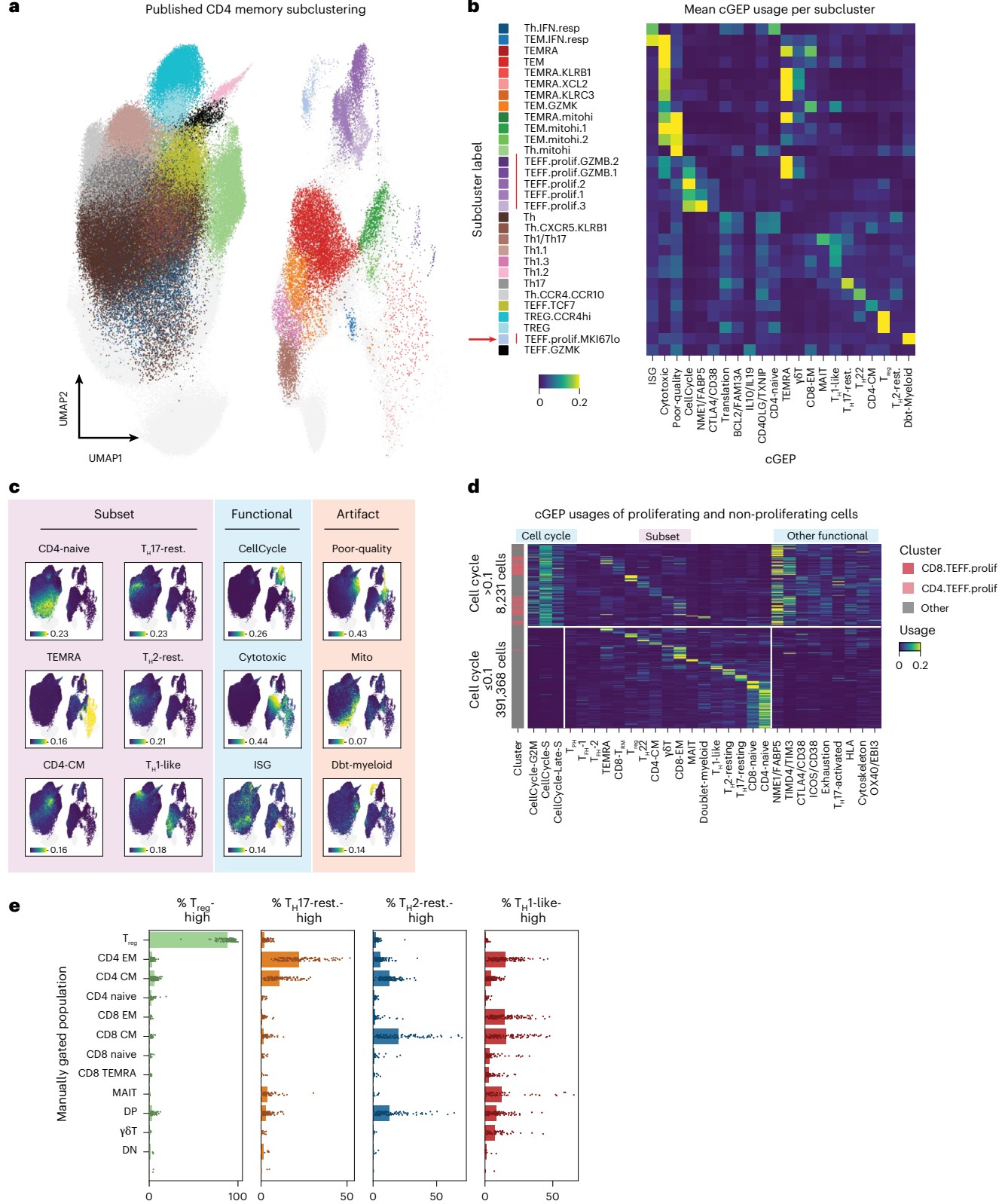

**Fig. 3 | Comparing TCAT to clustering in the COMBAT dataset. a**, UMAP of T cells showing published CD4 memory subclusters with other clusters shown in gray. 'CD4' is omitted from cluster names for space. **b**, Average usage of selected cGEPs in CD4 memory subclusters. Subcluster labels for **a** align with the corresponding row labels. The three cell cycle cGEPs are summed and labeled CellCycle. A red arrow is shown next to the myeloid doublet subcluster and a red line is shown next to subclusters annotated as proliferating. **c**, Same UMAP as in **a** colored by usage of selected subset, functional and artifact cGEPs. Intensities were averaged over 20 nearest neighbors to reduce overplotting. **d**, Usage of selected cGEPs in cells with high or low cell cycle cGEP usage. Cells were grouped by their most highly used subset cGEPs. **e**, Percentage of cells within each manual gate assigned to each polarization based on usage > 0.1. Points show the per-sample proportions ($n = 137$ samples). Bar represents the average across samples.

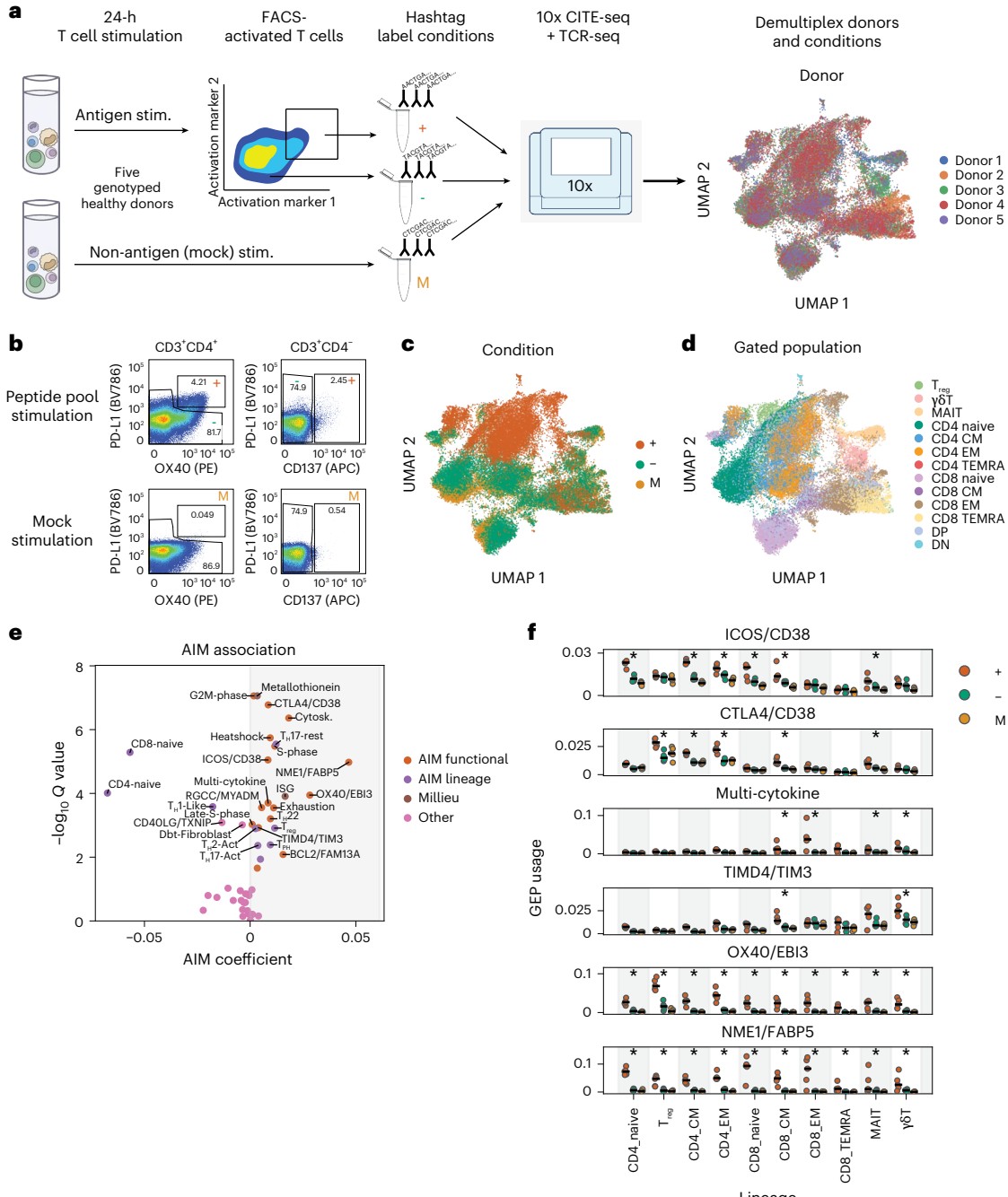

**Fig. 4 | Identifying cGEPs associated with TCR-dependent activation.**
**a**, Schematic of AIM-seq and UMAP of resulting data colored by donor. **b**, Flow cytometry data from an AIM-seq run showing surface activation markers in CD3⁺CD4⁺ and CD3⁺CD4⁻ gated populations with the gates used for AIM-positive (+), AIM-negative (−) and mock (M) populations. **c,d**, UMAP of AIM-seq dataset colored by sorting condition (**c**) or manually gated subset based on CITE-seq (**d**). **e**, cGEP association with AIM positivity. *x* axis shows the regression coefficient.

*y* axis shows the −log₁₀ false discovery rate (FDR)-corrected two-tailed *P* value (that is, *Q* value). cGEPs are labeled by assigned category. **f**, Average usage of selected AIM-associated cGEPs in +, − and M cells from different gated subsets, per sample. Lines show the median. *P* values are from a per-sample two-tailed rank-sum test comparing '+' with '−' and 'M' samples, for each lineage. *\**P* < 0.05 and average usage in '+' cells > 0.01.

rather than cGEP upregulation (Supplementary Table 5 and Extended Data Fig. 5f).

The 17 remaining AIM-associated cGEPs included several with clear links to TCR stimulation including cell cycle[38], actin cytoskeleton[39], heatshock[40,41] and major histocompatibility complex class II[42]. Additionally, 11 functional AIM-associated cGEPs may be specific to T cell activation, including CTLA4/CD38, ICOS/CD38, NME1/FABP5, OX40/EBI3, multi-cytokine, exhaustion, TIMD4/TIM3, T_H2-activated,

T_H17-activated and BCL2/FAM13A (see Supplementary Note and Supplementary Fig. 7c for more details). Several were most upregulated in specific subsets, such as multi-cytokine in CD8 memory; TIMD4/TIM3 in CD8 CM and γδT; CTLA4/CD38 in T_reg, CD4 memory and CD8 CM (Fig. 4f); and OX40/EBI3 in tumor-infiltrating T cells (Supplementary Fig. 7d).

Because proliferation is a core response to TCR activation, we tested if AIM-associated cGEPs were enriched in proliferating cells

in vivo. Results were concordant across datasets, with 15 cGEPs increased in cell cycle-high cells (aggregate usage > 0.1) in at least four of six datasets (Extended Data Fig. 5h, Supplementary Table 6 and Supplementary Fig. 7e). Of these, 14 were AIM-associated (Fisher exact test $P = 2.1 \times 10^{-5}$), further supporting a role for these cGEPs in TCR activation in vivo.

## Annotating antigen-dependent activation in disease

Next, we developed a per-cell antigen-specific activation (ASA) score to identify TCR-activated T cells in disease. Using forward stepwise selection, we identified four AIM-associated cGEPs (TIMD4/TIM3, ICOS/CD38, CTLA4/CD38 and OX40/EBI3) that together predict CD71/CD95 coexpression in the COMBAT and flu-vaccine datasets (Methods, Extended Data Fig. 6a,b and Supplementary Note). We selected CD71 and CD95 as activation markers because they are known to be upregulated within 24 h of TCR activation[43–46], were upregulated in the AIM assay (Extended Data Fig. 5c–e) and had high quality across subsets in both datasets.

ASA effectively predicted CD71/CD95 coexpression in vivo (COMBAT: area under the curve (AUC) = 0.920, flu-vaccine: AUC = 0.818) and AIM positivity in the AIM-seq data (AUC = 0.828; Extended Data Fig. 6c–e). It also correlated with expression of other surface activation markers (CD69: $R = 0.43$, CD25: $R = 0.52$, $P < 1 \times 10^{-100}$; Supplementary Fig. 8a). We chose a discrete ASA threshold of 0.0625 by balancing sensitivity and specificity (Extended Data Fig. 6c–e). This threshold resulted in a positive call for 76.7% of CD71$^+$CD95$^+$ and 5.2% of non-CD71$^+$CD95$^+$ T cells in the COMBAT dataset, and 60.6%, 7.0% and 3.2% of AIM-positive, AIM-negative and mock-stimulated cells in AIM-seq (Fig. 5a,b).

We benchmarked ASA against literature-derived T cell activation gene sets for predicting surface activation profiles. ASA outperformed 9/9 and 7/9 tested gene sets in the flu-vaccine and AIM-seq datasets, respectively, demonstrating its utility relative to a widely used approach (Extended Data Fig. 6f,g and Supplementary Fig. 8b,c).

ASA-high cells had several expected features of antigen activated cells in vivo. They were enriched in proliferating clusters (Fisher's exact odds ratio (OR) 2.8–58.8 across datasets, $P < 1 \times 10^{-100}$ all) and ASA correlated with cell cycle usage (mean $R$ 0.15; Fig. 5c,d and Extended Data Fig. 6h). However, ASA identified significantly more activated cells than the cell cycle alone, indicating greater sensitivity for classifying activation ($P = 8.8 \times 10^{-189}$, two-tailed paired $t$-test; Fig. 5e).

ASA-high cells were more likely to be clonally expanded in both COVID-19 datasets (COMBAT OR = 2.50, UK-COVID OR = 2.28, $P < 1 \times 10^{-100}$ for both). Furthermore, ASA and cell cycle were independently associated with clonal expansion in a multivariate logistic regression (beta values: ASA, 0.45 and 0.50; cell cycle, 0.66 and 0.52, in COMBAT and UK-COVID respectively; $P < 1 \times 10^{-22}$; Methods). The TCR

clone size distribution was also shifted upward in ASA-high relative to ASA-low cells ($P < 1 \times 10^{-100}$, both datasets; Fig. 5f and Extended Data Fig. 6i).

There were significantly more ASA-high cells in COVID-19 than healthy samples, consistent with viral activation (COMBAT: $P = 1.9 \times 10^{-7}$, UK-COVID: $P = 1.5 \times 10^{-6}$; Fig. 5g). ASA rates were comparable in CD4 and CD8 conventional subsets but higher in $T_{reg}$ cells for both healthy and COVID-19 samples (Fig. 5h and Supplementary Fig. 8d–f). In COVID-19, ASA-high cells were enriched in CD8 CM, CD8 EM and DN subsets (OR = 4.8, 2.8 and 3.1 respectively, all $P < 1 \times 10^{-10}$), although not in healthy samples, likely reflecting the antiviral response. ASA rates were higher in UK-COVID than COMBAT. This correlated with differences in sample quality reflected in poor-quality cGEP usage and library size and may reflect nonspecific activation related to sample processing (Supplementary Fig. 9a).

To further illustrate analyses enabled by TCAT, we characterized variation in T cell exhaustion and activation in breast cancer (BC), esophageal cancer (ESCA), hepatocellular carcinoma (HCC), pancreatic cancer (PACA), renal cell carcinoma (RC), thyroid carcinoma (THCA) and endometrial cancer (UCEC) (Fig. 5i). ASA positivity in TCAT-annotated CD4 cells ranged from 5.4% (breast) to 48.0% (esophageal). This correlated with analogous rates in CD8 cells for ASA ($R = 0.70$, $P = 2.6 \times 10^{-9}$) and exhaustion ($R = 0.38$, $P = 4.0 \times 10^{-3}$; Extended Data Fig. 6j) across tumor types. $T_{reg}$ cells had significantly higher ASA positivity in thyroid ($P = 3.0 \times 10^{-6}$) and esophageal ($P = 0.0045$) cancer relative to matched normal tissues (Extended Data Fig. 6k). As expected, tumor mutation burden was significantly correlated with the percentage of exhausted CD8$^+$ T cells per tumor (Spearman $\rho = 0.59$, $P = 6.9 \times 10^{-8}$; Fig. 5j and Supplementary Fig. 9b).

Many tumors included T cells with low ASA and exhaustion usage ('bystanders'). CD8 bystanders varied widely from 35.5% (endometrial) to 90.1% (breast) of total CD8 cells. Bystanders were enriched within populations marked by usage of the CD4-naive (OR = 15.9), $T_H2$-resting (OR = 10.6), $T_H1$-like (OR = 7.3), MAIT (OR = 4.42) and CD8-naive (OR = 4.03) cGEPs and were most depleted from $T_{PH}$ (OR = 0.19), $T_{reg}$ (OR = 0.23) and CD8 $T_{RM}$ (OR = 0.61) populations ($P < 1 \times 10^{-21}$ all; Fig. 5k). These analyses illustrate how TCAT and ASA scoring can enable disease exploration.

## Identifying disease-associated cGEPs

Next, we associated cGEPs with cancer, COVID-19 and rheumatoid arthritis phenotypes (Supplementary Table 7, Extended Data Fig. 7a–f and Supplementary Note). Using pseudobulk sample-level regression in the pan-cancer dataset (89 tumor, 47 matched normal samples, 13 cancer types), we identified $T_{reg}$[47], exhaustion[48] and ISG[49] as strongly tumor-associated, consistent with their known role in cancer (FDR-corrected $Q = 7.4 \times 10^{-12}$, $8.5 \times 10^{-6}$ and $9.3 \times 10^{-6}$, respectively).

**Fig. 5 | Annotating ASA in vivo. a**, Box plot of ASA score for cells stratified as activated (that is, CD71$^+$CD95$^+$, $N = 24,341$ cells) or not activated ($N = 375,258$ cells). Boxes represent the interquartile range and whiskers represent 1.5 times the interquartile range. The box center line indicates the median. **b**, Same as **a** but for AIM-seq data with cells stratified by sort condition (+: $N = 13,235$ cells; −: $N = 15,528$ cells; M: $N = 14,459$ cells). **c,d**, UMAP of the COMBAT dataset colored by published clustering (**c**) and ASA score (**d**). **e**, Percentage of activated (ASA > 0.065) or proliferating (sum of cell cycle cGEPs > 0.1) cells per sample across datasets. Boxes represent the interquartile range and whiskers represent the 95% quantile range. (AMP-RA, $N = 162$ samples; COMBAT, $N = 244$; HIV-vaccine, 16; pan-cancer, 272; pan-tissue, 24; Sparks, 82; AIM-seq, 10; TBRU, 518; UK-COVID, 242). **f**, Clonality in manually gated conventional CD4$^+$ and CD8$^+$ T cells annotated as activated (ASA > 0.065) or not activated (ASA < 0.065). Clonality is defined as the number of cells in the same sample with an identical alpha and beta CDR3 amino acid sequence. **g**, Percentage of activated (ASA > 0.065) CD4$^+$ and CD8$^+$ conventional T cells in COVID-19 and healthy control samples, by cohort. Boxes represent the interquartile range and whiskers represent 1.5 times the interquartile range. The box center line indicates

the median. (COMBAT, $N = 77$ COVID-19, 10 healthy; UK-COVID, $N = 80$ COVID-19, 21 healthy). **h**, log$_2$ OR for $2 \times 2$ association of ASA positivity and manual gating subset assignment. An asterisk indicates Bonferroni-adjusted two-tailed Fisher's exact test $P$ value < 0.05. **i**, Percentage of activated (ASA > 0.065), exhausted (exhaustion cGEP usage > 0.065) or bystander (ASA + exhaustion usage < 0.065) T cells in CD4$^+$ and CD8$^+$ conventional T cells, per sample stratified by tumor type and corresponding healthy tissues. Boxes represent the interquartile range and whiskers represent 1.5 times the interquartile range. The box center line indicates the median. (BC: $n = 2$ tumor, $n = 2$ normal; ESCA: $n = 7$ tumor, $n = 7$ normal; HCC: $n = 5$ tumor, $n = 5$ normal; PACA: $n = 26$ tumor, $n = 1$ normal; RC: $n = 10$ tumor, $n = 11$ normal; THCA: $n = 10$ tumor, $n = 8$ normal; UCEC: $n = 9$ tumor, $n = 8$ normal). BC, breast cancer; ESCA, esophageal cancer; HCC, hepatocellular carcinoma; PACA, pancreatic cancer; RC, renal cell carcinoma; THCA, thyroid carcinoma; UCEC, endometrial cancer. **j**, Per-individual comparison of percentage of exhausted CD8$^+$ T cells and average number of mutations per mutational burden (MB). **k**, log$_2$ OR for enrichment of bystander T cells by subset cGEP assignment. Bar value reflects the estimated OR, while error bars represent the analytical 95% confidence intervals around the estimate.

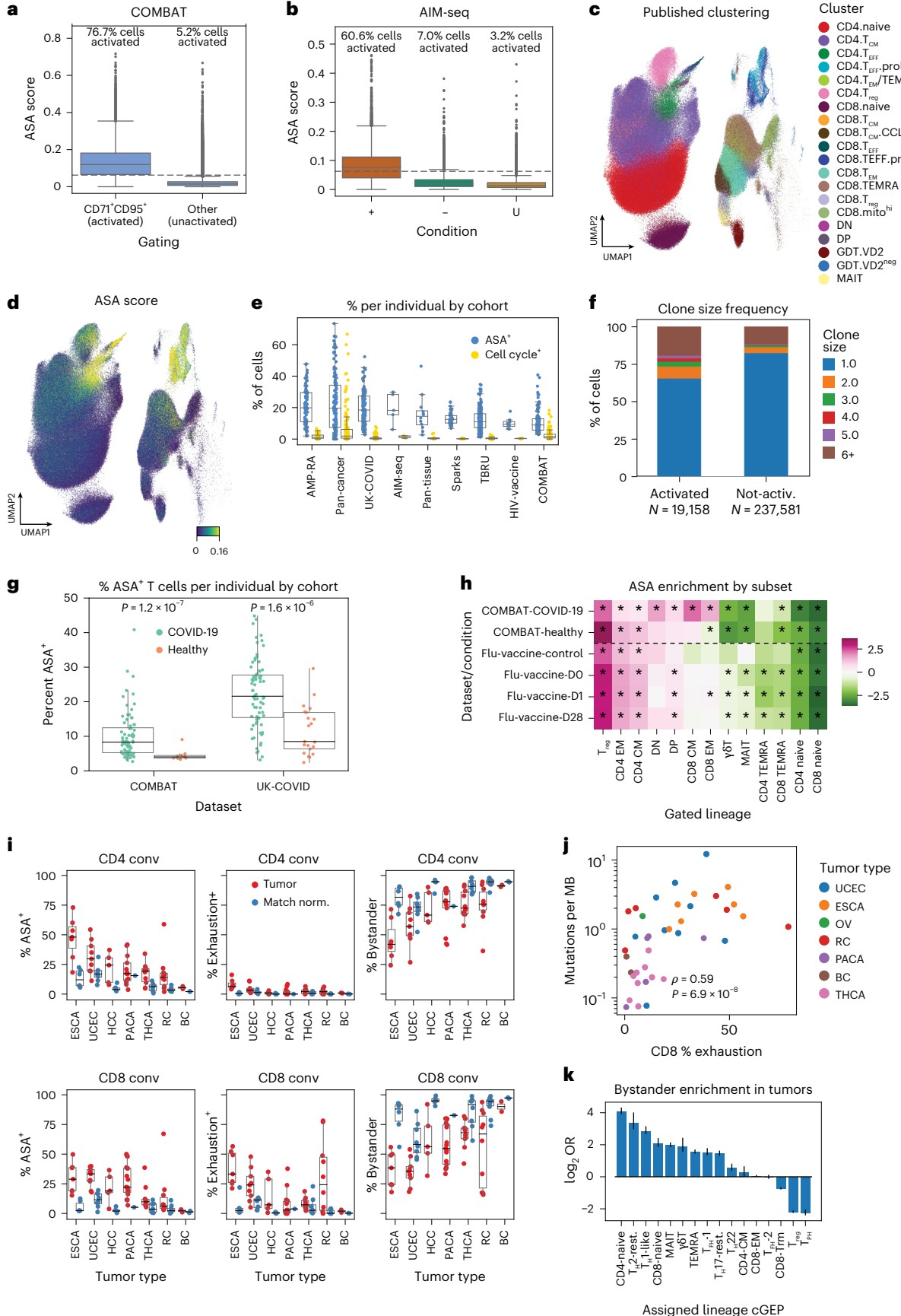

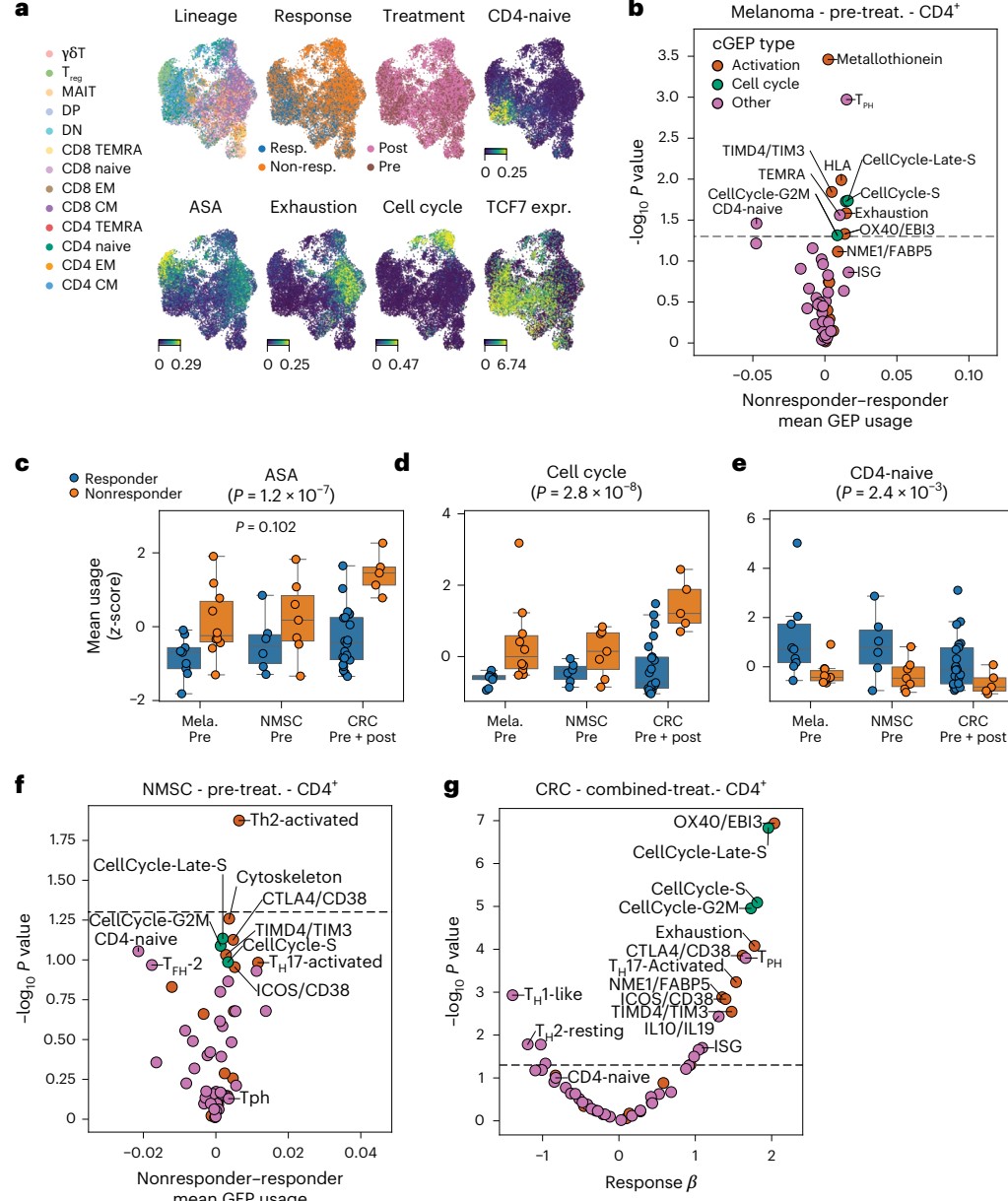

**Fig. 6 | cGEPs associated with ICI response. a,** UMAPs of the melanoma dataset showing TCAT predicted lineage; the CD4-naive and exhaustion cGEPs; the ASA and cell cycle scores; *TCF7* expression; and treatment status and response. **b,** Associations of cGEP usage with ICI response in CD4⁺ T cells of pretreatment melanoma. Dots are colored by cGEP type. *x* axis shows the average difference in usage between nonresponders and responders. *y* axis shows the −log₁₀ two-tailed *P* value. **c–e,** Average ASA score, cell cycle score and CD4-naive cGEP usage in CD4⁺ T cells from pretreatment melanomas and NMSC tumors and combined pretreatment and post-treatment CRC. The average scores are mean and variance normalized. *P* values are one-tailed *t*-tests for melanoma and NMSC

and mixed linear regression *P* values for CRC. *P* value in the title is a meta-analysis of the *P* values across the three cancer types. *\*P* < 0.05. Boxes represent the interquartile range and whiskers represent 1.5 times interquartile range. The box center line indicates the median. Sample sizes shown are *n* = 9, *n* = 6 and *n* = 15 for responders in melanoma, NMSC and CRC, respectively, and *n* = 10, *n* = 7 and *n* = 4 in nonresponders in melanoma, NMSC and CRC. **f,** Same as **b** in pretreatment NMSC. **g,** Same as **b** in combined pretreatment and post-treatment CRC, but showing coefficients and *P* values from mixed linear regression analysis, controlling for treatment time point and donor of origin.

Of 21 tumor-enriched cGEPs, 17 were AIM-associated (Fisher exact test *P* = 7.4 × 10⁻⁶).

We separately analyzed individual tumor types with ≥2 tumor and normal samples each (Methods). Results were highly concordant across cancers (sign test *P* < 0.05 for 14/15 tumor-type pairs; Extended Data Fig. 7b). T_reg, exhaustion and CTLA4/CD38 cGEPs were upregulated in all six tumor types tested (*P* < 0.05). However, some signals were more specific including T_H17-activated (thyroid: *P* = 5.3 × 10⁻⁶, hepatocellular: *P* = 0.013) and T_H2-activated cGEPs (esophageal, uterine, thyroid and hepatocellular: *P* < 0.05 all).

Of note, the T_FH-2 and T_PH cGEPs were both upregulated in cancer (*Q* = 3.6 × 10⁻⁴, *Q* = 3.3 × 10⁻¹⁰). T_FH and T_PH cells recruit B cells via *CXCL13* aiding in antibody production. T_FH cells are found primarily in lymphoid organs and T_PH cells are predominantly in inflamed tissues[50], including likely within tumors[51]. T_PH cell cGEP usage was associated with *CXCL13* expression and plasma cells abundance across tumors, indicating a role in tumor-associated lymphoid aggregates (Supplementary Fig. 9c–e and Supplementary Note).

cGEP associations with COVID-19 status revealed consistent associations between the two reference datasets (*R* = 0.64, *P* = 2.8 × 10⁻⁷)

highlighting $T_{PH}$ cells and ASA cGEP involvement ($Q < 0.05$; Extended Data Fig. 7c–e and Supplementary Note). Rheumatoid arthritis similarly showed increased usage of metallothionein, HLA, ICOS/CD38, $T_{PH}$ cells and other activation-associated cGEPs ($Q < 0.05$; Extended Data Fig. 7f and Supplementary Note).

#### Characterizing ICI response

We next demonstrate TCAT's utility by identifying cGEPs that predict tumor response to immune checkpoint inhibitors (ICIs). ICIs are state-of-the-art therapies for treating many types of cancer, yet 5-year survival remains poor for over half of treated patients[52]. To investigate T cell states associated with ICI response, we applied TCAT to melanoma[53], non-melanoma skin cancer (NMSC)[54] and colorectal cancer (CRC)[55] datasets containing responder and nonresponder tumors before and after treatment.

We first examined melanoma as the largest dataset containing 19 pretreatment and 48 total samples. TCAT revealed populations expressing ASA, exhaustion, cell cycle and CD4-naive signatures (Fig. 6a). We also noted a prominent subset of cells expressing *TCF7*, which was previously associated with ICI response in this dataset[53].

In melanoma pretreatment tumors, CD4+ T cells from nonresponders had significantly higher activation (for example, TIMD4/TIM3, OX40/EBI3, HLA) and cell cycle cGEP usage ($P < 0.05$ two-tailed $t$-test; Fig. 6b and Supplementary Table 8). Associations were concordant in pretreatment NMSC samples (sign test $P = 0.0016$; Extended Data Fig. 8a), including significant associations for CellCycle-Late-S and TIMD4/TIM3 (one-tailed $P = 0.037$ and $0.046$, respectively; Fig. 6c–f). Meta-analysis of associations between pretreatment melanoma and NMSC tumors was significant for the combined ASA and cell cycle scores ($P = 0.0072$ and $P = 0.0036$, respectively; Fig. 6c–e) and for many cGEPs (for example, CellCycle-Late-S, exhaustion, ICOS/CD38; $P < 0.05$ all). Thus, elevated pretreatment TCR activation and proliferation signatures predict a worse response to ICIs in these tumor types.

Responders also exhibited higher usage of the CD4-naive cGEP in pretreatment melanoma and NSMC tumors (meta-analysis $P = 0.0063$; Fig. 6e). This aligns with prior evidence linking *TCF7* to improved ICI outcome, as *TCF7* is a top marker of the CD4-naive cGEP (Supplementary Table 2). The naive T cells markers *TCF7*, *CCR7* and *SELL* all had higher expression in pretreatment responders than nonresponders (meta-analysis $P = 0.024$, $0.0016$ and $0.00047$, respectively; Extended Data Fig. 8b–d). Furthermore, the proportion of TCAT-classified naive CD4+ T cells was similarly predictive of response (meta-analysis $P = 0.016$; Extended Data Fig. 8e), suggesting infiltrating naive CD4 cells may predict positive ICI response.

The CRC dataset only had one pretreatment nonresponder sample, which precluded association testing in pretreatment tumors. However, we observed concordant associations between pretreatment and post-treatment samples across the three tumor types (Extended Data Fig. 8f–h and Supplementary Fig. 10a,b). For example, response-associated cGEPs ($P < 0.05$) were significantly concordant between pretreatment and post-treatment melanoma samples (Fisher's exact test $P = 0.0039$).

Assuming pretreatment and post-treatment samples share many immune states, we repeated associations using combined pretreatment and post-treatment samples, modeling treatment status with a fixed effect and patient of origin with random effects (Methods). This yielded consistent results with our prior findings, including increased activation and cell cycle cGEPs in CRC nonresponders (for example, $P < 1 \times 10^{-6}$ for OX40/EBI3, CellCycle-Late-S and ASA; Fig. 6g and Supplementary Fig. 10c,d). These distinct analyses support that our findings are robust and reproducible in three cancer types.

We obtained similar results in CD8+ T cells (Extended Data Fig. 8i–k). Activation and cell cycle cGEPs were elevated in pretreatment nonresponders of melanoma and NMSC (meta-analysis $P < 0.05$ for CellCycle-G2M, CellCycle-Late-S, TIMD4/TIM3, exhaustion, ICOS/ CD38, OX40/EBI3 and HLA) and in the combined CRC cohort ($P < 0.05$ for all). The CD8-naive cGEP was associated with positive ICI response in pretreatment melanoma and NMSC samples (meta-analysis $P = 0.032$). These analyses highlight TCAT's capacity to reveal clinically meaningful immune patterns across multiple datasets.

## Discussion

Here, we introduced starCAT, a tool that leverages the reproducibility of functionally informative cGEPs across datasets to annotate new scRNA-seq data. We illustrated starCAT with the most comprehensive T cell GEP catalog to date, including 16 subset-associated and 25 functional cGEPs derived from reference datasets spanning many tissues and diseases. Combining this catalog with starCAT yields TCAT.

TCAT offers key advantages over standard approaches. It simultaneously annotates functional and subset GEPs, disentangling conflated signals and revealing unexpected populations like abundant $T_H2$ polarized CD8+ T cells[33]. It outperformed RNA-based clustering and reference label transfer methods for subset annotation and enabled facile disease activity comparisons. TCAT is also much faster than de novo matrix factorization, avoids the need for GEP annotation and improves GEP inference accuracy for smaller datasets.

We developed the AIM-seq assay to identify cGEPs induced following TCR-dependent activation. This identified 24 TCR activation-associated cGEPs including context-specific responses like multi-cytokine in CD8 memory and CTLA4/CD38 predominantly in CD4 memory cells. We aggregated several of these cGEPs into an ASA score to identify activation in scRNA-seq data. This revealed numerous 'bystanders' within tumors that lacked either activation or exhaustion signatures and were enriched for naive and unconventional T cell subsets.

TCAT uncovered features of tumor-infiltrating T cells that predict ICI response. Surprisingly, nonresponsive tumors were enriched for activation, cell cycle and exhaustion cGEPs. By contrast ICI responsive samples were enriched for the CD4-naive cGEP and were predicted to contain more naive CD4+ T cells. These findings suggest that TCAT can help predict therapy response and characterize clinically relevant cell states.

Limitations of this work include that the catalog may be missing GEPs that are disease or tissue-context specific and were filtered due to non-reproducibility across datasets. Our AIM-seq experiment used only a single co-stimulation signal and set of microbial peptides and thus may not have identified all possible activation-associated GEPs. Future work incorporating additional datasets and stimulation conditions can address both limitations.

While demonstrated in T cells, starCAT is applicable to any cell type or tissue. We provide open-source software and a growing repository of GEPs, including human glioma myeloid[56] and bone marrow hematopoiesis references[57]. Similar to the molecular signatures database (MSigDB)[58,59] but for scRNA-seq annotation, the platform supports community-contributed GEP catalogs. We hope starCAT will enable comprehensive identification of GEPs across tissues and diseases.

## starCAT algorithm

Whereas cNMF learns both GEPs and their usage in cells, starCAT has the simpler problem of fitting the usage for a fixed set of GEPs. Specifically, cNMF runs NMF multiple times, each time solving the following optimization as given by equation (1):

$$\text{ArgMin}_{G,U} |X - UG|_F \tag{1}$$

where

$$U \geq 0, G \geq 0$$

where $X$ is a NxH matrix of N cells by the top H overdispersed genes, $U$ is a learned NxK matrix of the usages of K GEPs in each cell, and $G$ is a

learned K×H matrix where each row encodes the relative contribution of each highly variable gene in a GEP. H is usually a parameter set to ~2,000 overdispersed genes. $||_F$ denotes the Frobenius norm. $X$ includes variance-normalized overdispersed genes to ensure biologically informative genes are included and contribute similar amounts of information even when they may be expressed on different scales. For cNMF, the optimization is solved multiple times and the resulting $G$ matrices are concatenated, filtered and clustered to determine a final average estimate of $G$. Ultimately cNMF refits the GEP spectra into two separate representations, one reflecting the average expression of the GEP in units of transcripts per million $G^{tpm}$ and one in $z$-scored units used to define marker genes $G^{scores}$ (see ref. 12 for details).

By contrast, starCAT takes a fixed catalog of GEPs as input, denoted as $G^*$, and a new query dataset $X^{query}$ and solves the optimization as given by equation (2):

$$\text{ArgMin}_U |X^{query} - UG^*|_F \qquad (2)$$

where

$$U \geq 0$$

The columns of $X^{query}$ and $G^*$ correspond to a prespecified set of overdispersed genes. Analogous to cNMF, we use gene-wise standard-deviation-normalized counts for $X^{query}$. See below for how $G^*$ is calculated for TCAT. We solve for $U$ with nonnegative least squares using the NMF package in scikit-learn (version 1.1.3)[60] with $G^*$ fixed. We use the Frobenius error, the multiplicative update ('mu') solver, a tolerance of $1 \times 10^{-4}$ and maximum iterations of 1,000. We then perform row normalization of the $U$ matrix so that each cell's aggregate usage across all K GEPs sums to 1.

## Dataset preprocessing and batch-effect correction

To generate the input matrix for cNMF for each dataset, we first filtered genes detected in fewer than ten cells and cells with fewer than 500 unique molecular identifiers. We also excluded antibody-derived tags (ADTs) and genes containing a period in their gene name. We subsequently subsetted the data to the top 2,000 most overdispersed genes, identified by the 'seurat_v3' algorithm as implemented in Scanpy[61]. Next, we scaled each gene to unit variance. To avoid outliers with excessively high values, we calculated the 99.99th percentile value across all cells and genes and set this as a ceiling. We denote this matrix as $X^{raw}$.

We used an adapted version of harmonypy to correct batch effects and other technical variables from $X^{raw}$ before cNMF[20]. For this, we computed Harmony's maximum diversity clustering matrix from principal components calculated from a normalized version of $X$, which we label $X^{norm}$. Specifically, to compute $X^{norm}$, we started from the same initial gene list described above but first normalized the rows of the matrix so that each cell's counts sum to 10,000 (TP10K normalization). We then subsetted to the top 2,000 overdispersed genes, and scaled each column (gene) to unit variance, resulting in $X^{norm}$. We then performed PCA on $X^{norm}$ and supplied those principal components to the run_harmony function of harmonypy. We then used the mixture of experts model correction, implemented in harmonypy with the computed maximum diversity clustering matrix, but instead of correcting the principal components using this model, as standard Harmony does, we corrected $X^{raw}$. This creates a small amount of variability around 0 for the smallest values in $X^{raw}$. We therefore set a floor of 0, resulting in the corrected matrix $X^c$ used as the count matrix for cNMF.

## cNMF

We ran cNMF on the batch-corrected $X^c$ matrix, which only includes the top 2,000 overdispersed RNA genes. Spectra for the resulting GEPs were then refit by cNMF including all genes that passed the initial set of filters, including ADTs. Specifically, RNA counts were normalized to sum to 10,000, and ADT counts were separately normalized to sum

to 10,000 and the combined matrix was passed as the --tpm argument for cNMF. Thus, the GEP spectra output by cNMF incorporates ADTs and genes not included in the 2,000 overdispersed genes.

cNMF was run for each dataset with the number of components ($K$) varying between 15 and 55 and with 20 iterations. The final number of NMF components used for each dataset, $K^*$, was chosen by visualizing the trade-off between reconstruction error and stability for these runs (Supplementary Fig. 2). Once $K^*$ was selected, we ran cNMF a final time with only this value for $K$ and with 200 iterations to generate the final GEP spectra estimates.

## Constructing a catalog of cGEPs

Next, we identified consensus GEP spectra—that is, the average of correlated GEP spectra identified by cNMF in different datasets. Normalized input GEP vectors, denoted as $\mathbf{g}_i$, were computed by starting from the spectra_tpm output from cNMF, renormalizing each vector to sum to $10^6$, and then dividing each gene by its standard deviation in TP10K units. Then, we created an undirected graph where the 267 GEPs identified across all reference datasets were represented as nodes $\mathbf{g}_1 \ldots \mathbf{g}_{267}$. We drew edges, denoted as $E_{i,j}$ connecting a pair of GEPs $\mathbf{g}_i$ and $\mathbf{g}_j$ if the following criteria were met:

1. $\mathbf{g}_i$ and $\mathbf{g}_j$ were from different datasets
2. $R_{ij} > 0.5$ where $R_{ij}$ denotes the Pearson correlation between $\mathbf{g}_i$ and $\mathbf{g}_j$. For computing $R_{ij}$, $\mathbf{g}_i$ and $\mathbf{g}_j$ were subset to the union of the overdispersed genes for each dataset.
3. $\mathbf{g}_i$ was among the top seven most correlated GEPs with $\mathbf{g}_j$, and $\mathbf{g}_j$ was among the top seven most correlated GEPs with $\mathbf{g}_i$ with correlation defined as in 2.

Next, we initialized a set for each GEP: $x_1 = \{\mathbf{g}_1\} \ldots x_{267} = \{\mathbf{g}_{267}\}$. We then iterated through all edges $E_{i,j}$ in the graph in order of decreasing $R_{ij}$ and merged the sets $x_i$ and $x_j$ into a new set $x_{i,j} = \{\mathbf{g}_i, \mathbf{g}_j\}$. If either $\mathbf{g}_i$ or $\mathbf{g}_j$ were already members of a merged set from previous merges, we merged their containing sets only if at least two-thirds of the GEP pairs in the resulting consensus set were connected by edges. For example, if there is an edge $E_{4,9}$ and $\mathbf{g}_4$ is already merged into a set $\{\mathbf{g}_1, \mathbf{g}_2, \mathbf{g}_4\}$, then we only merged $\{\mathbf{g}_1, \mathbf{g}_2, \mathbf{g}_4\}$ and $\{\mathbf{g}_9\}$ if there were also edges $E_{1,9}$ and $E_{2,9}$. This resulted in 49 merged sets and 52 unmerged 'singleton' sets. We filtered 49 of the 52 singletons and retained 3 that had a biological explanation for being identified in only one dataset. This resulted in a final catalog of 52 cGEPs, including doublet programs.

Lastly, we subset each GEP to the union of overdispersed genes across all seven reference datasets that were present in all datasets and obtained the final consensus GEPs by taking the element-wise average GEPs in each merged set. This matrix was used as the reference for TCAT. For marker gene analyses (for example, Fig. 2b,d and Extended Data Fig. 2), we then averaged, element-wise, the $z$-score representation of the GEP output by cNMF for GEPs in a consensus set.

## Simulation analysis

We adapted the scsim simulation framework described previously[12] based on Splatter[62] into a new version, scsim2. Like with scsim, we distinguished between subset GEPs, which are mutually exclusive, and non-subset or 'activity' GEPs, which are not. For the original scsim framework, cells used one of multiple subset GEPs and potentially used a single-activity GEP. We adapted scsim to allow cells to use anywhere from none to all of the activity GEPs in addition to their single subset GEP. We kept the Splatter parameters used in the original publication to describe the distribution of gene expression data: mean_rate = 7.68, mean_shape = 0.34, libloc = 7.64, libscale = 0.78, expoutprob = 0.00286, expoutloc = 6.15, expoutscale = 0.49, diffexpprob = 0.025, diffexpdownprob = 0.025, diffexploc = 1.0, diffexpscale = 1.0, bcv_dispersion = 0.448 and bcv_dof = 22.087.

We simulated 10 subset GEPs and 10 activity GEPs based on 10,000 total genes. The extra-GEP reference included all 20, the missing-GEP

reference included 6 of the subset GEPs and 6 of the non-subset GEPs, and the query dataset included 8 subset GEPs and 8 non-subset GEPs. Each dataset consisted of 9,000 genes, randomly sampled from the total of 10,000. Each cell was randomly assigned a subset GEP with uniform probability, and each cell randomly selected whether it expressed each activity GEP with a probability of 0.3. The degree of usage of each activity GEP was sampled uniformly between 0.1 and 0.7. If the sum of the activity GEPs exceeded 0.8 for a cell, they were renormalized to sum to 0.8. Thus, each cell's usage of its subset GEP always exceeded 0.2. We simulated 100,000 cells each for the extra-GEP and missing GEP references. We simulated multiple query datasets containing 100, 500, 1,000, 5,000, 10,000, 20,000, 50,000 or 100,000 cells. The same parameters were used for Supplementary Fig. 1 but with different numbers of GEPs in the references and query.

We subsequently ran cNMF using 1,000 overdispersed genes, 20 iterations, local_neighborhood_size = 0.3 and density_threshold = 0.15. We used $K = 20$, $K = 12$ and $K = 16$ for the extra-GEP reference, missing-GEP reference and query datasets, respectively. We then used starCAT to fit the usage of the reference GEPs on the query dataset. To evaluate the performance of starCAT and cNMF, we calculated the Pearson correlation of the inferred GEP usage with the simulated ground-truth usage.

### Gene-set enrichment analysis

We used Fisher's exact test in Python's Scipy library to associate cGEPs with gene sets (Supplementary Note). For the T cell polarization dataset[63], we defined polarization gene sets as genes that had an FDR-corrected $P$ value < 0.05 and fold change > 2 with the stimulation condition. We excluded genes with FDR-corrected $P$ value between 0.05 and 0.2 and fold change > 1, as many of these are upregulated by the stimulation but just did not reach FDR significance. We also obtained literature gene sets corresponding to immediate early genes[64] and gene ontologies[65,66]. We tested these for enrichments with each cGEP thresholded with a $z$ score > 0.015, which corresponded to the 99th percentile across all genes and cGEPs, using Fisher's exact test as implemented in scipy.stats in Python.

### Manual subset gating analysis

We library size normalized ADT protein measurements to sum to $10^4$ (TP10K) and applied the centered log-ratio transformation. We then scaled each protein to unit variance, and set a ceiling of 15 to remove excessively high outliers. Next, we performed PCA and ran batch correction using harmonypy with the same batch features as for cNMF. We then computed the $K$-nearest neighbor graph with $K = 5$ neighbors, using the Harmony-corrected principal components. We then smoothed the normalized protein estimates using MAGIC[67] using the $K$-nearest neighbor graph and the diffusion operator powered to $t = 3$.

We gated canonical T cell subsets using the smoothed normalized ADTs. First, we gated γδ T cells using expression of Vδ2 TCR. Then, we separated MAIT cells using expression of CD161 and TCR Vα 7.2. We then used CD4 and CD8 to separate CD4 (CD4$^+$CD8$^-$), CD8 (CD4$^-$CD8$^+$), double-positive (CD4$^+$CD8$^-$) and double-negative (CD4$^-$CD8$^-$) T cells. We then subset to CD4$^+$ T cells and gated T$_{reg}$ cells using expression of CD25 and CD39. Of the remaining CD4$^+$ T cells, we used CD62L and CD45RA to define CD4-naive (CD62L$^+$CD45RA$^+$), CD4 central memory (CD62L$^+$CD45RA$^-$), CD4 effector memory (CD62L$^-$CD45RA$^-$) and CD4 TEMRA (CD62L$^-$CD45RA$^+$) populations. For the CD8$^+$ T cells, we similarly used CD62L and CD45RA to define CD8 naive (CD62L$^+$CD45RA$^+$), CD8 central memory (CD62L$^+$CD45RA$^-$), CD8 effector memory (CD62L$^-$CD45RA$^-$) and CD8 TEMRA (CD62L$^-$CD45RA$^+$) populations.

### T cell subset classification benchmarking analyses

We used T cell subsets defined by manual gating of ADTs in the flu-vaccine dataset as ground truth for prediction. For single cGEP

prediction, we ran TCAT to predict cGEP usage, and identified the cGEP that best predicted the lineage based on AUC.

We also used all the cGEPs to perform simultaneous multi-label prediction. We scaled the normalized usages for all cGEPs to zero mean and unit variance. Using COMBAT as a training dataset, we trained a multinomial logistic regression using scikit-learn[60] version 1.0.2 with lbfgs solver to predict gated subset from usages. Model weights were adjusted by the inverse of subset size using class_weight = 'balanced', allowing subsets with different cell counts to contribute to the model equally. We excluded CD4 TEMRA, double-negative and double-positive subsets from this analysis due to low cell counts in both the training and testing datasets. We evaluated this model in the independent flu-vaccine query dataset.

Analogous comparisons were made using GEPs from Yasumizu et al. fit to the data using the NMFproject software[11]. We also obtained gene sets derived from NMF analyses of T cells in a pan-cancer dataset[14]. To assess the ability of these gene sets to predict gated subsets, we used the score_genes function in Scanpy[61] on data normalized following the standard pipeline (library size normalizing to TP10K, log transformation, scaling each gene to unit variance). We then assigned each subset to the gene set that yielded the maximal AUC.

To evaluate clustering, we first normalized the data as above, and data were subset to highly variable genes using the highly_variable_genes function in Scanpy with default parameters. We then ran PCA and Harmony batch correction of the principal components[20]. We then computed the $K$-nearest neighbor graph using 31 harmony-corrected principal components and 30 nearest neighbors. We then performed Leiden clustering[68] with resolution parameters ranging from 0.25 to 2.25 increasing by 0.25. For each clustering resolution, we performed a greedy search to assign clusters to manually gated subsets based on maximization of the balanced accuracy (that is, the average recall across all subsets). In each iteration, we considered all unassigned clusters and possible gated subset assignments and selected the cluster and assignment that most increased the overall balanced accuracy. When no remaining cluster assignments would increase the balanced accuracy, we assigned the cluster to a subset that least decreased the balanced accuracy. We continued this process until each cluster was assigned to a subset.

To evaluate other reference mapping methods, we followed the normalization methods directed by each method. We supplied the flu-vaccine raw counts matrix to Azimuth and utilized its human PBMC reference (the HIV-vaccine dataset). For reference mapping with Symphony, we built a reference for the HIV-vaccine dataset, performing library size normalization (TP10K normalization) followed by log transformation. We selected the top 2,000 variable genes per donor using VST selection, centered and scaled the normalized counts, and performed PCA (irlba package) and batch correction (on lane and donor) using Harmony. We built a Symphony reference using the batch-corrected PCs. We then performed TP10K library size normalization and log transformation on the flu-vaccine query and annotated cells using Symphony's reference mapping and annotation transfer algorithms. For reference mapping with ProjecTILs, we supplied the flu-vaccine raw count matrix and utilized the default comprehensive T cell reference provided by ProjecTILs (ref_TILAtlas_mouse_v1). Internally, ProjecTILs maps human queries to its mouse reference using gene orthologs.

For all accuracy calculations, we utilized sklearn's balanced_accuracy_score, an approach appropriate for cases of class imbalance. In the binary case, balanced accuracy refers to the mean sensitivity and specificity of a prediction. In the multi-class case, balanced accuracy refers to the mean sensitivity across classes.

### AIM-seq

Patients were recruited for this study through the Partners Biobank[69]. Informed consent was obtained from all participants. We have complied

with all ethical regulations, and the study protocol was approved by the Mass General Brigham Institutional Review Board. PBMCs were collected from five genotyped participants with no autoimmune diseases or use of immunomodulatory medications. PBMCs were quickly thawed and placed in prewarmed xVIVO15 cell culture medium (Lonza) supplemented with 5% heat-inactivated FBS. To reduce cell clumping, PBMCs were incubated in xVIVO15 containing 50 U ml$^{-1}$ of benzonase nuclease (Sigma-Aldrich) for 15 min at 37 °C and filtered using a 70-µm cell strainer. Washed and nuclease-treated cells were seeded in a 96-well cell culture plate at a concentration of $2.5 \times 10^6$ per ml. Peptide stimulations were performed using the CEFX Ultra SuperStim Pool (JPT Peptide Technologies, PM-CEFX-1) at a final concentration of 1.25 µg ml$^{-1}$ per peptide for 22 h at 37 °C and 5% $CO_2$. Recombinant anti-CD28 and anti-CD49d antibodies (BioLegend) were added at a final concentration of 5 µg ml$^{-1}$ and 0.625 µg ml$^{-1}$, respectively, to provide co-stimulation for peptide-reactive T cells. Separately mock-stimulated cells were treated with anti-CD28 and anti-CD49d antibodies at the same concentration.

Peptide responsive T cells were detected by the expression of the surface activation markers PD-L1, OX40 and CD137 via flow cytometry. Following the stimulation, peptide-treated and mock-stimulated cells were washed in cell staining buffer (PBS + 2 mM EDTA + 2% FBS) to end the stimulation. Fc receptor blocking was performed using a 1:50 dilution of Human TruStain FcX (BioLegend) in cell staining buffer for 10 min at 4 °C. Cell viability staining was performed using a 1:500 dilution of Zombie Yellow Fixable Viability Dye (BioLegend) prepared in PBS for 30 min at 4 °C. Surface staining was performed using 1:100 dilutions of BV421-conjugated anti-CD3, FITC-conjugated anti-CD4, BV786-conjugated anti-PD-L1, PE-conjugated anti-OX40 and APC-conjugated anti-CD137 (BioLegend) for 25 min at 4 °C in cell staining buffer. Following cell staining, antigen reactive and non-reactive T cells were identified using a BD FACSAria II cell sorter and collected in cRPMI medium (100 U ml$^{-1}$ penicillin–streptomycin + 2 mM L-glutamine + 10 mM HEPES + 0.1 mM non-essential amino acids + 1 mM sodium pyruvate + 0.05 mM 2-mercaptoethanol) supplemented with 20% FBS. Sorted T cell populations were then labeled with 75 µl of TotalSeq oligonucleotide-conjugated hashing antibody mix, incubated for 30 min at 4 °C with gentle mixing after 15 min, and pooled in equal quantities. Staining with the TotalSeq-C Human Universal Cocktail (BioLegend) was then performed according to the manufacturer's instructions. The cells were then resuspended in PBS supplemented with 0.04% FBS at a final concentration of 500 cells per µl and submitted for single-cell profiling on the Chromium Next GEM instrument. Library preparation was completed for the hashtag oligonucleotides, scRNA-seq, CITE-seq and TCR-repertoire sequencing following the manufacturer's instructions.

We collected AIM-seq data from two separate 10x runs. In the first experiment, PBMCs from three donors were processed independently as described above and were pooled after fluorescence-activated cell sorting. In the second run, PBMCs from four donors, two of which overlapped with the first run, were stimulated separately and pooled before fluorescence-activated cell sorting.

### Preprocessing of AIM-seq data

The AIM-seq dataset was processed using Cell Ranger version 6.1.1 with default parameters and alignment to hg38 reference genome. The donor of origin for each cell was determined using Demuxlet version 1.0 with a doublet-prior of 0.1 (ref. 70). Cells with a null or ambiguous Demuxlet result, fewer than 10 counts of the hashtag oligonucleotides, or fewer than 50 total RNA counts were filtered. To account for staining differences between the hashtag oligonucleotides and different sequencing depths of the two 10x runs, the counts for each hashtag oligonucleotide in each 10x run were scaled to have the same median value. Next we added a pseudocount to the hashtag oligonucleotide counts and $\log_{10}$ transformed these data. Then we ran Gaussian mixture models separately for each hashtag oligonucleotide with $K = 2$ clusters.

Each cell was assigned to a single condition if it was in the high cluster for one oligonucleotide and the low clusters for all others, a doublet if it was in the high cluster for more than one oligonucleotide, or an empty droplet if it was in the low cluster for all oligonucleotides. Empty droplets or doublets based on the hashtag oligonucleotide clustering were filtered, as were doublets based on Demuxlet. Genes detected in fewer than ten cells were filtered before running TCAT.

### Statistics and reproducibility

We did not perform a statistical analysis for choosing sample size. We chose to replicate the study across five samples for reproducibility, as we sequenced thousands of cells per donor and were well powered to find significant associations between stimulation conditions. We excluded cells based on unique molecular identifier counts and ability to be demultiplexed, as above. No blinding was performed.

### cGEP associations with AIM positivity, proliferation and disease

To associate cGEPs with the AIM-seq stimulus, we first ran TCAT to fit the usages of the cGEPs in the AIM-seq dataset. We then computed the average usage of each cGEP in cells from each sort condition in each donor. We created two dummy variables, the first indicating whether a sample was treated with CEFX or mock, and the second indicating whether a sample was both CEFX-treated and AIM-positive or not. We used ordinary least-squares regression to estimate the effects of these two variables and an intercept. cGEPs associated with the CEFX or mock dummy variable were labeled 'milieu-associated', while cGEPs positively associated with the AIM-positive dummy were labeled 'AIM-associated'.

To associate cGEPs with proliferation, we defined cells as proliferating or non-proliferating in each dataset by setting a threshold of 0.1 on the sum of the three cell cycle cGEPs—CellCycle-S-phase, CellCycle-Late-S and CellCycle-G2M. We then computed the mean usage of each cGEP per sample separately in high cell-cycle (sum usage > 0.1) and low cell-cycle (sum usage < 0.1) cells. We filtered samples that did not have at least 10 high cell-cycle cells and 100 low cell-cycle cells. Then, for each cGEP, we performed a two-tailed paired $t$-test (ttest_rel in Scipy, default parameters) between average cGEP usage for high and low cell-cycle cells. We meta-analyzed $P$ values across datasets using Fisher's Method (combine_pvalues in Scipy).

To associate cGEPs with sample-level disease phenotypes, we calculated the average usage of each cGEP in each sample for a given dataset. We then used ordinary least-squares regression to find cGEPs with higher average usage in disease samples than controls, controlling for sample-level batch variables as covariates. For all datasets, disease status was modeled as a binary dummy variable, and an intercept was included. For UK-COVID, the processing site was included as a dummy variable covariate. For COMBAT, sequencing pool and processing institute were included as dummy variable covariates. For the pan-cancer dataset, all cancer types were initially included in the analysis and dummy variable covariates were included for tissue of origin. In addition, sequencing technology was included as a dummy variable. When there were multiple tumor samples or matched normal samples from the same donor, we excluded the duplicates with fewer total cells. For all association tests, we performed FDR correction of the $P$ values using the Benjamini–Hochberg method (fdrcorrection in Statsmodels with method = 'indep').

For the ICI response analyses, we used one-tailed or two-tailed Welch's $t$-tests as indicated in the main text for analyses of isolated pretreatment or post-treatment samples. For analyses of combined pretreatment and post-treatment samples, we used mixed linear models with an intercept, treatment status and clinical response as fixed effects, patient of origin as random intercepts, and average cGEP usage as the response variable. We used the MixedLM package in statsmodels with REML = False (that is, using maximum likelihood estimation) and computed $P$ values for the response fixed effect using

the likelihood-ratio test. For meta-analyses, we used Fisher's method to combine one-tailed *P* values.

## Defining the ASA score

We used $CD71^+CD95^+$ surface protein coexpression in the COMBAT and flu-vaccine datasets as an in vivo correlate of TCR activation to help prioritize AIM-associated cGEPs for predicting TCR-activated cells. First, we preprocessed the ADT surface proteins in these datasets as described in the manual subset gating section. We then subsetted cells by their manual gating-defined broad cell types (conventional CD4, $CD4^+ T_{reg}$, conventional CD8, other) and gated $CD71^+CD95^+$ cells separately for each cell type as the response feature to be predicted by AIM-associated cGEPs.

We then performed forward stepwise selection, evaluating how well the summation of usages of different combinations of AIM-associated cGEPs would predict $CD71^+CD95^+$ gating. At each stage, the per-cell ASA score was computed as the sum of normalized usages of cGEPs in the predictive set. At each forward step, we determined which cGEP should be added to the predictive set based on which would most improve the average AUC across the flu-vaccine and COMBAT datasets. We used a reduction in AUC in both datasets as the stopping criterion for adding cGEPs. We considered all AIM-associated cGEPs identified as candidates for this but excluded those known to have a broader function outside T cell activation (for example, cytoskeleton, metal-lothionein and cell cycle) and those reflecting activation-associated T cell subsets ($T_{PH}$ and $T_H17$-activated). We also excluded exhaustion from the ASA score as it reflects a distinct inhibitory response to antigen stimulation that users may wish to annotate separately.

## Benchmarking the ASA score

We benchmarked ASA's prediction of activation with published T cell activation gene sets, where ground-truth activation is defined by AIM positivity in the AIM-seq dataset and $CD71^+CD95^+$ coexpression in the flu-vaccine dataset. We utilized all T cell activation gene sets present in the immunologic signature (C7) collection on MSigDB. We then scored each cell's usage of each of the gene sets using scanpy's score_genes function following data preprocessing (TP10K normalization, log transformation, mean and variance scaling).

## Reporting summary

Further information on research design is available in the Nature Portfolio Reporting Summary linked to this article.

# Online content

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

## Methods

### Materials and reagents

| Reagent or resource | Source | Identifier |
|---|---|---|
| XVIVO15 culture media | Lonza | Catalog no. 02-060Q |
| RPMI 1640 medium | Thermo Fisher | Catalog no. 11875093 |
| Benzonase nuclease | Sigma-Aldrich | CAS no. 9025-65-4 |
| Trial grade CEFX peptide pool | JPT | PM-CEFX-2 |
| Anti-CD28 antibody | BioLegend | Catalog no. 302933 RRID: AB_11150591 |
| Anti-CD49d antibody | BioLegend | Catalog no. 304339 RRID: AB_2810443 |
| Human TruStain FcX (Fc receptor blocking solution) | BioLegend | Catalog no. 422302 RRID: AB_2818986 |
| Zombie yellow fixable viability kit | BioLegend | Catalog no. 423104 |
| TotalSeq-C human universal cocktail, V1.0 | BioLegend | Catalog no. 399905 |
| Human TOTAL-SeqC repertoire (5') hashing antibodies | BioLegend | Catalog nos. 394661, 394663 and 394665 |
| Anti-CD3-BV421 (SK7) | BioLegend | Catalog no. 344833 RRID: AB_2565674 |
| Anti-CD134-PE (Ber-ACT35) | BioLegend | Catalog no. 350003 RRID: AB_10641708 |
| Anti-CD274-BV785 (29E.2A3) | BioLegend | Catalog no. 329735 RRID: AB_2629581 |
| Anti-CD137-APC (4-B4-1) | BioLegend | Catalog no. 309809 RRID: AB_830671 |
| Anti-CD4-FITC (RPA-T4) | BioLegend | Catalog no. 300505 RRID: AB_314073 |
| Chromium next GEM single-cell 5' kit v2, 16 reactions | 10x | Catalog no. 1000263 |
| Dual index kit TN set A, 96 reactions | 10x | Catalog no. 1000250 |
| Chromium next GEM chip K single-cell kit, 48 reactions | 10x | Catalog no. 1000286 |
| Chromium single-cell human TCR amplification kit, 16 reactions | 10x | Catalog no. 1000252 |
| Library construction kit, 16 reactions | 10x | Catalog no. 1000190 |
| 5' feature barcode kit, 16 reactions | 10x | Catalog no. 1000256 |

## Data availability

The data used in this study for training and validating TCAT are publicly available, and can be downloaded from the following sources: https://doi.org/10.7303/syn52297840 (AMP-RA), https://zenodo.org/records/5461803 (pan-cancer), Gene Expression Omnibus (GEO): GSE164378 (HIV-Vaccine), https://www.ebi.ac.uk/biostudies/arrayexpress/studies/E-MTAB-10026 (UK-COVID), https://zenodo.org/records/6120249 (COMBAT), https://www.tissueimmunecellatlas.org/ (Pan-Tissue) and the GEO: GSE158769 (TBRU) and GSE206265 (flu-vaccine). The count matrices and metadata for the AIM-seq data produced in this study are located on Zenodo (https://zenodo.org/records/15271929)[71] and on the GEO (GSE297814). Sequencing data produced in our AIM-seq study are located on dbGaP (phs004043). Source data are provided with this paper.

## Code availability

The code for starCAT is available at https://github.com/immunogenomics/starCAT/. starCAT can also be run on our website (https://immunogenomics.io/starcat/). The analysis scripts used in this paper are available at https://github.com/immunogenomics/TCAT_analysis/.

## References

71. Kotliar, D., Curtis, M. & Raychaudhuri, S. Activation-induced marker (AIM) sequencing of healthy human T cells. Zenodo https://doi.org/10.5281/zenodo.15271929 (2025).

## Acknowledgements

This work was supported by National Institutes of Health grants P01AI148102, P30AR070253, UC2AR081023, U01HG012009, R01AR063759 and R56HG013083. P.C.S. is supported by the Howard Hughes Medical Institute. K.W. is supported by NHGRI T32HG002295 and NIAMS T32AR007530.

## Author contributions

Conceptualization: D.K., M.C. and S.R. Software: D.K. and M.C. Formal analysis: D.K. and M.C.; Methodology: D.K., M.C., R.A. and S.R. Writing: D.K., M.C., K.W., A.N., Y.B., K.S., Y.Z., P.C.S., D.A.R. and S.R. Funding acquisition: S.R.

## Competing interests

S.R. is a founder for Mestag, on advisory boards for Pfizer, Janssen and Sonoma, and a consultant for AbbVie, Biogen, Nimbus and Magnet. D.A.R. is a coinventor on a patent using $T_{PH}$ cells as a biomarker in autoimmune diseases. P.C.S. is a cofounder of, shareholder in, and consultant to Sherlock Biosciences, and Delve Bio, as well as a Board member of and shareholder in Danaher Corporation. The other authors declare no competing interests.

## Additional information

**Extended data** is available for this paper at https://doi.org/10.1038/s41592-025-02793-1.

**Correspondence and requests for materials** should be addressed to Soumya Raychaudhuri.

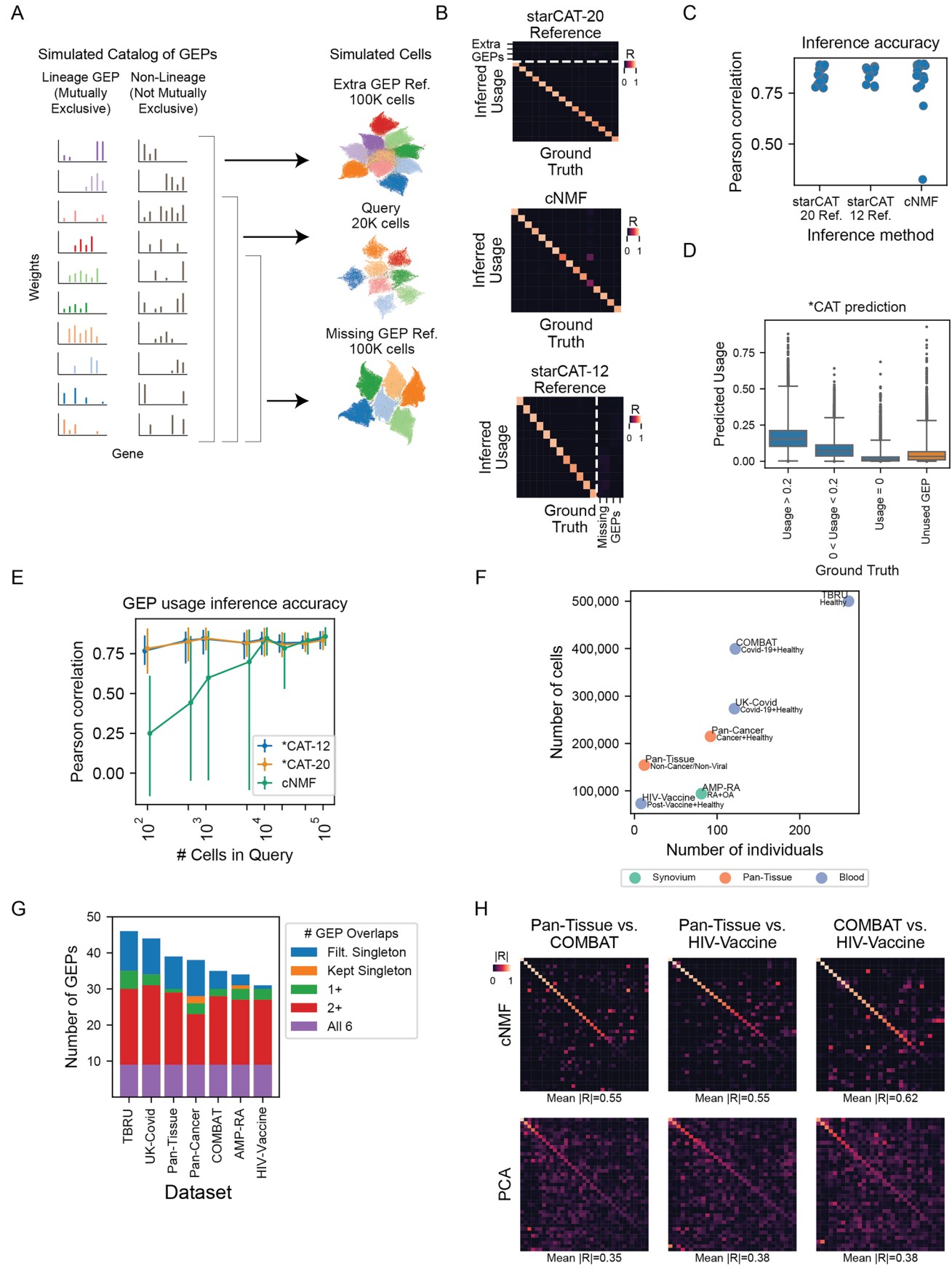

**Extended Data Fig. 1 | See next page for caption.**

**Extended Data Fig. 1 | Characterizing starCAT. a**, Schematic of simulation with toy illustrations of gene expression programs (GEPs) and resulting Uniform Manifold Approximation and Projections (UMAPs). Cells are colored by subset GEPs. **b**, Pearson correlation of ground truth simulated usages of each GEP (columns) vs inferred usages (rows) for starCAT with the 20 GEP reference (top), starCAT with the 12 GEP reference (bottom) or cNMF of the query with 16 inferred components (middle). **c**, Pearson correlation between inferred gene expression programs and the corresponding ground truth usages, extracted from **b**. **d**, starCAT predicted GEP usage for cells with ground-truth usage > 0.2, 0-0.2, or 0 (blue), and GEPs present in the reference but absent in the query (orange). Boxes represent interquartile range and whiskers represent 1.5 x interquartile range. The box center line indicates the median. Sample sizes for each category are n = 46,535 (Usage > 0.2), n = 46,867 (0 < Usage < 0.2), n = 226,598 (Usage = 0), and n = 80,000 (Unused GEP). **e**, Pearson correlation between ground truth and GEP usages inferred by starCAT and cNMF for different query dataset sizes. Simulation parameters described in Methods. Marker represents mean, error bars represent range. **f**, Summary of reference datasets including number of individual donors (x-axis), number of cells (y-axis), and tissue source (dot color). Phenotypes are listed below the dataset names. **g**, Number of GEPs identified per dataset, colored by whether they clustered with one or more other dataset GEPs (purple, red, or green), did not cluster with a GEP from another dataset but were kept as dataset-specific (orange), or did not cluster with a GEP from another dataset and were filtered (blue). **h**, Absolute value of Pearson correlation of spectra learned by cNMF (top) or PCA (bottom) between different pairs of datasets. PCs were computed from the same batch-corrected expression matrices used for cNMF. Mean |R| refers to the average along the matrix diagonal, which corresponds to pairs of components with highest correlation across the dataset pair.

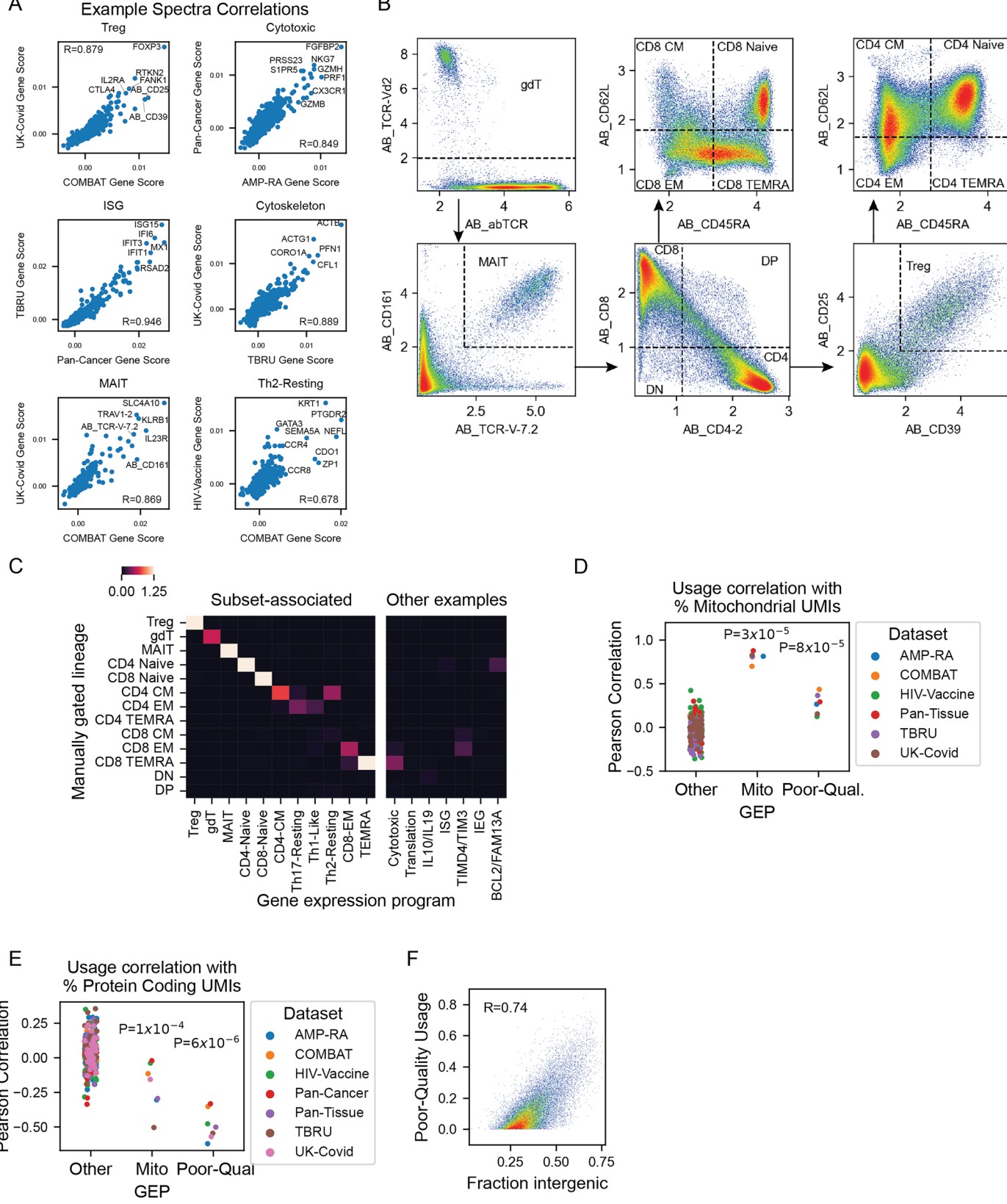

**Extended Data Fig. 2 | Annotating cGEPs. a**, Scatter plots of selected correlated GEP pairs with source dataset indicated in the axis labels (P < 1×10⁻¹⁰⁰ for all correlations). **b**, Manual gating of COMBAT dataset using smoothed surface protein antibody-derived tag (ADTs, Methods). **c**, Multivariate logistic regression coefficients for cGEPs (columns) predicting manually gated populations (rows). For visualization, the minimum and maximum values are thresholded to 0 and 1.25. Seven example non-subset cGEPs are shown on the right. **d**, Pearson correlation of cGEPs with percentage of mitochondrial transcript per cell, for each dataset. All cGEPs excluding Mito and Poor-Quality are included in "Other". P-values are from a two-tailed Ranksum test of the selected cGEP against the Other cGEPs. **e**, Same as **d** but showing correlation with the percentage of UMIs assigned to protein coding genes. **f**, Scatter plot of the proportion of UMIs mapping to intergenic regions against Poor-Quality cGEP usage for cells in the AMP-RA dataset.

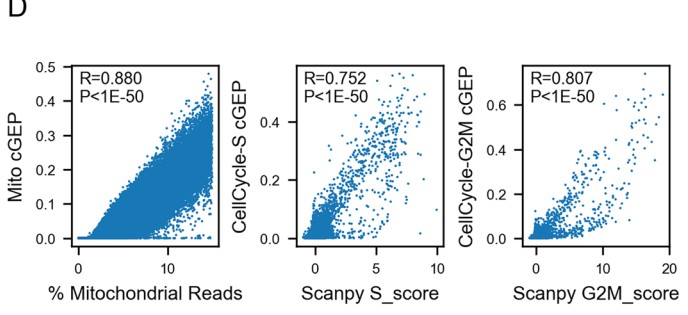

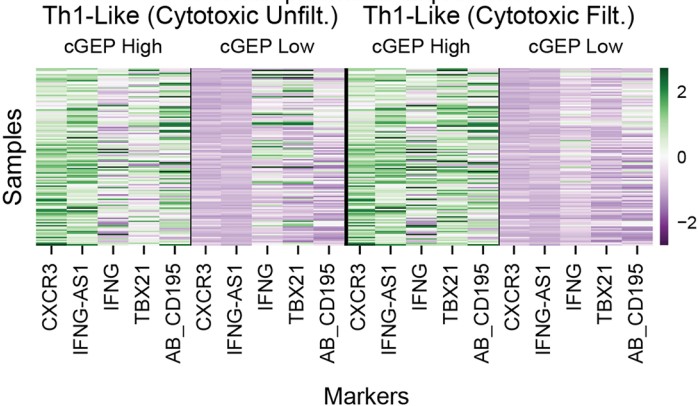

**Extended Data Fig. 3 | Benchmarking starCAT's subset annotation. a**, Manual gating for the Flu-Vaccine dataset analogous to Extended Data Fig. 2b. **b**, Proportion of cells within each manually gated lineage (columns) assigned to each lineage annotation (rows). **c**, Area under the curve (AUC) for prediction of manually gated subset based on TCAT multilabel prediction, a single most associated TCAT cGEP, analogous predictions using the single most associated NMF component published in Yasumizu et al., 2024[11], or using gene sets from NMF components in Gavish et al., 2023[14]. **d**, Usage of the mitochondria cGEP against the percentage of mitochondrial reads per cell (left). Usage of the CellCycle-S (middle) and CellCycle-G2M (right) cGEPs against the S and G2M scores output by Scanpy's score_genes_cell_cycle function with published proliferation gene sets[27]. **e**, Expression of selected Th1 marker genes in pseudobulk profiles of Th1-Like-high and low cells, per sample (rows). Cytotoxic-high cells are included (left) and filtered (right). Sample expression is normalized by library size and z-scored across rows, separately for the two filtering conditions.

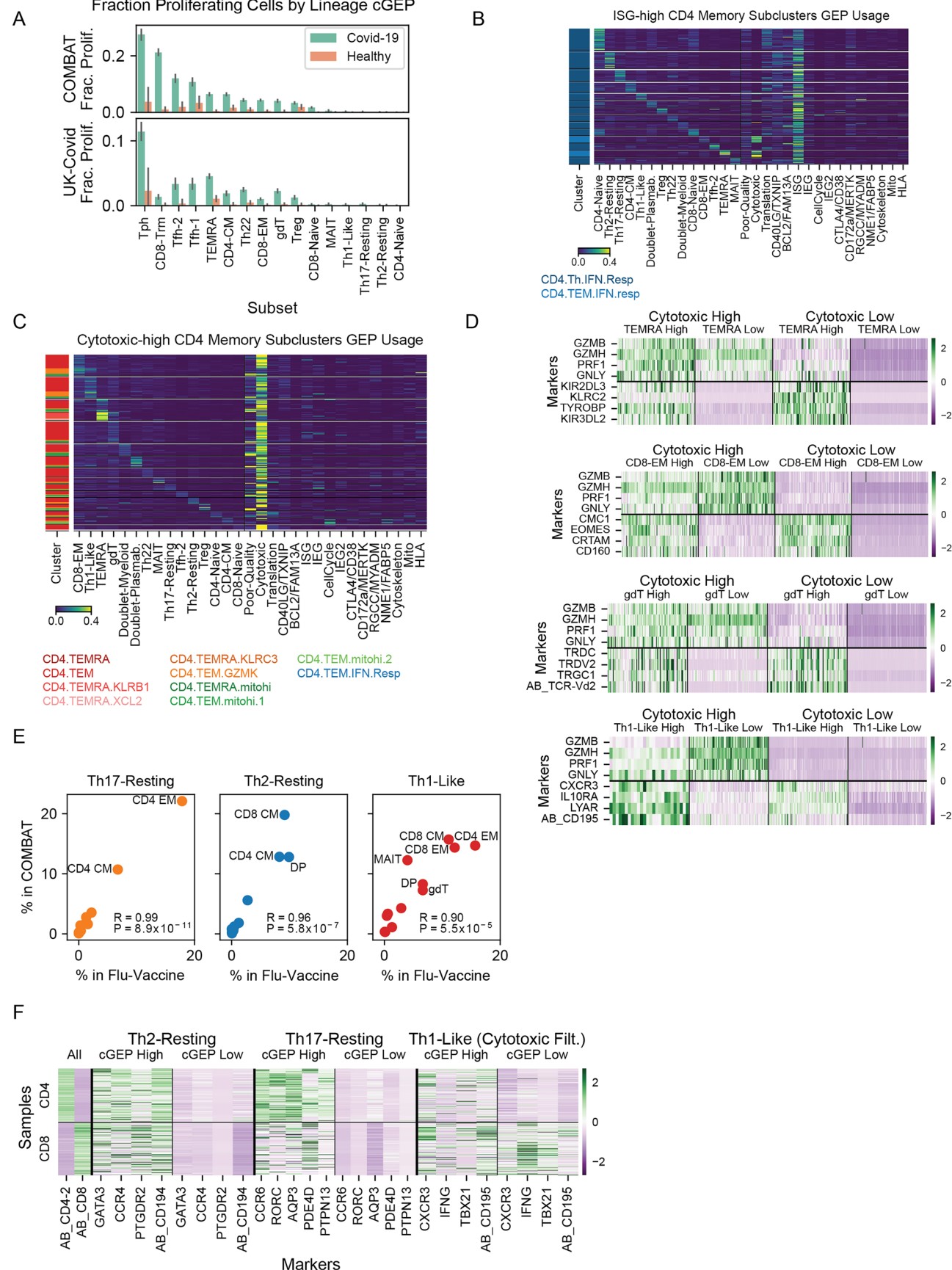

**Extended Data Fig. 4 | See next page for caption.**

**Extended Data Fig. 4 | Comparing TCAT with COMBAT dataset clustering.**
**a**, Fraction of proliferating cells (cell cycle usage>0.1) assigned to each subset based on the most highly used subset-associated GEPs, for cells from Covid-19 or healthy donors in the two Covid-19 datasets. Bar represents the proportion of cells and error bars represent 95% bootstrap confidence intervals around this proportion. **b**, Usage of selected cGEPs (columns) in cells (rows) grouped by maximum subset cGEP. Cells are drawn from subclusters with high usage of the ISG cGEP, indicated in the colorbar. **c**, Same as **b**, but only showing cells

from subclusters with high cytotoxicity cGEP usage. **d**, Heatmap of pseudobulk expression of marker genes in cytotoxic-high and low cells and subset cGEP high and low cells, per sample. Expression is normalized by library size and z-scored across rows. **e**, Average fraction of polarized cells (usage>0.1) per gated subset, across samples, within COMBAT and Flu-Vaccine datasets. **f**, Pseudobulk expression profiles of selected marker genes in polarization-high and low cells, separately for gated CD4 and CD8s T cells, per sample. Sample expression is normalized by library size and z-scored across rows, for each polarization.

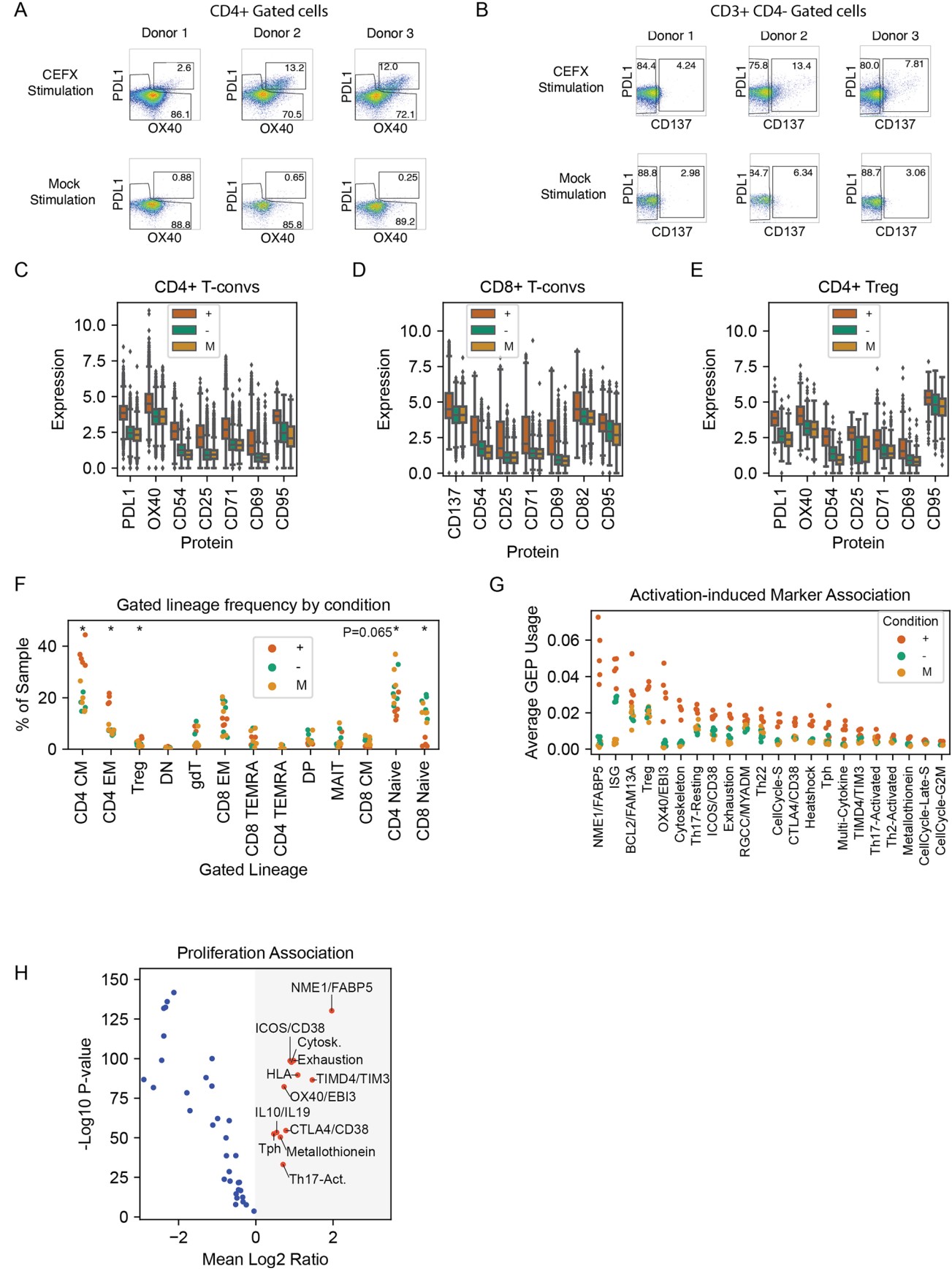

**Extended Data Fig. 5 | See next page for caption.**

**Extended Data Fig. 5 | Identifying activation associated cGEPs with AIM-Seq.**
**a**, **b**, Flow cytometry data of CD3 + CD4+ and CD3 + CD4- gated populations
for 3 donor samples for CEFX and mock conditions. **c**−**e**, Activation-induced
marker (AIM) surface protein expression based on CITE-Seq for CD4 + , CD8 + ,
and Treg subsets, stratified by sort condition. Boxes represent interquartile
range and whiskers represent 1.5x interquartile range. The box center line
indicates the median. Sample sizes are **c**: 23,532 cells, **d**: 13,284 cells, and **e**: 932
cells. **f**, Percentage of each sample assigned to each subset based on manual
gating, colored by stimulation condition. * indicates t-test P < .05 comparing +
and U. Exact P-values are CD4 CM: 3.17x10-3, CD4 EM: 3.48x10$^{-2}$, Treg: 3.93x10$^{-2}$,
CD8_CM: 6.49x10-2, CD4_Naive: 4.47x10$^{-2}$, CD8_Naive: 8.60x10$^{-4}$. **g**, Average cGEP
usage in each donor and condition, for AIM-associated cGEPs. **h**, Paired t-test of
pseudobulk cGEP usage in high and low cell cycle usage cells (threshold 0.1) from
each sample. X-axis shows the mean Log$_2$ ratio of average usages across datasets.
Y-axis shows the -Log$_{10}$ P-value. Statistically significant and positively associated
cGEPs are indicated in red.

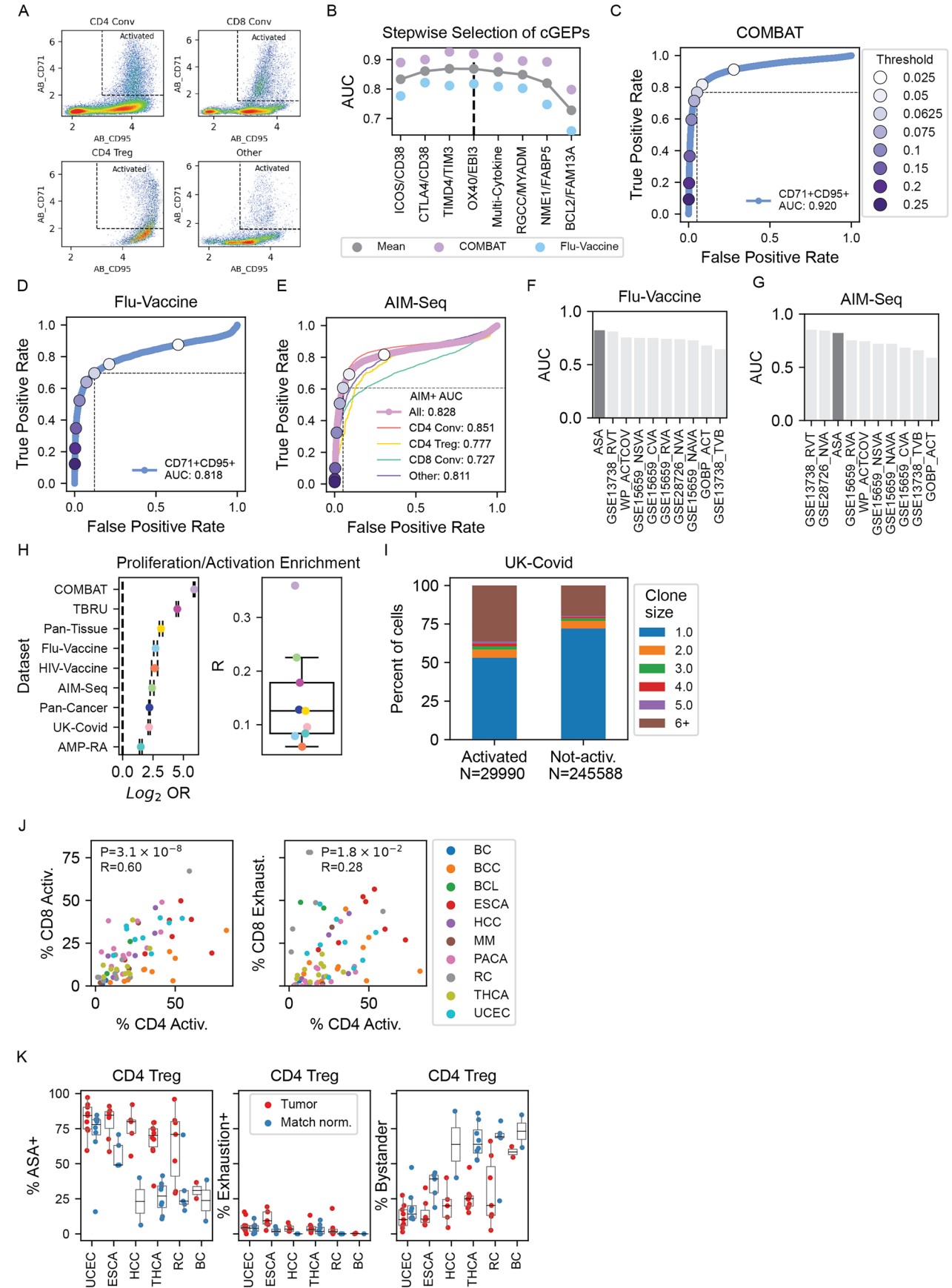

**Extended Data Fig. 6 | See next page for caption.**

**Extended Data Fig. 6 | Annotating antigen-specific activation *in vivo*.**
**a**, Definition of activation used for training the antigen-specific activation (ASA) score in the COMBAT dataset for manually gated subsets. **b**, AUC estimates for CD71/CD95 co-expression prediction based on summation of cGEPs sequentially added to the score from left to right. **c**, **d**, Receiver operator curve (ROC) for ASA prediction of CD71/CD95-based activation labels, with various thresholds denoted as colored points. **e**, ROC for ASA prediction of AIM-positivity in the AIM-Seq dataset. **f**, AUCs for prediction of CD71 + CD95+ co-expression by ASA as compared to cell scoring with alternative T cell gene sets in the Flu-Vaccine dataset. Full gene set names are GOBP_ACTIVATED_T_CELL_PROLIFERATION (GOBP_ACT), GSE13738_RESTING_VS_TCR_ACTIVATED_CD4_TCELL_DN (GSE13738_RVT), GSE13738_TCR_VS_BYSTANDER_ACTIVATED_CD4_TCELL_UP (GSE13738_TVB), GSE15659_CD45RA_NEG_CD4_TCELL_VS_ACTIVATED_TREG_DN (GSE15659_CVA), GSE15659_NAIVE_CD4_TCELL_VS_ACTIVATED_TREG_DN (GSE15659_NAVA), GSE15659_NONSUPPRESSIVE_TCELL_VS_ACTIVATED_TREG_DN (GSE15659_NSVA), GSE15659_RESTING_VS_ACTIVATED_TREG_DN (GSE15659_RVA), GSE28726_NAIVE_VS_ACTIVATED_CD4_TCELL_DN (GSE28726_NVA), WP_TCELL_ACTIVATION_SARSCOV2 (WP_ACTCOV). **g**, Same as **f** but prediction of AIM-positivity in the AIM-Seq dataset. **h**, Left - Odds ratio of enrichment between proliferation (aggregate cell cycle cGEP usage>0.1) and activation (ASA > 0.065) for each dataset. Estimates reflect odds ratios and error bars denote 95%

confidence intervals around the estimate. Right - Pearson correlation between ASA and aggregate cell cycle cGEP usage with colors mapping to dataset. Box represents the interquartile range and whiskers represent 1.5 x interquartile range. The box center line indicates the median (n = 9 datasets). **i**, Clonality in manually gated conventional CD4 and CD8 T cells annotated as activated (ASA > 0.065) or not activated (ASA < 0.065). Clonality is defined as the number of cells in the same sample with an identical alpha and beta CDR3 amino acid sequence. **j**, Percentage of activated conventional CD4 T cells (ASA > 0.065) versus percentage of activated or exhausted (exhaustion usage>0.065) conventional CD8 T cells across tumor samples. **k**, Percentage of activated, exhausted, or bystander (ASA + exhaustion usage<0.065) Tregs in tumors and match normal samples. Boxes represent the interquartile range and whiskers represent 1.5 x interquartile range. The box center line indicates the median. Sample sizes are BC: n = 2 Tumor, n = 2 Normal; BCC: n = 11 Tumor; BCL: n = 2 Tumor; ESCA: n = 7 Tumor,n = 7 Normal; HCC: n = 5 Tumor, n = 5 Normal; LUNG: n = 2 Tumor, n = 4 normal; MM: n = 3 Tumor; RC: n = 10 Tumor, n = 11 Normal; THCA: n = 10 Tumor, n = 8 Normal; UCEC: n = 9 Tumor, n = 8 Normal. BC - breast cancer, BCC - basal cell carcinoma, BCL - b-cell lymphoma, ESCA - esophageal cancer, HCC - hepatocellular carcinoma, MM - multiple myeloma, PACA - pancreatic cancer, RC - renal carcinoma, THCA - thyroid carcinoma, UCEC - uterine carcinoma.

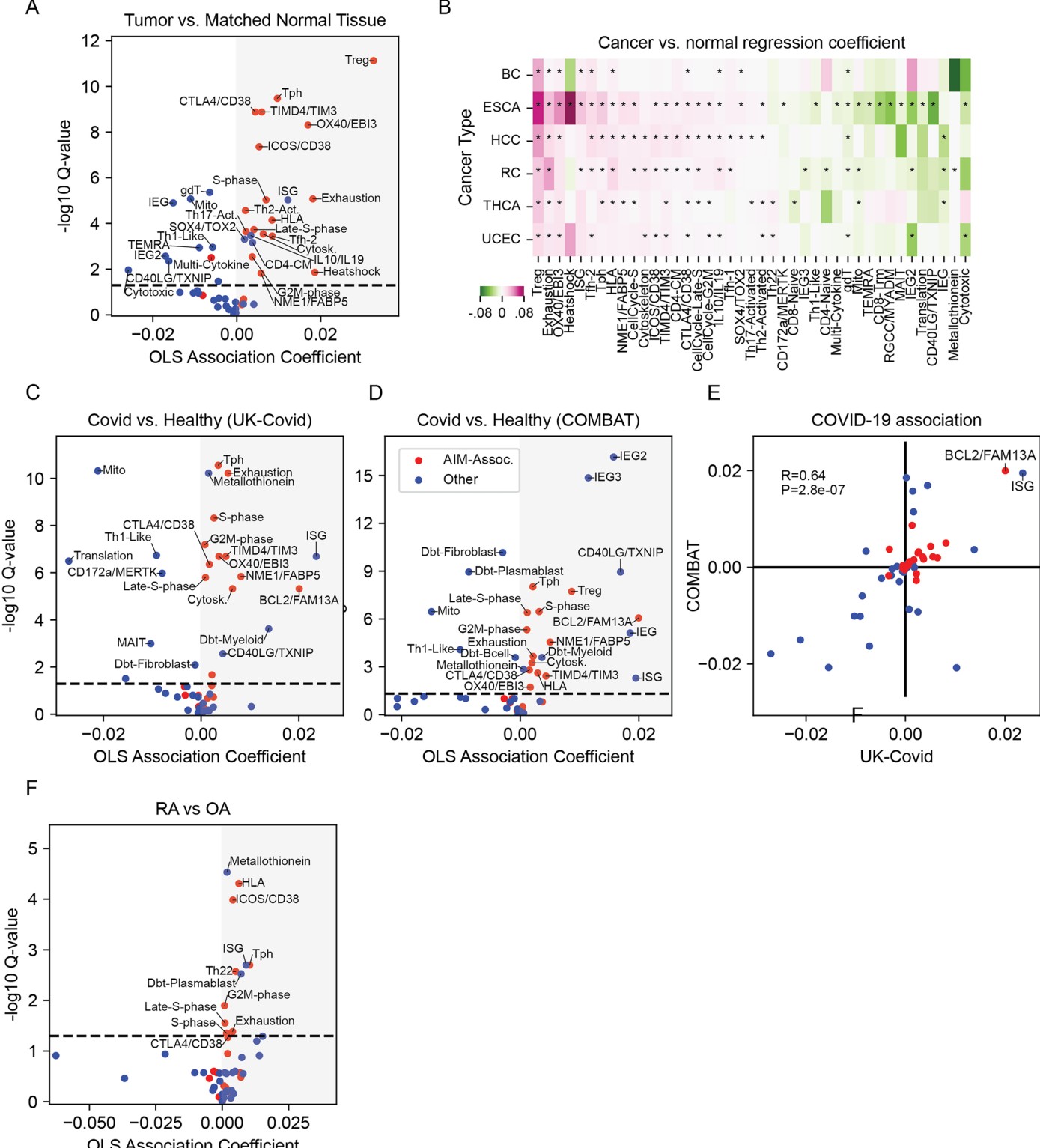

**Extended Data Fig. 7 | Identifying cGEPs associated with disease phenotypes.**
**a**, Associations of cGEP usage with tumor versus matched normal tissue. X-axis shows the regression coefficient. Y-axis shows the -Log10 FDR-corrected P-value (I.e. Q-value). **b**, Regression coefficients for tumor vs. normal samples for each tissue of origin. * denotes P < .05 for the corresponding coefficient. Cancer type abbreviations are: breast cancer (BC), esophageal cancer (ESCA), hepatocellular carcinoma (HCC), renal cell carcinoma (RC), thyroid carcinoma (THCA), and endometrial cancer (UCEC). **c, d**, Same as **a** but for association with Covid-19 status in the UK-Covid and COMBAT datasets. **e**, Scatter plot of regression coefficients from **c** and **d**. **f**, Same as **a** but comparing synovial T cells from patients with Rheumatoid Arthritis and Osteoarthritis.

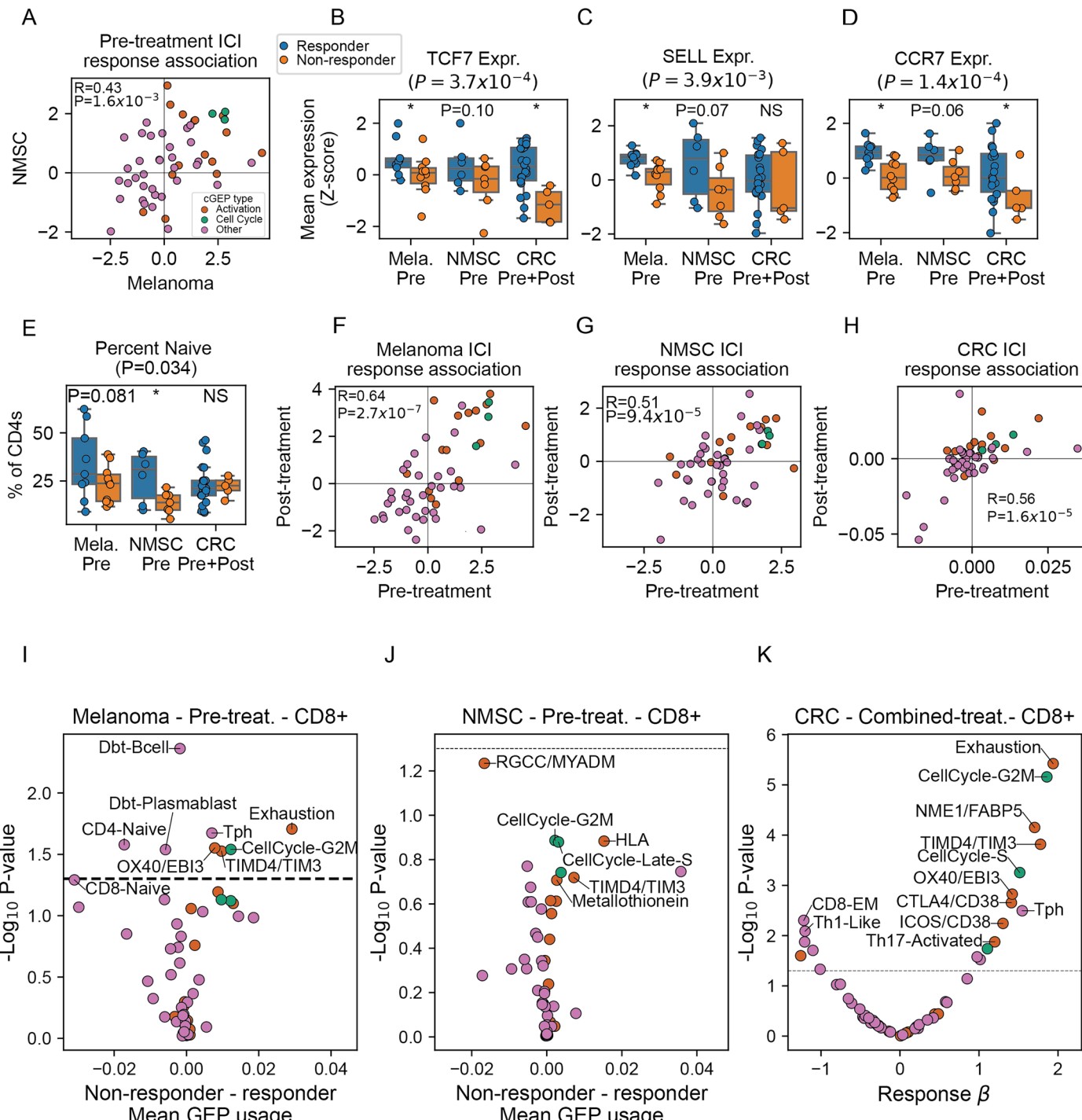

**Extended Data Fig. 8 | cGEPs associated with immune checkpoint inhibitor response. a**, T-statistics from pre-treatment melanoma and NMSC associations. R denotes Pearson correlation. Dots are colored by cGEP type. **b–d**, Average expression of naive T cell marker genes. Melanoma and NMSC P-values are from one-tailed T-tests and CRC P-values are from a mixed linear regression likelihood ratio tests. P-value in the title is from a meta-analysis across the three cancer types. Box represents the interquartile range and whiskers represent 1.5 x interquartile range. The box center line indicates the median. Sample sizes shown are n = 9, n = 6, n = 15 for responders in melanoma, NMSC, CRC respectively and n = 10, n = 7, n = 4 in non-responders in melanoma, NMSC, CRC **e**, Same as **b** but showing percentage of CD4 Naive T cells out of total CD4 T cells based on TCAT lineage classification. **f**, **g**, T-statistics computed in pre- and post- treatment CD4 T cells for melanoma and non-melanoma skin cancer (NMSC). **h**, Same as F but showing mean difference rather than T-statistic due to the presence of only one pre-treatment non-responder. **i**–**k**, Same as Fig. 6b–d but for CD8 T cells.

# Reporting Summary

## Statistics

For all statistical analyses, confirm that the following items are present in the figure legend, table legend, main text, or Methods section.

| n/a | Confirmed | |
|---|---|---|
| ☐ | ☒ | The exact sample size (*n*) for each experimental group/condition, given as a discrete number and unit of measurement |
| ☐ | ☒ | A statement on whether measurements were taken from distinct samples or whether the same sample was measured repeatedly |
| ☐ | ☒ | The statistical test(s) used AND whether they are one- or two-sided<br>*Only common tests should be described solely by name; describe more complex techniques in the Methods section.* |
| ☐ | ☒ | A description of all covariates tested |
| ☐ | ☒ | A description of any assumptions or corrections, such as tests of normality and adjustment for multiple comparisons |
| ☐ | ☒ | A full description of the statistical parameters including central tendency (e.g. means) or other basic estimates (e.g. regression coefficient) AND variation (e.g. standard deviation) or associated estimates of uncertainty (e.g. confidence intervals) |
| ☐ | ☒ | For null hypothesis testing, the test statistic (e.g. *F*, *t*, *r*) with confidence intervals, effect sizes, degrees of freedom and *P* value noted<br>*Give P values as exact values whenever suitable.* |
| ☐ | ☒ | For Bayesian analysis, information on the choice of priors and Markov chain Monte Carlo settings |
| ☐ | ☒ | For hierarchical and complex designs, identification of the appropriate level for tests and full reporting of outcomes |
| ☐ | ☒ | Estimates of effect sizes (e.g. Cohen's *d*, Pearson's *r*), indicating how they were calculated |

*Our web collection on statistics for biologists contains articles on many of the points above.*

## Software and code

Policy information about availability of computer code

| Data collection | BD FACSDiva 9.4 was used for FACS sorting cells prior to sequencing. |
|---|---|
| Data analysis | The code for star-CellAnnotator (starCAT) is available at https://github.com/immunogenomics/starCAT. The analysis scripts used in this paper are available at https://github.com/immunogenomics/TCAT_analysis.<br><br>Softwares used in these analysis scripts include:<br>cNMF 1.5.0<br>scsim2<br>anndata 0.10.9<br>anyio 4.3.0<br>argon2-cffi 23.1.0<br>argon2-cffi-bindings 21.2.0<br>array_api_compat 1.8<br>arrow 1.3.0<br>asttokens 2.4.1<br>async-lru 2.0.4<br>attrs 23.2.0<br>Babel 2.14.0<br>backports.tarfile 1.2.0<br>beautifulsoup4 4.12.3<br>bleach 6.1.0 |

```
blinker              1.8.2
Brotli               1.1.0
cached-property      1.5.2
certifi              2024.8.30
cffi                 1.16.0
charset-normalizer   3.3.2
click                8.1.7
cloudpickle          3.0.0
cnmf                 1.5.4
colorama             0.4.6
colorcet             3.1.0
comm                 0.2.2
contourpy            1.2.1
cryptography         43.0.0
cycler               0.12.1
dask                 2024.5.0
datashader           0.16.1
debugpy              1.8.1
decorator            5.1.1
defusedxml           0.7.1
Deprecated           1.2.14
docutils             0.21.2
entrypoints          0.4
et-xmlfile           1.1.0
exceptiongroup       1.2.2
executing            2.0.1
fastcluster          1.2.6
fastjsonschema       2.19.1
Flask                3.0.3
fonttools            4.53.1
fqdn                 1.5.1
fsspec               2024.3.1
future               1.0.0
get-annotations      0.1.2
graphtools           1.5.3
gseapy               1.1.3
h11                  0.14.0
h2                   4.1.0
h5py                 3.11.0
harmonypy            0.0.9
hpack                4.0.0
httpcore             1.0.5
httpx                0.27.0
hyperframe           6.0.1
idna                 3.8
igraph               0.11.5
importlib_metadata   7.1.0
importlib_resources  6.4.0
ipykernel            6.29.3
ipython              8.22.2
ipython-genutils     0.2.0
ipywidgets           8.1.2
isoduration          20.11.0
itsdangerous         2.2.0
jaraco.classes       3.4.0
jaraco.context       5.3.0
jaraco.functools     4.0.1
jedi                 0.19.1
jeepney              0.8.0
Jinja2               3.1.4
joblib               1.4.2
json5                0.9.25
jsonpointer          2.4
jsonschema           4.22.0
jsonschema-specifications 2023.12.1
jupyter              1.0.0
jupyter_client       7.4.9
jupyter-console      6.6.3
jupyter_core         5.7.2
jupyter-events       0.10.0
jupyter-lsp          2.2.5
jupyter_server       2.14.0
jupyter_server_terminals 0.5.3
jupyterlab           4.2.0
jupyterlab_pygments  0.3.0
jupyterlab_server    2.27.1
```

```
jupyterlab_widgets      3.0.10
keyring                 25.2.1
kiwisolver              1.4.5
legacy-api-wrap         1.4
leidenalg               0.10.2
llvmlite                0.43.0
locket                  1.0.0
magic-impute            3.0.0
markdown-it-py          3.0.0
MarkupSafe              2.1.5
matplotlib              3.8.4
matplotlib-inline       0.1.7
mdurl                   0.1.2
mistune                 3.0.2
more-itertools          10.3.0
mudata                  0.2.3
multipledispatch        0.6.0
munkres                 1.1.4
muon                    0.1.6
natsort                 8.4.0
nbclassic               1.0.0
nbclient                0.10.0
nbconvert               7.16.4
nbformat                5.10.4
nest_asyncio            1.6.0
networkx                3.3
nh3                     0.2.18
notebook                6.5.6
notebook_shim           0.2.4
numba                   0.60.0
numpy                   1.26.4
openpyxl                3.1.2
overrides               7.7.0
packaging               24.1
palettable              3.3.3
pandas                  2.2.2
pandocfilters           1.5.0
param                   2.1.0
parso                   0.8.4
partd                   1.4.2
patsy                   0.5.6
pexpect                 4.9.0
pickleshare             0.7.5
pillow                  10.4.0
pip                     24.0
pkginfo                 1.10.0
pkgutil_resolve_name    1.3.10
platformdirs            4.2.1
ply                     3.11
prometheus_client       0.20.0
prompt-toolkit          3.0.42
protobuf                4.25.3
psutil                  5.9.8
ptyprocess              0.7.0
pure-eval               0.2.2
pycparser               2.22
pyct                    0.5.0
Pygments                2.18.0
PyGSP                   0.5.1
pynndescent             0.5.12
pyparsing               3.1.2
PyQt5                   5.15.9
PyQt5-sip               12.12.2
PySocks                 1.7.1
python-dateutil         2.9.0.post0
python-json-logger      2.0.7
pytz                    2024.1
PyYAML                  6.0.2
pyzmq                   24.0.1
qtconsole               5.5.2
QtPy                    2.4.1
readme_renderer         44.0
referencing             0.35.1
requests                2.32.3
requests-toolbelt       1.0.0
rfc3339-validator       0.1.4
```

```
rfc3986              2.0.0
rfc3986-validator       0.1.1
rich              13.7.1
rpds-py              0.18.1
scanpy              1.10.1
scikit-learn           1.5.1
scikit-misc            0.1.4
scipy              1.14.1
scprep              1.2.3
seaborn              0.13.2
SecretStorage           3.3.3
Send2Trash            1.8.3
session_info           1.0.0
setuptools            69.5.1
sip               6.7.12
six               1.16.0
sniffio              1.3.1
soupsieve             2.5
stack-data            0.6.2
starcatpy             1.0.8
statsmodels            0.14.1
stdlib-list            0.10.0
tasklogger            1.2.0
terminado             0.18.1
texttable             1.7.0
threadpoolctl           3.5.0
tinycss2             1.3.0
toml               0.10.2
tomli              2.0.1
toolz              0.12.1
tornado              6.4
tqdm               4.66.5
traitlets             5.14.3
twine              5.1.1
types-python-dateutil    2.9.0.20240316
typing_extensions        4.11.0
typing-utils           0.1.0
tzdata              2024.1
umap-learn            0.5.5
unicodedata2           15.1.0
uri-template           1.3.0
urllib3              2.2.2
wcwidth              0.2.13
webcolors             1.13
webencodings           0.5.1
websocket-client         1.8.0
Werkzeug             3.0.3
wheel              0.43.0
widgetsnbextension       4.0.10
wrapt              1.16.0
xarray              2024.3.0
xlrd               1.2.0
zipp               3.17.0
```

For manuscripts utilizing custom algorithms or software that are central to the research but not yet described in published literature, software must be made available to editors and reviewers. We strongly encourage code deposition in a community repository (e.g. GitHub). See the Nature Portfolio guidelines for submitting code & software for further information.

# Data

Policy information about availability of data

All manuscripts must include a data availability statement. This statement should provide the following information, where applicable:

- Accession codes, unique identifiers, or web links for publicly available datasets
- A description of any restrictions on data availability
- For clinical datasets or third party data, please ensure that the statement adheres to our policy

The data used in this study for training and validating TCAT is publicly available, and can be downloaded from the following sources: https://doi.org/10.7303/syn52297840 (AMP-RA), https://zenodo.org/records/5461803 (Pan-Cancer), GEO: GSE164378 (HIV-Vaccine), https://www.ebi.ac.uk/biostudies/arrayexpress/studies/E-MTAB-10026 (UK-Covid), https://zenodo.org/records/6120249 (COMBAT), https://www.tissueimmunecellatlas.org/ (Pan-Tissue), GEO: GSE158769 (TBRU), GEO: GSE206265 (Flu-Vaccine). The count matrices for the Activation Induced Marker (AIM)-Seq data produced in this study are located on Zenodo (https://zenodo.org/records/15271929).

## Human research participants

Policy information about <u>studies involving human research participants and Sex and Gender in Research.</u>

| | |
|---|---|
| Reporting on sex and gender | All five participants were female. No sex-specific analyses were performed. |
| Population characteristics | Participants were healthy adults between the ages 40 and 50. We excluded individuals with autoimmune diseases or on immunomodulatory medications. |
| Recruitment | These individuals were recruited from the Partners Biobank. We excluded individuals with autoimmune diseases or on immunomodulatory medications. Recruitment occurred at clinics associated with MGB and may be biased towards more complex cases and individuals representative of the Greater Boston area. Self-selection biases may be present as partaking in Partners Biobank is optional. |
| Ethics oversight | Mass General Brigham Institutional Review Board (IRB) |

Note that full information on the approval of the study protocol must also be provided in the manuscript.

# Field-specific reporting

Please select the one below that is the best fit for your research. If you are not sure, read the appropriate sections before making your selection.

☒ Life sciences          ☐ Behavioural & social sciences          ☐ Ecological, evolutionary & environmental sciences

For a reference copy of the document with all sections, see <u>nature.com/documents/nr-reporting-summary-flat.pdf</u>

# Life sciences study design

All studies must disclose on these points even when the disclosure is negative.

| | |
|---|---|
| Sample size | Our primary analysis of public data included 1.7 million T cells from 905 samples from 695 individuals. This allowed us to define a comprehensive atlas of T cell states in diseases using some of the largest scRNA-seq datasets publicly available for T cells. As we were highly powered to resolve cell states in this large dataset, we then applied it to annotate cell states in our smaller experimental study, which included 43,222 cells across five samples from three stimulation conditions (stimulated, unstimulated, mock). We did not perform an analysis of number of samples necessary prior to performing the experiment. However, this sample size provided us many thousands of cells per donor, per stimulation condition. We were interested in testing the effects of simulation condition on cell states. This number of cells and samples allowed us to be powered to detect significant cell state differences between stimulation conditions. |
| Data exclusions | No data was excluded from analyses. |
| Replication | Data was collected on five participants (five replicates). All replications were successful and included in the data analysis. No further replication of data was performed. |
| Randomization | All cells from all samples were randomly sorted into two groups corresponding to peptide treated and mock-stimulated cells. |
| Blinding | Blinding was not relevant to the study as all samples were assigned to both peptide treated and mock-stimulated conditions. Data analysis was unbiased and tested differences between antigen-stimulation statuses. |

# Reporting for specific materials, systems and methods

We require information from authors about some types of materials, experimental systems and methods used in many studies. Here, indicate whether each material, system or method listed is relevant to your study. If you are not sure if a list item applies to your research, read the appropriate section before selecting a response.

#### Materials & experimental systems

| n/a | Involved in the study |
|---|---|
| ☐ | ☒ Antibodies |
| ☒ | ☐ Eukaryotic cell lines |
| ☒ | ☐ Palaeontology and archaeology |
| ☒ | ☐ Animals and other organisms |
| ☒ | ☐ Clinical data |
| ☒ | ☐ Dual use research of concern |

#### Methods

| n/a | Involved in the study |
|---|---|
| ☒ | ☐ ChIP-seq |
| ☐ | ☒ Flow cytometry |
| ☒ | ☐ MRI-based neuroimaging |

nature portfolio | reporting summary

March 2021

# Antibodies

| | |
|---|---|
| Antibodies used | Co-stimulation:<br>Anti-CD28 antibody, Biolegend, Catalog #: 302933 RRID: AB_11150591<br>Anti-CD49d antibody, Biolegend, Catalog #: 304339 RRID: AB_281044<br><br>Proteogenomics:<br>TotalSeq™-C Human Universal Cocktail, V1.0, Biolegend, Catalog #: 399905<br>Human TOTAL-SeqC Repertoire (5') Hashing Antibodies, BioLegend, Catalog #: 394661, 394663, 394665<br><br>Flow Cytometry:<br>Anti-CD3-BV421 (SK7), Biolegend, Catalog #: 344833 RRID: AB_2565674<br>Anti-CD134-PE (Ber-ACT35), Biolegend, Catalog #: 350003 RRID: AB_10641708<br>Anti-CD274-BV785 (29E.2A3), Biolegend, Catalog #: 329735 RRID: AB_2629581<br>Anti-CD137-APC (4-B4-1), Biolegend, Catalog #: 309809 RRID: AB_830671<br>Anti-CD4-FITC (RPA-T4), Biolegend, Catalog #: 300505 RRID: AB_314073 |
| Validation | All antibodies used are publicly available through Biolegend. Biolegend provides the following statements on its website:<br><br>Antibodies used for co-stimulation:<br>Each lot of this antibody is quality control tested by immunofluorescent staining with flow cytometric analysis. FC - Quality tested IHC-F, Costim - Reported in the literature, not verified in house<br><br>Antibodies used for proteogenomics:<br>TotalSeq™ Antibodies<br>Bulk lots are tested by PCR and sequencing to confirm the oligonucleotide barcodes. They are also tested by flow cytometry to ensure the antibodies recognize the proper cell populations.<br>Bottled lots are tested by PCR and sequencing to confirm the oligonucleotide barcodes.<br><br>Antibodies used for flow cytometry:<br>Each lot of this antibody is quality control tested by immunofluorescent staining with flow cytometric analysis. FC - Quality tested |

# Flow Cytometry

## Plots

Confirm that:

☒ The axis labels state the marker and fluorochrome used (e.g. CD4-FITC).

☒ The axis scales are clearly visible. Include numbers along axes only for bottom left plot of group (a 'group' is an analysis of identical markers).

☒ All plots are contour plots with outliers or pseudocolor plots.

☒ A numerical value for number of cells or percentage (with statistics) is provided.

## Methodology

| | |
|---|---|
| Sample preparation | PBMCs from 5 healthy donors were quickly thawed and placed in pre-warmed xVIVO15 cell culture medium (Lonza) supplemented with 5% heat-inactivated FBS. To reduce cell clumping, PBMCs were incubated in xVIVO15 containing 50 U/mL of benzonase nuclease (Sigma-Aldrich) for 15 minutes at 37 degrees and filtered using a 70 µm cell strainer. Washed and nuclease treated cells were seeded in a 96 well cell culture plate at a concentration of 2.5 x 106/mL. Peptide stimulations were performed using the CEFX Ultra SuperStim Pool (JPT Peptide Technologies, Product Code: PM-CEFX-1) at a final concentration of 1.25 µg/mL per peptide for 22 hours at 37 degrees and 5% $CO_2$. Recombinant anti-CD28 and anti-CD49d antibodies (BioLegend) were added at a final concentration of 5 µg/mL and 0.625 µg/mL, respectively, to provide co-stimulation for peptide reactive T-cells. Separately mock-stimulated cells were treated with anti-CD28 and anti-CD49d antibodies at the same concentration. Peptide responsive T-cells were detected by the expression of the surface activation markers PD-L1, OX40, and CD137 via flow cytometry. Following the stimulation, peptide treated and mock-stimulated cells were washed in cell staining buffer (PBS + 2mM EDTA + 2% FBS) to end the stimulation. Fc receptor blocking was performed using a 1:50 dilution of Human TruStain FcX (Biolegend) in cell staining buffer for 10 minutes at 4 degrees. Cell viability staining was performed using a 1:500 dilution of Zombie Yellow Fixable Viability Dye (BioLegend) prepared in PBS for 30 minutes at 4 degrees. Surface staining was performed using 1:100 dilutions of BV421 conjugated anti-CD3, FITC conjugated anti-CD4, BV786 conjugated anti-PD-L1, PE conjugated anti-OX40, and APC conjugated anti-CD137 (BioLegend) for 25 minutes at 4 degrees in cell staining buffer. Following cell staining, antigen reactive and non-reactive T-cells were identified using a BD FACSAria II cell sorter and collected in cRPMI medium (100 U/mL penicillin-streptomycin + 2 mM L-glutamine + 10 mM HEPES + 0.1 mM non-essential amino acids + 1 mM sodium pyruvate + .05 mM 2-Mercaptoethanol) supplemented with 20% FBS. |
| Instrument | BD FACSAria II |
| Software | BD FACSDiva 9.4 |

| | |
|---|---|
| Cell population abundance | CD4+ and CD3+CD4- cell populations were 58% and 25% of total live gated PBMCs in the peptide stimulated condition. Within peptide-stimulated CD4 T cells, 4.21% were AIM-positive and 81.7% were AIM-negative. Within peptide-stimulated CD8 T cells, 2.45% were AIM-positive and 74.9% were AIM-negative. |
| Gating strategy | Gating on CD3+CD4+ PBMCs isolated CD4 T cells. Gating on CD3+CD4- PBMCs isolated CD8 T cells. Gating on PDL1+OX40+ CD4 T cells was then performed to sort Antigen Induced Marker (AIM)-positive from AIM-negative cells. Gating on PDL1 +CD137+ CD8 T cells was performed to sort AIM-positive and AIM-negative cells. |

☒ Tick this box to confirm that a figure exemplifying the gating strategy is provided in the Supplementary Information.

