## [Peer Review File · Nature Methods]

Reproducible single cell annotation of programs underlying T cell subsets, activation states, and functions

Corresponding Author: Soumya Raychaudhuri

A version of this paper was originally rejected for publication by Nature Methods, however that decision was reconsidered after appeal by the authors.

Version 0:

Decision Letter:

28th Aug 2024

Dear Soumya,

One again, we apologize for the delay in reaching a decision on your manuscript. As I explained we had to assign a fourth review in order to be able to fairly assess your manuscript.

Your Resource entitled "Reproducible single cell annotation of programs underlying T-cell subsets, activation states, and functions" has now been seen by the fourth ref, whose comments are attached. While they find your work of potential interest, they have raised serious concerns which in our view are sufficiently important that they preclude publication of the work in Nature Methods, at least in its present form.

As you will see, the reviewers raise concerns about validation and the utility of this resource. We also thought, the manuscript would benefit from emphasizing the value this Resource can provide. Should further experimental data allow you to fully address these criticisms we would be willing to look at a revised manuscript (unless, of course, something similar has by then been accepted at Nature Methods or appeared elsewhere). This includes submission or publication of a portion of this work somewhere else. We hope you understand that until we have read the revised paper in its entirety we cannot promise that it will be sent back for peer-review.

If you are interested in revising this manuscript for submission to Nature Methods in the future, please contact me to discuss your appeal before making any revisions. Otherwise, we hope that you find the reviewers' comments helpful when preparing your paper for submission elsewhere.

Sincerely,
Madhura

Madhura Mukhopadhyay, PhD
Senior Editor
Nature Methods

Reviewers' Comments:

Reviewer #1:

Remarks to the Author:

1. gene expression program GEP needs to be defined and explained in the introduction
2. The abbreviation of the tool is confusing -- with "***". Not sure how it will be searchable on the web etc
3. Not sure what wildcard character stands for in the name of the tool

4. In the results section, the author says that the batch effect could not hinder the identification of irreproducible GEP. This needs to be explained in more detail
5. Line 135. Authors claimed that reference datasets shared only 90% of genes. This needs to be explained and explained why this is expected
6. Line 153. It would be helpful to provide the table with a description of all datasets being used with details about those databases. Also explain what criteria was used to select those public datasets
7. Line 151. Which other tissues in addition to blood were used
8. Line 160. p-value was not provided
9. Line 181. Mentions IGKV gene. The relationship to the T cell needs to be explained
10. Line 222. Mentioned ground truth. It needs to be explained why manual gating of surface proteins will provide ground truth. Also, it would be helpful to discuss if such a dataset indeed can be considered as ground truth or gold standard See this paper discussing the type of gold standard datasets <https://www.nature.com/articles/s41467-019-09406-4>

Reviewer #2:

Remarks to the Author:

In this paper by Kotlia et al., the authors effectively apply their previously published cNMF algorithm to create an annotator for T-cell gene expression programs, with successful results. Annotating T cell subsets and unifying prior descriptions of T cell functional states is a very common and important task for many studies involving single-cell sequencing analysis of T cells. Their aggregation and consolidation of 46 GEPs describing possible T cell states into a single easy-to-use package is potentially quite useful. Although this package and method does seem to be generally useful, it is not clear to me that this method is highly differentiated from standard approaches. For the examples provided, there also does not seem to be major new insights gained based on the use of this approach.

Comments:

It's a bit unclear why the authors chose to only select two GEPs to turn into a binary classification variable; for an annotator, it would be expected and helpful to have binary classifications for all programs, otherwise it's unclear how this would be an improvement over simply using a calculated gene score with a set of marker genes for a given phenotype/differentiation state. The multinomial labeling tool does this somewhat, but despite the high number of GEPs here, the level of granularity of T cell phenotypes identified by the multinomial labeling tool seems quite limited. As those phenotypes are all well-characterized by a few marker genes, it's unclear how this tool is a significant improvement over existing methods (e.g. Azimuth).

Lines 129-147; Fig S1:

Removing only 4 GEPs out of 16 for performance testing purposes seems low; what if a dataset had only CD4 or CD8 T cells for example? How would *CAT perform on such a dataset? Figure S1 is hard to interpret quantitatively; extracted accuracy values would be helpful.

Fig 2A:

The UMAP in Fig 2A does not appear to effectively separate or cluster the identified cell types; since these labels are being used as ground truth this may be concerning and should be addressed. A corresponding UMAP showing the multinomial label predictions would be helpful here.

Fig 2C, Fig S3C:

The figure description in S3C does not match the legend in the figure; is multilabel prediction the light blue color or the dark blue color? If the legend is correct, then for 2C and S3C, why does the multilabel prediction perform better than the single cGEP version (the explanation for how this was calculated is also somewhat unclear)?

Minor comments:

- The text in some figures is quite small and almost unreadable (e.g. Fig 1A, S2B)
- Color schemes in figures are completely mixed throughout; it would be clearer if the same color schemes were used for representing the same data types
- Typo in line 984

Reviewer #4:

Remarks to the Author:

A. Summary of the key results

This paper proposes a new way to analyse single T cell expression as being the sum of a variety of independent gene expression profiles, i.e. instead of an individual T cell being described as exhausted or expanding it can be thought of as occupying a state somewhere between both (and indeed up to 46 different GEPs).

To do this the authors applied the recently published cNMF algorithm, but first had to design a new batch normalisation strategy to allow the data from 7 previously published studies to be amalgamated in such a way as to be compatible with cNMF.

They find several GEPs to be represented in all the T cell datasets considered, though these include trivial GEPs such as cell cycle and cytoskeletal associated GEPs that should be in all cells which flatters the comparison. GEPs were annotated by manual inspection of the top weighted genes, and poorly supported GEPs removed.

TCAT GEPs were benchmarked against the previous tool NMFproj and clustering approaches using an independent CITEseq experiment. TCAT was then applied to the COMBAT covid database and shown to be able to deconvolute subsets of proliferating T cells which had previously been grouped together, and showed that the previous dataset contained many cells of poor quality that had not been filtered out. TCAT also discriminates between CD8+ T cells polarisation states.

The authors introduce their AIMseq technique, looking at TCR dependent T cell activation and cGEPs that correlate with antigen reactivity. Using the cGEP enriched in activated T cells they then generate an ASA score by which the antigen reactivity of a cell can be assessed, and use this to analyse T cells derived from various cancer samples (in part by determining a ASA threshold at which to specify a cell as antigen reactive). The authors show that tumor mutation burden and the proportion of antigen-specific T cells correlate as would be expected, and then further analyse the cGEP phenotype of bystander T cells.

B. Originality and significance: if not novel, please include reference

Originality: NMF approaches have recently been applied to circulating CD4+ T cells (Single-cell transcriptome landscape of circulating CD4+ T cell populations in autoimmune diseases - ScienceDirect); this new study seeks to define gene expression profiles more broadly and determines a larger number of GEPs in an effort to make a shared reference representation of cell states, in particular T cell states.

The significance is hard to determine: the authors are correct that having a well characterised reference panel of GEPs that (in theory) captures all possible T cell GEPs would be of value in mapping studies (especially smaller studies) onto a common representation of cell states is of value to the field of single cell sequencing. The downside is that the cNMF approach is conceptually harder to understand than simpler approaches using reference maps such as Azimuth and ProjectILs, which may limit adoption. However the authors have made an admirable effort to make their tool available on github, present appropriate reference implementations, and are working on a web based tool that can be deployed on remote compute resources, which should facilitate implementation.

As the costs of sequencing continue to fall, and the size and availability of single cell datasets increases, this tool will likely be of increasing value (in particular to meta analyses)

While the authors present a huge amount of data, this paper would benefit from less data and a more compelling use case – for example the Sakaguchi group paper puts a clear focus on autoimmunity, I'd encourage the authors to put more emphasis on the last third of their paper looking at cancer to emphasise the novelty or additional value (though I appreciate that *CAT has uses beyond TCAT).

C. Data & methodology: validity of approach, quality of data, quality of presentation

As a general point the paper seems undecided as to whether it's a technical paper for a specialist audience with a high background level of understanding, or a newly generalisable method that might be of interest to less specialist readers who have single cell sequencing data. Given the journal is Nature Methods the figure legends are excessively terse which can make the data hard to understand: for example Fig 1A providing the overview of the whole approach is described in a single line of text. The figure itself could be expanded upon (or at least made larger) to help the reader grasp the core concept of this paper, while Fig1B is difficult to read and could be better linked to Fig1A_step2.

Compared to the previous NMF paper this paper has a more detailed methods section and a similar or more detailed supplementary data. The main figures are less crowded than the supplementals and have been prepared with care commensurate for a top tier journal.

D. Appropriate use of statistics and treatment of uncertainties

I do not have the requisite background in statistics to comment on the mathematical suitability of the new normalisation techniques used to combine the 7 datasets. Some figures lack statistics, and/or the statistical tools to compare these types of highly complex single cell datasets are not yet well established.

The authors have made the code underlying every figure available to allow readers to repeat their analyses which is admirable and underscores their high level of technical ability. If one were to criticise this might benefit from a little more curation - for example Analysis/TCAT/Polarization_MarkerHeatmaps.ipynb contains a number of KeyErrors which suggest that this might not be the final version used.

E. Conclusions: robustness, validity, reliability

The authors demonstrate a number of use cases for *CAT using TCAT to analyse diverse T cell repertoires to show that they can more accurately deconvolute T cell subsets.

F. Suggested improvements: experiments, data for possible revision

In general review figures and legends for clarity.

Fig S5B not mentioned in text

Section 1

I would also suggest tying the results of single cell sequencing/TCAT more closely to what the scientific audience know about T cells – for example in Fig1D how are known markers like ENTPD1 and CXCL13 that happen not to be among the top genes

per GEP distributed? Some information here – such as that HLA-DRA and HLA-DRB are top genes in the HLA cGEP – doesn't add much information.

I was surprised to see CXCL13 as a top weighted gene in T follicular helped cells, as this has been reported by many groups (Platten/Green, Wu, Rosenberg) as being a key marker for tumor reactive T cells

Section 3

Fig 2B – Accuracy does not reflect false positive assessments, a metric like G-Mean might better show the improved performance of the TCAT multilabel approach vs the single cGEP especially for CD8 EM. Is this what the authors mean in Fig2C with “balanced accuracy”? Possibly this is stated in the methods, but would be easier to state this in the legend/results.

Section 4

While TCAT does deconvolute the proliferating T cell cluster well, many studies perform regression of cell cycle genes so this is more a limitation of the COMBAT Covid-19 dataset than a feature of TCAT (though it nicely shows how previous data benefits from mapping onto a common reference).

Looking at Fig S4 comparing the two covid databases COMBAT with UK-Covid, while TCAT can deconvolute proliferating T cells into constituent subsets to show the important role of the T peripheral helper cGEP in covid patients vs healthy, there is little explanation of the large differences between these two covid datasets. Could the authors comment on this – I believe this is important given that a core argument is that datasets can (should?) be mapped to a common reference panel of GEPs. Dataset quality seems to be part of the answer given that the authors later comment on the high incidence of a cGEP indicative of poor cell quality in the COMBAT study. Why is the proportion of CD8trm cGEP cells in the proliferative state so different between studies? I would speculate that some cGEP are dependent on genes with low expression, such that poor sample handling (e.g. long delay between sample acquisition and single cell capture) depletes some cGEP more than others? The authors used an overdispersion metric to identify key genes for the cNMF analysis, but does this come at a price of robustness? I would like to see some discussion of these differences

When the authors state “Cells with high usage of the CD4-Naïve cGEP expressed CD4 naïve markers including CD45RA protein and SELL RNA, confirming that clustering misclassified them as memory T-cells (Extended Data 4)” some memory T cells also express CD45RA and SELL (e.g. TEMRA or Tcm)

Section 5 & 6

By definition naïve cells should not have encountered their cognate antigen. Clearly the immune system is constantly generating new TCR clonotypes, so indeed some CEFX reactive T cells could be naïve, however can the authors rule out a false positive result? By sequencing TCR clonotypes (as in section 6) one would expect that clonotypes in the naïve population should be less clonal/more diverse, and that T cells sharing the same clonotypes should not be found in both groups. Some analysis of TCR clonality would support this analysis and give additional confidence in the scoring (some of which is described in Fig S6).

While I would agree that proliferation is a core response of TCR activation, using it to validate cGEP choice seems less useful – I believe Fig 4F is one of the least convincing in the paper, with very large error bars and no stats. This should be better explained.

The ASA data is very interesting, but again large differences between COMBAT and UK-covid data in terms of absolute percentage are not explained. I would like to see more on this part of the paper – for example why are some cGEPs specific to particular cancers, presumably due to cancer microenvironment factors or similar? Fig 5J starts to get at this with a correlation between tumor mutational load and exhaustion, though seeing how ASA correlates with survival of individual patients would be more convincing (though such datasets are only just becoming available).

A large number of recent papers have shown CXCL13 to be key marker for tumor-reactive T cells (i.e. not just T peripheral helper cells), while virus reactive T cells seem to show a slightly difference activation pattern. See previous points on CXCL13, I'd like to see this explored more: could it be that the use of a viral peptide pool in AIMseq gives slightly different cGEP enrichment than true tumor reactivity?

G. References: appropriate credit to previous work?

Yes

H. Clarity and context: lucidity of abstract/summary, appropriateness of abstract, introduction and conclusions

The abstract, introduction and discussion sections are very well written.

Although we cannot publish your paper, it may be appropriate for another journal in the Nature Portfolio. If you wish to explore the journals and transfer your manuscript please use our manuscript transfer portal. You will not have to re-supply manuscript metadata and files, unless you wish to make modifications. For more information, please see our http://www.nature.com/authors/author_resources/transfer_manuscripts.html?WT.mc_id=EMI_NPG_1511_AUTHORTRANSF&WT.ec_id=AUTHOR manuscript transfer FAQ page.

** For Nature Portfolio general information and news for authors, see <http://npg.nature.com/authors>.

Version 1:

Decision Letter:

18th Dec 2024

Dear Soumya,

Thank you for your letter asking us to reconsider our decision on your Resource, "Reproducible single cell annotation of programs underlying T-cell subsets, activation states, and functions". After careful consideration we have decided that we are willing to consider a revised version of your manuscript as you have outlined in your plan that highlights the technical advance and usability of your method..

- * include a point-by-point response to our referees and to any editorial suggestions
- * please underline/highlight any additions to the text or areas with other significant changes to facilitate review of the revised manuscript
- * address the points listed described below to conform to our open science requirements
- * ensure it complies with our general format requirements as set out in our guide to authors at www.nature.com/naturemethods
- * resubmit all the necessary files electronically by using the link below to access your home page

Link Redacted

We hope to receive your revised paper within ten weeks. If you cannot send it within this time, please let us know. In this event, we will still be happy to reconsider your paper at a later date so long as nothing similar has been accepted for publication at Nature Methods or published elsewhere.

OPEN SCIENCE REQUIREMENTS

REPORTING SUMMARY AND EDITORIAL POLICY CHECKLISTS

When revising your manuscript, please submit reporting summary and editorial policy checklists.

DATA AVAILABILITY

CODE AVAILABILITY

Please include a "Code Availability" subsection in the Online Methods which details how your custom code is made available. Only in rare cases (where code is not central to the main conclusions of the paper) is the statement "available upon request"

allowed (and reasons should be specified).

MATERIALS AVAILABILITY

SUPPLEMENTARY PROTOCOL

To help facilitate reproducibility and uptake of your method, we ask you to prepare a step-by-step Supplementary Protocol for the method described in this paper. We [encourage authors to share their step-by-step experimental protocols](https://www.nature.com/nature-research/editorial-policies/reporting-standards#protocols) on a protocol sharing platform of their choice and report the protocol DOI in the reference list. Nature Portfolio's protocols.io is a free-to-use and open resource for protocols; protocols deposited onto protocols.io are citable and can be linked from the published article. More details can be found at [protocols.io](https://www.protocols.io/help/publish-articles).

ORCID

Happy Holidays !

Sincerely,
Madhura

Madhura Mukhopadhyay, PhD
Senior Editor
Nature Methods

Version 2:

Decision Letter:

Our ref: NMETH-RS56554B

18th Feb 2025

Dear Soumya,

Thank you for submitting your revised manuscript "Reproducible single cell annotation of programs underlying T-cell subsets, activation states, and functions" (NMETH-RS56554B). It has now been seen by the original referees and their comments are below. The reviewers find that the paper has improved in revision, and therefore we'll be happy in principle to publish it in Nature Methods, pending minor revisions to satisfy the referees' final requests and to comply with our editorial and formatting guidelines.

TRANSPARENT PEER REVIEW

Nature Methods offers a transparent peer review option for new original research manuscripts submitted from 17th February 2021. We encourage increased transparency in peer review by publishing the reviewer comments, author rebuttal letters and

editorial decision letters if the authors agree. Such peer review material is made available as a supplementary peer review file. **Please state in the cover letter 'I wish to participate in transparent peer review' if you want to opt in, or 'I do not wish to participate in transparent peer review' if you don't.** Failure to state your preference will result in delays in accepting your manuscript for publication.

ORCID

Sincerely,
Madhura

Madhura Mukhopadhyay, PhD
Senior Editor
Nature Methods

Reviewer #2 (Remarks to the Author):

Reviewer #4 (Remarks to the Author):

In this revised manuscript the authors have added additional material to previous manuscript describing a *CAT, a new cell annotator pipeline that replaces discrete cell states (a hangover from the days of FACS analysis where cells were gated into bins) with flexible GEP (gene expression programs) that reflect how cells can have an intermediate or uncommitted state.

The authors are applying recent techniques (cNMF) into a new package that is readily accessible online. Compared to the previous version of this manuscript, the authors have addressed reviewer points in considerable detail: amongst others the authors have made the use of *CAT by the wider community even simpler by providing a website implementation (<https://immunogenomics.io/starcat/>), have performed benchmarking against existing tools showing superior performance, and have illustrated the utility of *CAT on real-world datasets concerning immune checkpoint inhibition. This resolves questions as to whether the authors' pre-trained GEPs will add value to the T cell community. While the *CAT results for the different clinical ICI datasets do not always generate the same results, there is considerable similarity which gives confidence in the robustness of the GEPs, and the authors reasonably point out that T cell states are affected by the tumor microenvironment, and may persist after ICI treatment, leading to different results in different tumor entities.

The paper is well written throughout, and now contains appropriate statistics for the few points where they were previously missing. The authors have also clarified figures and expanded methods where appropriate. As before the authors' code is fully accessible to the community, and extraneous or confusing code has been tidied up for this revised manuscript.

The authors have to my mind appropriately addressed all reviewer comments, and the extensive new analyses and comparisons underscore the value of the method. I am particularly happy to see a simple web implementation of *CAT making it accessible to the scientific community, in contrast to so many computational methods that gain little traction due to complex installation or use requirements. I am therefore happy to recommend accepting the manuscript in its current form

Reviewer #4 (Remarks on code availability):

I previously reviewed the code and see that the authors have made suggested edits, though I have not this time run the code - the new website the authors present makes this unnecessary.

Version 3:

Decision Letter:

11th Jul 2025

Dear Soumya,

I am pleased to inform you that your Resource, "Reproducible single cell annotation of programs underlying T cell subsets, activation states, and functions", has now been accepted for publication in Nature Methods. The received and accepted dates will be 24 May 2024 and 11 Jul 2025. This note is intended to let you know what to expect from us over the next month or so, and to let you know where to address any further questions.

Over the next few weeks, your paper will be copyedited to ensure that it conforms to Nature Methods style. Once your paper is typeset, you will receive an email with a link to choose the appropriate publishing options for your paper and our Author Services team will be in touch regarding any additional information that may be required. It is extremely important that you let us know now whether you will be difficult to contact over the next month. If this is the case, we ask that you send us the contact information (email, phone and fax) of someone who will be able to check the proofs and deal with any last-minute problems.

Authors may need to take specific actions to achieve compliance with funder and institutional open access mandates.

If your research is supported by a funder that requires immediate open access (e.g. according to [Plan S principles](https://www.springernature.com/gp/open-science/plan-s-compliance) or the [NIH public access policy](https://www.springernature.com/gp/open-science/us-federal-agency-compliance)) then you should select the gold OA route, and we will direct you to the compliant route where possible. Because authors warrant under our subscription licensing terms that they haven't committed to licensing any version of their article under a licence inconsistent with the terms of our agreement – including the applicable embargo period – publication under the subscription model isn't suitable for authors whose funders require no embargo.

If you are active on Twitter/X or Bluesky, please e-mail me your and your coauthors' handles so that we may tag you when the paper is published.

Best regards,
Madhura

Madhura Mukhopadhyay, PhD
Senior Editor
Nature Methods

** Visit the Springer Nature Editorial and Publishing website at www.springernature.com/editorial-and-publishing-jobs for more information about our career opportunities. If you have any questions please click here.**

We thank the reviewers and the editor for their thoughtful feedback on our manuscript. Overall,
we were encouraged by the comments. For example, reviewer #2 noted “*Annotating T cell*
*subsets and unifying prior descriptions of T cell functional states is a very common and*
*important task for many studies involving single-cell sequencing analysis of T cells.*” Reviewer
#4 noted “*having a well characterised reference panel of GEPs that (in theory) captures all*
*possible T cell GEPs would be of value in mapping studies (especially smaller studies)... [and] is*
*of value to the field of single cell sequencing... the authors have made an admirable effort to*
*make their tool available... which should facilitate implementation.*”

Based on the feedback from the reviewers, we have made four major revisions to emphasize
the unique value of the method and to address concerns about validation:

**Key Revision 1.** We analyzed 3 new cancer datasets showing that TCAT reveals gene
expression programs (GEPs) associated with immune checkpoint inhibitor responsive
and resistant tumors (**new Section 8, Figure 6, R2.1, R4.2, R4.15**).

**Key Revision 2.** We performed additional benchmarking against Azimuth, ProjectTILs,
and Symphony. TCAT out-performed all three for T-cell lineage prediction (**new Figure**
**2B-C, S3B, R2.1, R2.2, R4.2**).

**Key Revision 3.** We benchmarked TCAT’s activation prediction against 9 literature-
derived gene sets. TCAT out-performed 9/9 for prediction *in vivo* and 7/9 *ex vivo* (**new**
**Figure S6F-G, R2.1, R2.2**).

**Key Revision 4.** We created a web tool to expand starCAT beyond T-cells to any cell
type or tissue. We currently include published myeloid and bone marrow hematopoiesis
references (**R4.2**).

We describe the four major revisions in detail below and respond point by point to the reviewer
comments after that. We greatly appreciate the opportunity to submit a revised manuscript and
look forward to your response.

**Key revision 1 - Biological novelty**

We added a more clinically impactful analysis to address reviewer 2's concern that there "does
not seem to be major new insights gained based on the use of this approach" and reviewer 4's
comment that "seeing how ASA correlates with survival of individual patients" would be more
convincing. Using T-CellAnnoTator, we demonstrated important characteristics of CD4 T-cells
that are consistently associated with immune-checkpoint inhibitor (ICI) therapy response across
three cancer types (Section 8 and new figure 6 in the revised text). We re-analyzed
published scRNA-Seq from melanoma, colorectal cancer, and non-melanoma skin cancer using
TCAT. We found consistent up-regulation of antigen-specific activation and cell cycle cGEPs in
ICI non-responders and consistent up-regulation of naive T-cell cGEPs in ICI responders. These
findings are biologically and clinically impactful since they demonstrate new properties of tumor
immunology that are strongly and consistently associated with response to ICI. TCAT was
essential for this analysis since it enables fast and rigorous cross-comparisons of T cell gene
expression programs across datasets. For convenience, we reproduce the new section of the
manuscript and the corresponding figures below:

**8. Characterizing immune checkpoint inhibitor response using TCAT**

*We demonstrate an application of TCAT by identifying cGEPs that predict tumor*
*response to immune checkpoint inhibitors (ICIs). While ICIs are state of the art for*
*treating many cancer types, five-year survival following treatment remains poor for over*
*half of treated patients¹. There remains much to learn about the properties of tumor-*
*infiltrating T-cells that determine ICI response. We therefore applied TCAT to*
*melanoma², non-melanoma skin cancer (NMSC)³, and colorectal cancer (CRC)^{3,4}*
*datasets. Each of these datasets contained pre- and post-treatment tumors with*
*information about therapeutic response. This enabled us to define T-cell cGEPs that*
*potentially predict ICI clinical response.*

*To define the prominent T-cell states within tumors, we first examined the melanoma*
*dataset², which had the most pre-treatment and total samples (N=19 and 48*
*respectively). Application of TCAT revealed T-cell populations expressing antigen-*
*specific activation (ASA), exhaustion, cell cycle, and CD4-Naive signatures (Figure 6A).*
*We also noted a prominent subset of cells expressing TCF7 which was previously*
*associated with good ICI response².*

*Next, to find predictors of ICI response in melanoma, we systematically associated*
*cGEPs with ICI response in pre-treatment tumors (nine responders and ten non-*
*responders) for CD4 and CD8 T-cells, separately. In CD4 T-cells, non-responders had*
*significantly higher usage of cell cycle and activation cGEPs such as TIMD4/TIM3,*
*OX40/EBI3, CellCycle-Late-S, and CellCycle-G2M (P<0.05 two-tailed T-test, Table S8,*
*Figure 6B). To assess if similar associations were seen in an independent dataset, we*
*examined a non-melanoma skin cancer (NMSC) dataset³ containing seven non-*

responders and six responders, pre-treatment. There was significant concordance
between the melanoma and NMSC datasets in the direction of the associations for the
52 TCAT signatures, suggesting generally consistent signals (sign test $P=0.0016$,
**Figure S8A**). Focusing specifically on the melanoma treatment response cGEPs, we
observed a consistent association of cell cycle and activation-associated cGEPs such as
CellCycle-Late-S and TIMD4/TIM3 (one-tailed T-test $P=0.037$ and $P=0.046$, respectively,
**Figure 6C**). Meta-analysis of the associations between these two datasets was
statistically significant for many activation and cell cycle cGEPs including TIMD4/TIM3,
CellCycle-Late-S, Exhaustion, and ICOS/CD38 ($P<0.05$ for all, **Table S8**). Associations
were similarly significant for the combined ASA and cell cycle scores (Meta-analysis
$P=0.0072$ and $P=0.0036$ for ASA and cell cycle, respectively, **Figure 6E-F**). These
findings suggest that having a greater amount of TCR activated and/or proliferating T-
cells in pre-treatment skin cancers predicts a worse response to ICI.

We also observed higher usage of the CD4-Naive cGEP in responders for pre-treatment
melanoma and NMSC tumors (meta-analysis $P=0.0063$, **Figure 6G, Table S8**). This is
consistent with a previous observation that TCF7 expression is associated with a
positive response to ICI, as TCF7 is the 5th most strongly associated gene with the
CD4-Naive cGEP (**Table S2**). In melanoma, cells annotated as CD4 Naive represented
a distinct subset of the TCF7 expressing cells; this subset was especially enriched for
ICI responsiveness (**Figure 6A**). Consistent with this, we found increased expression of
the naive T-cells markers CCR7 and SELL in pseudobulked CD4 T-cells from pre-
treatment responders (meta-analysis $P=0.024$, 0.0016 , and 0.00047 for TCF7, SELL,
and CCR7, respectively, **Figure S8B-D**). Furthermore, the percentage of CD4 T-cells
classified as naive by TCAT's lineage prediction was also significantly associated with
ICI responsiveness (meta-analysis $P=0.016$, **Figure S8E**). These findings suggest that
increased tumor infiltrating naive CD4 T-cells may predict a good response to ICI.

We sought to replicate these analyses in a similar CRC dataset, but unfortunately, it only
had one pre-treatment non-responder sample. However, we note that there is strong
overall agreement in association results from samples that are pre-treatment or post
treatment initiation (**Table S8**). For example, there was a significant correlation in the
association effect sizes between pre-treatment and post-treatment samples for all three
datasets ($R=0.64$, 0.51 , 0.56 for melanoma, NMSC, and CRC respectively, $P<1\times 10^{-4}$,
**Figure S8F-H**). In addition, six out of ten cGEPs that were significantly increased in pre-
treatment non-responder melanomas were also significant in post-treatment samples,
including many cell cycle and activation cGEPs (**Figure 6B** and **Extended Data 10A**,
Fisher exact test $P=0.0039$). This suggests that some key immune states are driven by
the tumor environment and persist after ICI treatment.

Making the assumption that pre and post-treatment samples largely share many immune
states, we repeated the associations using linear regression in combined pre- and post-
treatment samples, modeling treatment status as a fixed effect and patient of origin with
random effects (**Methods**). With this approach, we observed results that were consistent

with the prior analyses. For CRC, this model showed increased activation and cell cycle
cGEPs in ICI non-responders (e.g. $P < 1 \times 10^{-6}$ for OX40/EBI3 and CellCycle-Late-S,
**Figure 6D**). CRC pre-treatment tumors followed the expected trend of lower ASA and
cell cycle scores in responders than non-responders (**Extended Data Figure 10B**).
Application of this model to the melanoma and NMSC datasets was consistent with the
previous analyses in pre-treatment tumors (**Extended Data Figure 10C-D**). These
distinct analyses support that our findings are robust and reproducible in three cancers.

Lastly, we repeated these analyses in CD8 T-cells (**Figure S8I-K**). As was the case for
CD4s, there was increased activation and cell cycle cGEPs usage in pre-treatment non-
responders (**Table S8**, meta-analysis $P < 0.05$ for CellCycle-G2M, CellCycle-Late-S,
TIMD4/TIM3, Exhaustion, ICOS/CD38, OX40/EBI3, and HLA). These cGEPs were also
significant in the mixed linear models analysis of CD8 T-cells from CRC, modeling
treatment status as a fixed effect and patient of origin as random effects ($P < 0.05$ for all).
Similarly, the CD8-Naive cGEP was significantly associated with a positive response to
ICI in pre-treatment samples (meta-analysis $P = 0.032$). Thus, the above findings from
CD4s are consistent in CD8s as well. These analyses illustrate how TCAT can help
reveal consistent and clinically significant patterns of GEPs across multiple datasets.

A

B

C

D

E

F

G

**New Figure 6. cGEPs associated with immune checkpoint inhibitor response.** (A)
UMAPs of the melanoma dataset showing TCAT predicted lineage; the CD4-Naive and
Exhaustion cGEPs; the ASA and cell cycle scores; TCF7 expression; and treatment
status and response. (B-D) Associations of cGEP usage with ICI response in CD4 T-
cells of (B) pre-treatment melanoma, (C) pre-treatment non-melanoma skin cancer
(NMSC) and (D) combined pre- and post- treatment colorectal cancer (CRC) controlling
for treatment timepoint and donor of origin. Dots are colored by cGEP type. X-axis
shows the average difference in usage between non-responders and responders or the
regression coefficient. Y-axis shows the $-\text{Log}_{10}$ two-tailed P-value. (E-G) Average ASA
score, Cell Cycle score, and CD4-Naive cGEP usage in pre-treatment melanomas and
NMSCs and combined pre- and post-treatment CRCs. The average scores are mean
and variance normalized. P-values are one-tailed T-tests for melanoma and NMSC and
mixed linear regression P-values for CRC. P-value in the title is a meta-analysis of the P-
values across the three cancer types. * indicates $P < 0.05$. Boxes show interquartile
range.

New Figure S8. cGEPs associated with immune checkpoint inhibitor response. (A) T-statistics from pre-treatment melanoma and NMSC associations. R denotes Pearson correlation. Dots are colored by cGEP type. (B-D) Average expression of naive T-cell marker genes. Melanoma and NMSC P -values are from one-tailed T-tests and CRC P -values are from a mixed linear regression likelihood ratio tests. P -value in the title is from a meta-analysis across the three cancer types. (E) Same as B but showing percentage of CD4 Naive T-cells out of total CD4 T-cells based on TCAT lineage classification. (F-G) T-statistics computed in pre- and post- treatment CD4 T-cells for melanoma and non-melanoma skin cancer (NMSC). (H) Same as F but showing mean difference rather than

178 *T*-statistic due to the presence of only one pre-treatment non-responder. (I-K) Same as
**Figure 6B-D** but for CD8 T-cells.

**Key revision 2 - Additional benchmarking**

We wanted to confirm that TCAT quantitatively and qualitatively outperformed commonly used
alternative reference mapping strategies. Hence, we benchmarked TCAT's lineage prediction
feature against Azimuth⁵ and ProjectTils⁶, the two reference mapping methods specifically
mentioned by the reviewers, and Symphony⁷, a commonly used categorical-annotation method
developed by our group. We found substantially higher prediction accuracy for classifying T-cell
subsets than these alternative methods in our validation Flu-Vaccine dataset (**New Figure 2B,**
**S3B**). For example, TCAT greatly outperformed other methods for classification of CD4 effector
memory cells in the Flu-Vaccine dataset (accuracy 0.80 compared to 0.57, 0.60, and 0.52 for
Azimuth, Symphony, and ProjectTILs respectively). TCAT also predicted TEMRA lineage
(accuracy 0.83), which was not predicted by any of these alternative methods. While discrete
lineage prediction is only a single facet of TCAT, it was reassuring that TCAT remains state of
the art for this task.

**New Figure 2B.** Cross-method comparison of balanced accuracy for prediction of
manually gated subsets.

 **New Figure S3B.** Proportion of cells within each manually gated lineage (columns)
 assigned to each lineage annotation (rows). Proportion of cells sums to 1 within each
 column, for each annotation method.

**Key revision 3 - Clarifying biological value of the method**

 Reviewer comments helped identify a lack of clarity about the method which we now specifically
 address in the text. Alternative methods such as Azimuth, ProjectTils, and Symphony label cells
 with a single mutually exclusive label. In contrast, starCAT simultaneously annotates cells with
 multiple distinct features. This avoids confounding of multiple unrelated signals, such as when
 cell activation or cell quality obscures T-cell subset, or vice versa. To illustrate the additional
 insights enabled by TCAT over these other methods, we added a new benchmark of annotating
 antigen-specific activation (ASA), a crucial feature of T-cells that the other reference transfer
 methods do not explicitly tackle. To address this task, researchers typically score cells based on
 literature-derived activation gene sets. However, *a priori*, it is difficult to know which gene set to
 use, and different gene sets can yield widely varying results. Our manuscript adds biological
 value by developing two novel benchmarks to address this: (1) prediction of AIM-Seq, an
 experimental assay of T-cell activation, and (2) prediction of a CITE-Seq derived activation
 label. We compared TCAT's ASA score against 9 literature-derived gene sets for classifying
 activation based on surface activation markers in the Flu-Vaccine and AIM-Seq datasets. ASA

outperformed all gene sets for activation classification in the Flu-Vaccine dataset and all but 2
 gene sets in the AIM-Seq data (New Figure S6F-G, Extended Data 8B-C). Thus, in addition to
 outperforming reference label transfer methods for lineage classification, it also outperformed
 most literature-defined gene sets for classifying T-cell activation. We also added emphasis in
 the manuscript that starCAT concurrently annotates a broad range of T-cell features, such as
 polarization and exhaustion, which existing reference transfer pipelines currently do not predict.

 **New Figure S6. (F)** In the Flu-Vaccine dataset, AUCs for prediction of CD71+CD95+ co-
 expression by ASA as compared to cell scoring with alternative T-cell gene sets. Full
 gene set names are GOBP_ACTIVATED_T_CELL_PROLIFERATION (GOBP_ACT),
 GSE13738_RESTING_VS_TCR_ACTIVATED_CD4_TCELL_DN (GSE13738_RVT),
 GSE13738_TCR_VS_BYSTANDER_ACTIVATED_CD4_TCELL_UP (GSE13738_TVB),
 GSE15659_CD45RA_NEG_CD4_TCELL_VS_ACTIVATED_TREG_DN
 (GSE15659_CVA), GSE15659_NAIVE_CD4_TCELL_VS_ACTIVATED_TREG_DN
 (GSE15659_NAVA),
 GSE15659_NONSUPPRESSIVE_TCELL_VS_ACTIVATED_TREG_DN
 (GSE15659_NSVA), GSE15659_RESTING_VS_ACTIVATED_TREG_DN
 (GSE15659_RVA), GSE28726_NAIVE_VS_ACTIVATED_CD4_TCELL_DN
 (GSE28726_NVA), WP_TCELL_ACTIVATION_SARSCOV2 (WP_ACTCOV). (G) Same
 as (F) but prediction of AIM-positivity in the AIM-Seq dataset.

**New Extended Data 8. (B)** ROC curves for ASA prediction of CD71+CD95+ in the Flu-
Vaccine dataset, as compared to prediction using per-cell gene scoring with T-cell
activation gene sets. (C) Same as (B) for AIM-positivity prediction in the AIM-Seq

dataset. Full gene set names (in order) are

GOBP_ACTIVATED_T_CELL_PROLIFERATION,

GSE13738_RESTING_VS_TCR_ACTIVATED_CD4_TCELL_DN,

GSE13738_TCR_VS_BYSTANDER_ACTIVATED_CD4_TCELL_UP,

GSE15659_CD45RA_NEG_CD4_TCELL_VS_ACTIVATED_TREG_DN,

GSE15659_NAIVE_CD4_TCELL_VS_ACTIVATED_TREG_DN,

GSE15659_NONSUPPRESSIVE_TCELL_VS_ACTIVATED_TREG_DN,

GSE15659_RESTING_VS_ACTIVATED_TREG_DN,

GSE28726_NAIVE_VS_ACTIVATED_CD4_TCELL_DN,

WP_TCELL_ACTIVATION_SARSCOV2.

Key revision 4 - Application to other cell types

Additionally, we created a web tool to expand the application of starCAT beyond T-cells to any

cell type or tissue. Since its publication in 2019, consensus non-negative matrix factorization

(cNMF) has become increasingly popular for characterizing cell states. Each time it is used

creates the opportunity to develop a reference that can be applied to new query datasets,

enabling researchers to annotate their datasets using a shared set of biological GEPs. To

broaden the use of starCAT, we created a website <https://immunogenomics.io/starcat> to host

such references and enable users to annotate their data from the browser. We use

WebAssembly, a recently developed, powerful software infrastructure that enables all of the

computation to be run in the web browser without requiring user data to be uploaded to a

remote server⁸. In addition to our T-cell reference, we have already included a human glioma

myeloid cell reference⁹ and a human bone marrow hematopoiesis reference¹⁰. These

references can also be used easily from Python, the command line, or through the starCAT

website. This database substantially expands the impact of our work beyond T-cells to all cell

types, greatly increasing the utility of our resource.

**Reviewer #1 comments:**

**R1.1. gene expression program GEP needs to be defined and explained in the**
**introduction**

We thank the reviewer for their comment. We have added additional text in the introduction to
define gene expression program:

*Transcriptomes reflect the expression of multiple gene expression programs (GEPs) –*
*co-regulated genes that are co-expressed for specific biologic functions such as defining*
*a cell type, activation state, lifecycle process, or response to an external stimulus¹¹.*

**R1.2. The abbreviation of the tool is confusing -- with "*". Not sure how it will be**
**searchable on the web etc**

**R1.3. Not sure what wildcard character stands for in the name of the tool**

We thank the reviewer for these comments. We now refer to the tool as starCAT throughout the
manuscript, rather than *CAT. We agree that this will improve the searchability of this term and
its code. We have added the extra text below to better explain the name:

*Here, we present star-CellAnnoTator (starCAT), an approach to score cells based on a*
*fixed, multidataset catalog of GEPs. The wildcard character "*", written as "star" in the*
*name, indicates that the approach can be applied using a reference from any tissue or*
*cell-type, whereas our specific instantiation for T-cell GEPs is written as T-CellAnnoTator*
*or TCAT.*

**R1.4. In the results section, the author says that the batch effect could not hinder the**
**identification of irreproducible GEP. This needs to be explained in more detail**

We thank the reviewer for this comment. Batch effects can result in learning GEPs reflecting
non-generalizable dataset-specific signals. For example, running cNMF prior to batch-correction
identified multiple highly correlated versions of the Translation GEP, where each version was
significantly enriched within one batch. However, after batch correction, a single GEP was used
consistently across the batches and had higher correlation across datasets. To make this
potential pitfall more clear to the readers, we have added the following additional text to the
introduction:

*For our approach, it was essential to amalgamate the GEP spectra from multiple*
*datasets. However, we found that batch effects could sometimes cause cNMF to learn*
*multiple batch-specific versions of a GEP, resulting in redundant GEPs with less*
*correlation across datasets. Most batch correction methods are not compatible with*
*cNMF since they create many negative values or correct low-dimensional embeddings*
*rather than gene-level data. We therefore modified Harmony¹² to produce batch-*
*corrected non-negative values for gene-level data rather than principal components.*

R1.5. Line 135. Authors claimed that reference datasets shared only 90% of genes. This needs to be explained and explained why this is expected

We thank the reviewer for this comment. Public datasets may share only a subset of the same genes due to differences in alignment pipelines, reference transcriptome versions, and filtering steps taken by the generators of the dataset. We therefore wanted to ensure that our simulation was robust to non-overlapping gene lists. We have added the text below to clarify why we chose this simulation parameter:

To ensure that starCAT is robust to incompletely overlapping genes between the reference and query dataset, we simulated these datasets to share only 90% of genes in common.

R1.6. Line 153. It would be helpful to provide the table with a description of all datasets being used with details about those databases. Also explain what criteria was used to select those public datasets

We thank the reviewer for their comment. We have added a new tab to **Table S1** providing information about the reference and validation datasets included in the paper. We have also added text clarifying the criteria we used to select these datasets:

We analyzed T-cells from 7 diverse datasets including blood and tissues from healthy individuals or individuals with Covid-19, cancer, rheumatoid arthritis, or osteoarthritis (Table S1, Figure S1F). In addition to selecting datasets for a breadth of phenotypes, we also selected for large sample sizes (>70,000 T-cells each), and inclusion of CITE-Seq data, when possible, to aid in GEP interpretation. We also included two Covid-19 PBMC, two healthy PBMC, and two tissue datasets each to help quantify cross-dataset GEP reproducibility.

R1.7. Line 151. Which other tissues in addition to blood were used

We thank the reviewer for their comment. We have included with the new **Table S1** a list of all tissues included for each dataset. This includes the following: joint synovium, lung, lung-draining lymph node, mesenteric lymph node, bone marrow, thymus, skeletal muscle, liver, spleen, omentum, duodenum, jejeunal lamina propria, jejeunum epithelium, Ileum, caecum, transverse colon, sigmoid colon, basal cell carcinoma, B-cell lymphoma, bladder cancer, esophageal cancer, fallopian tube carcinoma, hepatocellular carcinoma, lung cancer, multiple myeloma, ovarian cancer, pancreatic cancer, renal carcinoma, thyroid carcinoma, and uterine carcinoma.

R1.8. Line 160. p-value was not provided

We thank the reviewer for their comment. We have provided the P-values for the correlation statistics in the text ($P < 1 \times 10^{-50}$ for all pairs of GEPs).

**R1.9. Line 181. Mentions IGKV gene. The relationship to the T cell needs to be explained**

We thank the reviewer for their comment and recognize that this reference to an
immunoglobulin gene in a T-cell paper may be surprising. Although each of the discovery
datasets was subset specifically to T-cells, there are still cases where a non-T-cell may have
escaped the filters, likely due to being a doublet with a T-cell (i.e. when a T-cell and non T-cell
are captured in the same droplet). Thus, cNMF can learn some gene expression programs
(GEPs) that correspond to non-T-cell cell-types. We suspect the cGEP containing *IGKV*
reflected a plasmablast doublet GEP and thus labeled it as such. We have clarified this by
adding the following additional text:

*In addition to the 46 T-cell cGEPs, we identified six cGEPs corresponding to non T-cell*
*populations, including erythrocytes (HBA2, HBA1, HBB) and plasmablasts (JCHAIN,*
*IGKC, IGKV3-20). We suspect these cGEPs were due to residual doublets in the*
*datasets and retained them to flag doublet-associated transcriptional signals.*

**R1.10. Line 222. Mentioned ground truth. It needs to be explained why manual gating of**
**surface proteins will provide ground truth. Also, it would be helpful to discuss if such a**
**dataset indeed can be considered as ground truth or gold standard See this paper**
**discussing the type of gold standard datasets [https://www.nature.com/articles/s41467-](https://www.nature.com/articles/s41467-019-09406-4)**
**[019-09406-4](https://www.nature.com/articles/s41467-019-09406-4)**

We thank the reviewer for this comment. We agree that defining cell subsets based on manual
gating of a small number of surface markers is fundamentally limited. However, flow cytometry
based gating of canonical surface protein markers is the primary way immunologists have
traditionally defined T-cell subsets. Thus, much of immunology knowledge about T-cell subsets
relates to populations defined based on these markers. Consequently, this approach is
frequently used as a gold standard in computational immunology, e.g. in benchmarking studies
of bulk deconvolution¹³ or single-cell RNA-Seq cell type annotation¹⁴.

While we used gating of CITE-Seq data rather than flow cytometry for our predictive task, the
data looked very comparable to standard flow cytometry T-cell gating (**See Figure S3A**
**reproduced below for convenience**). Even if the gating is unlikely to be 100% accurate for
classifying functionally specific T-cell subsets, we still find it useful as a feature to be predicted
in this analysis. First, it is based on protein data and is therefore independent of the
transcriptome-features used for prediction. Second, it connects back to the conventional flow
cytometry paradigm used in traditional T-cell immunology and can thus ground single-cell
analyses in the literature.

Thus, we have removed the term “ground truth” from this section of the text and instead focus
on explaining why this feature is a useful one to predict in the task. We have revised the text as
below to explain this:

We used manual gating of surface proteins to define ten conventional T-cell subsets as
 prediction targets (**Figure S3A**). These protein-based annotations are useful for
 benchmarking as they are measured independently from the transcriptome-based
 features used for prediction and connect directly to the wealth of literature about
 canonical flow-cytometry defined T-cell subsets.

**Figure S3. Benchmarking starCellAnnoTator on simulated and real datasets.** (A) Manual
 gating of Flu-Vaccine dataset analogous to **Figure S2A**.

**Reviewer #2 comments:**

 **R2.1** In this paper by Kotlia et al., the authors effectively apply their previously published
 **cNMF** algorithm to create an annotator for T-cell gene expression programs, with
 **successful results**. Annotating T cell subsets and unifying prior descriptions of T cell
 **functional states** is a very common and important task for many studies involving single-
 **cell sequencing analysis** of T cells. Their aggregation and consolidation of 46 GEPs
 **describing possible T cell states** into a single easy-to-use package is potentially quite
 **useful**. Although this package and method does seem to be generally useful, it is not
 **clear to me** that this method is highly differentiated from standard approaches. For the
 **examples provided**, there also does not seem to be major new insights gained based on
 **the use of this approach**.

 We thank the reviewer for their comment about the general usefulness of the approach.

 To address the reviewer's point about differentiation from standard approaches, we have added
 new analyses (**Key revision 2-3, above**) demonstrating how it outperforms reference transfer
 algorithms for annotating cell types while also outperforming gene set based approaches for
 annotating T-cell activation. See response to **R2.2** for a further discussion about how the
 approach is distinct from previous methods.

To address the reviewer's point about major new insights, we have added a new analysis using
TCAT to identify novel factors that predict tumor responsiveness to immune checkpoint inhibitor
(ICI) therapies (**Key revision 1, above**). Briefly, TCAT enabled us to quickly annotate multiple
tumor datasets with a consistent set of gene expression features, demonstrating a reproducible
finding of increased activation and proliferation gene sets in CD4 and CD8 T-cells across
multiple tumor types in non-responders. It also showed consistently increased CD4-naive cGEP
usage in responders. We believe this represents a significant new insight that was enabled by
TCAT and illustrates the utility of the approach. Accordingly, we have also added the following
text to the discussion:

*TCAT enabled us to identify novel properties of tumor-infiltrating CD4 and CD8 T-cells*
*that predict response to immune checkpoint inhibitor (ICI) therapy (**Figure 6**). Across*
*three cancer types, non-responsive tumors were enriched for exhaustion, activation, and*
*cell cycle cGEPs. This is potentially surprising as, a priori, one might assume that more*
*immunologically active tumors would respond better to ICI therapy. Instead, these*
*findings suggest that ICI non-responsive tumors contain a higher fraction of activated,*
*proliferating, and exhausted CD4 T-cells reflecting an enhanced ability to suppress*
*effective cytotoxic responses. By contrast ICI responsive samples were enriched for the*
*CD4-Naive cGEP, consistent with previous associations of ICI response with stem-like*
*T-cells marked by TCF7². The presence of these cells may suggest the potential of ICI to*
*reinvigorate infiltrating T-cells, or alternatively, may suggest the inability of the cancer*
*cells to fully exhaust and terminally differentiate all T-cells in the tumor environment.*
*These results illustrate how TCAT can aid in characterizing clinically impactful cell states*
*in vivo and can help with predicting therapy response.*

**R2.2 It's a bit unclear why the authors chose to only select two GEPs to turn into a binary**
**classification variable; for an annotator, it would be expected and helpful to have binary**
**classifications for all programs, otherwise it's unclear how this would be an improvement**
**over simply using a calculated gene score with a set of marker genes for a given**
**phenotype/differentiation state. The multinomial labeling tool does this somewhat, but**
**despite the high number of GEPs here, the level of granularity of T cell phenotypes**
**identified by the multinomial labeling tool seems quite limited. As those phenotypes are**
**all well-characterized by a few marker genes, it's unclear how this tool is a significant**
**improvement over existing methods (e.g. Azimuth).**

We thank the reviewer for their comment and believe it represents a significant point of
confusion about this method which we have sought to improve in the revision. The fundamental
advantage of TCAT is the ability to simultaneously annotate a dataset with multiple GEPs that
are specifically relevant to T-cells and have been defined in a consistent and reproducible
fashion. While it can output some discrete annotations for cases where we had orthogonal data
to train a discrete classifier (see below), it primarily outputs continuous GEP scores. While this
is similar to calculating scores from a literature-defined gene set, TCAT possesses several
advantages over this common approach. First, GEPs used in TCAT provide different weights for

each gene, rather than arbitrarily weighting each gene uniformly as in gene set scoring. This
can increase accuracy in scoring GEP activity and can increase the ability to resolve related
GEPs that may share genes in common but with different weights. In addition, TCAT scores
each GEP simultaneously, which allows the relative usage of multiple GEPs to be compared for
each cell. This mitigates the risk of confounding multiple related signals, which may happen with
gene set scoring. For example, exhausted T-cells may score highly for both an exhaustion and
an activation gene set because of overlapping genes between these pathways. TCAT mitigates
this by simultaneously modeling the exhaustion and activation GEPs so they can be directly
compared. Lastly, there are often multiple gene sets available for a given process which may be
derived from distinct experimental contexts, and it can be hard to know which is the most
appropriate to use. TCAT provides a practical advantage of expediently scoring a
comprehensive set of GEPs that are derived in a consistent manner from *in vivo* data with
demonstrated generalizability across multiple scRNA-Seq datasets. We add the new text below
to discuss these advantages of TCAT in the introduction:

*It is possible to compute continuous GEP scores for each cell in a dataset, similar to the*
*common practice of scoring cells based on a predetermined gene sets¹⁵. However,*
*unlike gene sets, component-based models account for variable gene weights,*
*simultaneously model multiple GEPs which reduces confounding of related signals and*
*enables comparison of relative GEP activities, and use GEPs derived from a consistent*
*in vivo context rather than from potentially diverse experimental systems like microarrays*
*or RNA-Seq.*

In addition to these theoretical benefits of TCAT over gene set scoring, we also now perform
direct empirical benchmarking (**Key revision 3, above**) showing that TCAT outperforms nine
out of nine literature-derived activation gene sets for predicting T-cell activation in an *in vivo*
dataset and outperforms seven out of the nine for predicting activation in the AIM-Seq dataset
(**New Figure S6F-G, Extended Data 8B-C, above**). We added the following text to the
manuscript to highlight this:

*We benchmarked the ASA score against T-cell activation gene sets for predicting T-cell*
*activation as defined above in the AIM-Seq and Flu-Vaccine datasets, two validation*
*datasets for this approach. ASA obtained higher accuracy than 9/9 literature-derived T-*
*cell activation gene sets in the Flu-Vaccine dataset and 7/9 gene sets in the AIM-Seq*
*dataset (**Figure S6F-G**). Thus, using the TCAT-derived ASA score can yield more*
*accurate inferences of antigen activation than gene sets both in vivo and ex vivo.*

We also previously showed that TCAT out-performed a set of scRNA-seq derived T-cell gene
sets¹⁶ for annotation of cellular lineage (**Figure S3C, reproduced below for convenience**).

C

Figure S3C. Benchmarking CellAnnoTator on simulated and real datasets. Area under the curve (AUC) for prediction of manually gated subset based on a single most associated subset (dark blue), TCAT multilabel prediction (light blue), analogous predictions using the single most associated NMF component published in (Yasumizu et al., 2024), or using gene sets from NMF components in Gavish et al., 202314.

These findings show empirically that TCAT provides a significant advantage over gene set annotation for scoring of expression GEPs.

We believe that much of the confusion about the method derives from the fact that it also provides the user with several discrete phenotype annotations (subset, antigen-specific activation, and cell cycle) in addition to the continuous GEP scores. We do so because in certain cases, it is useful to have a discrete annotation in addition to a continuous score. For predicting subset and antigen-specific activation (ASA), we could use CITE-Seq data as an orthogonal ground truth label to train a discrete classifier for these features. For most other features, we lack such a ground truth and thus were unable to quantify specificity and sensitivity for specific thresholds. We therefore have opted to not discretize other features with the exception of cell cycle, a metric often discretized in scRNA-seq processing pipelines. We clarified this point in the revised manuscript by adding the following text and modifying our schematic Figure 1A to show these two distinct results (**Revised Figure 1A**).

In addition, we utilize these cGEPs to predict and return annotations for additional protein-informed per-cell labels, including lineage, activation score, and cell cycle score.

*We then also utilize the per-cell GEP usages to predict additional per-cell features, including lineage, activation, and cell cycle (**Figure 1A - bottom**).*

Revised Figure 1A. Overview of star-CellAnnoTator (starCAT). Schematic of the starCAT pipeline.

Lastly, we demonstrated that TCAT outperforms three commonly used reference label transfer
softwares (Azimuth, ProjectTILs, and Symphony) for subset prediction (**Key revision 2, above**).
While this task is only one small part of what TCAT does, it was useful to demonstrate that
TCAT outperforms the state of the art methods for this task.

**R2.3 Lines 129-147; Fig S1:**

**Removing only 4 GEPs out of 16 for performance testing purposes seems low; what if a**
**dataset had only CD4 or CD8 T cells for example? How would *CAT perform on such a**
**dataset? Figure S1 is hard to interpret quantitatively; extracted accuracy values would be**
**helpful.**

We thank the reviewer for these comments. To address the first point, we have now tested the
accuracy of using a starCAT reference with 20 GEPs in the reference, of which only 8 are in the
query (analogous to using a reference trained on both CD4 and CD8 T-cells to predict solely
CD8 T-cells). We also tested the inverse where a reference contains only 8 GEPs but the query
contains 16. We found generally comparable results to what we presented previously in **Figure**
**S1 (New Extended Data Figure 1** reproduced below for convenience). We now include the
following additional text in the manuscript describing this finding:

*We tested the robustness of starCAT when a lower fraction of GEPs overlapped*
*between the reference and query datasets. We observed similar prediction accuracies*
*when predicting a simulated query dataset with 8 GEPs using a reference with 20 GEPs,*
*and predicting a query with 16 GEPs using a reference with only 8 GEPs (**Extended***
***Data Figure 1**). This shows starCAT can be robust to references and queries that*
*contain a substantially smaller fraction of overlapping GEPs.*

 **New Extended Data Figure 1. Additional starCAT simulation analyses** (A) Pearson
 correlation of ground truth simulated usages of each GEP (columns) vs inferred usages
 (rows) for starCAT using a simulated 20 GEP reference to predict an 8 GEP query
 dataset, where the 8 GEPs overlap with the reference. (B) Same as A but using an 8
 GEP reference and a 16 GEP query dataset, containing 8 overlapping GEPs. (C)
 Extracted pearson correlation values from (A) and (B) indicating whether an inferred
 GEP matched the ground truth GEP based on the color.

Regarding the second comment, we agree that the correlation heatmap figures in **Figure S1**
 could be confusing to the reader. Therefore, to help with interpretation, we have added a
 summary of the correlation between the inferred GEP usages and the corresponding simulated
 ground truth usages (**Revised Figure S1C**). The stripplot shows that the starCAT reference with
 extra or missing cGEPs had similarly good correlations for their overlapping GEPs in the query
 dataset, while cNMF had less accurate GEP inferences for 2 out of the 16 GEPs.

 **Revised Figure S1C. Characterizing starCAT.** Pearson correlation between inferred
 gene expression programs and the corresponding ground truth usages, extracted from
 (B).

**R2.4 Fig 2A:**
**The UMAP in Fig 2A does not appear to effectively separate or cluster the identified cell**
**types; since these labels are being used as ground truth this may be concerning and**
**should be addressed. A corresponding UMAP showing the multinomial label predictions**
**would be helpful here.**

We thank the reviewer for this comment. We believe this comment suggests a useful way for us
to help illustrate an important aspect of our manuscript which we have hoped to clarify. For this
annotation (previously called “ground truth”), we used CITE-seq protein markers to separate
canonical subsets of T-cells, an approach analogous to traditional isolation of T-cell lineages by
immunologists using flow cytometry sorting on protein markers (**See R1.10**). However, single
cell transcriptomes reflect additional gene expression programs besides cell lineage, such as
functional (ex. cell cycle, activation) or artifact gene expression programs. Thus, cells may not
cleanly separate by lineage on a UMAP.

In addition to adding a new UMAP showing the multinomial label predictions (**Revised Figure**
**2A** reproduced below), we have added the following text to the corresponding results section to
clarify this:

*These protein-based annotations are useful for benchmarking as they are measured*
*independently from the transcriptome-based features used for prediction and connect*
*directly to the wealth of literature about canonical flow-cytometry defined T-cell subsets.*
*Protein-derived subsets largely separated on a gene-expression derived UMAP (**Figure***
***2A - left**). However, there was also significant overlap of memory populations, which*
*may reflect the impact of non-subset-specific signals influencing the transcriptome,*
*which we further explored in the next section.*

**Revised Figure 2A.** Left - UMAP of the Flu-Vaccine dataset colored by the manually
gating shown in **Figure S3A**. Right - UMAP colored by TCAT multinomial label
prediction.

**R2.5 Fig 2C, Fig S3C:**
**The figure description in S3C does not match the legend in the figure; is multilabel**
**prediction the light blue color or the dark blue color? If the legend is correct, then for 2C**
**and S3C, why does the multilabel prediction perform better than the single cGEP version**
**(the explanation for how this was calculated is also somewhat unclear)?**

We thank the reviewer for this comment. For the previous **Figure 2C and S3B** (which are
**Extended Data 5A-B** in the revision), and **Figure S3C** (which remains as is), the dark blue line
indicates the multilabel prediction and the light blue line indicates prediction using the single
cGEP that was most correlated with the subset. For multilabel prediction, we fit a multinomial
logistic regression that predicts the probability a cell belongs to each subset as a function of its
usage of all cGEPs. To determine the AUC for a given subset prediction using this model, we
obtain a receiver operator characteristic (ROC) by varying the threshold on the probability
assigned to that subset between 0 and 1. By contrast, for the single cGEP prediction, we
identify the single cGEP that has the best AUC for classifying each subset using a ROC based
on varying the thresholds on the usage of that cGEP between 0 and 1. The multilabel prediction
can outperform the single cGEP prediction by leveraging the ability to combine multiple cGEPs
rather than just using a single cGEP. This is especially helpful for complex subsets such as CD4
EM which benefit from considering multiple effector states (e.g. Th17, Th2, cytotoxic, etc.). We
have added additional text to help clarify this point for the reader:

*Since subsets contain heterogeneity not captured in univariate analysis such as multiple*
*polarized populations within CD4 effectors, we performed multivariate analysis using all*
*cGEPs for simultaneous multi-label prediction (**Methods, Figure 2A - right**). We trained*
*the classifier on the COMBAT dataset and evaluated its performance on the Flu-Vaccine*
*dataset. Across lineages, the classifier outperformed commonly used categorical*
*reference mapping methods, including Azimuth⁵, Symphony⁷, and ProjecTILs⁶, as well*
*as transcriptome clustering (Figure 2B-C). For example, TCAT greatly outperformed*
*other methods for classification of CD4 effector memory cells in the Flu-Vaccine dataset*
*(accuracy 0.80 compared to 0.57, 0.60, and 0.52 for Azimuth, Symphony, and*
*ProjecTILs respectively). TCAT also predicted TEMRA lineage (accuracy 0.83), which*
*was not predicted by any of these alternative methods. The TCAT classifier was also*
*more accurate than de novo clustering across all nine clustering resolutions tested, with*
*average balanced accuracy differences ranging from 0.10 to 0.032 (**Figure 2B-C,***
***Extended Data Figure 5A**). By incorporating information from multiple cGEPs, the multi-*
*label predictor yielded increased prediction accuracy than the single most associated*
*cGEP for eight out of ten lineages.*

We have also updated the description of **Extended Data 5B** to correctly match the figure
legends.

*Receiver operator curves (ROCs) for prediction of manually gated subset based on*
*TCAT multilabel prediction (dark blue), a single most associated TCAT cGEP (light*
*blue), analogous predictions using the single most associated NMF component*

*published in Yasumizu et al., 2024¹¹, or using gene sets from NMF components in*
*Gavish et al., 2023¹⁴. Individual points show accuracies of discrete predictions based on*
*cGEP multilabel regression, or clustering with the Leiden resolution specified in the*
*legend.*

**R2.6 The text in some figures is quite small and almost unreadable (e.g. Fig 1A, S2B)**

We thank the reviewer for their comment. We have improved the legibility of text in figures,
including in Fig 1A and S2B.

**R2.7 Color schemes in figures are completely mixed throughout; it would be clearer if the** 687 **same color schemes were used for representing the same data types**

We thank the reviewer for their comment. We have updated the figure color schemes to be
consistent throughout the manuscript. For instance, we used a single color scheme for plotting
cGEP usages throughout the manuscript (e.g. **Figure 3B-D, Figure S4B-C, Figure 5D, Figure**
**6A**) and for coloring cell lineages (**Figure 1A, Figure 2A, Figure 4D**).

**R2.8 Typo in line 984**

We thank the reviewer for their comment. We have edited this text.

**Reviewer #4 comments:**

**R4.1**

**A. Summary of the key result**

**This paper proposes a new way to analyse single T cell expression as being the sum of a**
**variety of independent gene expression profiles, i.e. instead of an individual T cell being**
**described as exhausted or expanding it can be thought of as occupying a state**
**somewhere between both (and indeed up to 46 different GEPs).**

**To do this the authors applied the recently published cNMF algorithm, but first had to**
**design a new batch normalisation strategy to allow the data from 7 previously published**
**studies to be amalgamated in such a way as to be compatible with cNMF.**

**They find several GEPs to be represented in all the T cell datasets considered, though**
**these include trivial GEPs such as cell cell cycle and cytoskeletal associated GEPs that**
**should be in all cells which flatters the comparison. GEPs were annotated by manual**
**inspection of the top weighted genes, and poorly supported GEPs removed.**

**TCAT GEPs were benchmarked against the previous tool NMFproj and clustering**
**approaches using an independent CITEseq experiment. TCAT was then applied to the**
**COMBAT covid database and shown to be able to deconvolute subsets of proliferating T**
**cells which had previously been grouped together, and showed that the previous dataset**
**contained many cells of poor quality that had not been filtered out. TCAT also**
**discriminates between CD8+ T cells polarisation states.**

The authors introduce their AIMseq technique, looking at TCR dependent T cell
activation and cGEPs that correlate with antigen reactivity. Using the cGEP enriched in
activated T cells they then generate an ASA score by which the antigen reactivity of a cell
can be assessed, and use this to analyse T cells derived from various cancer samples (in
part by determining a ASA threshold at which to specify a cell as antigen reactive). The
authors show that tumor mutation burden and the proportion of antigen-specific T cells
correlate as would be expected, and then further analyse the cGEP phenotype of
bystander T cells.

Thank you for your thorough and thoughtful summary of our manuscript.

**R4.2**

**B. Originality and significance: if not novel, please include reference**

**Originality: NMF approaches have recently been applied to circulating CD4+ T cells**

**(Single-cell transcriptome landscape of circulating CD4+ T cell populations in**

**autoimmune diseases - ScienceDirect); this new study seeks to define gene expression**

**profiles more broadly and determines a larger number of GEPs in an effort to make a**

**shared reference representation of cell states, in particular T cell states.**

**The significance is hard to determine: the authors are correct that having a well**

**characterised reference panel of GEPs that (in theory) captures all possible T cell GEPs**

**would be of value in mapping studies (especially smaller studies) onto a common**

**representation of cell states is of value to the field of single cell sequencing. The**

**downside is that the cNMF approach is conceptually harder to understand than simpler**

**approaches using reference maps such as Azimuth and ProjecTILs, which may limit**

**adoption. However the authors have made an admirable effort to make their tool available**

**on github, present appropriate reference implementations, and are working on a web**

**based tool that can be deployed on remote compute resources, which should facilitate**

**implementation.**

**As the costs of sequencing continue to fall, and the size and availability of single cell**

**datasets increases, this tool will likely be of increasing value (in particular to meta**

**analyses)**

**While the authors present a huge amount of data, this paper would benefit from less data**

**and a more compelling use case – for example the Sakaguchi group paper puts a clear**

**focus on autoimmunity, I'd encourage the authors to put more emphasis on the last third**

**of their paper looking at cancer to emphasise the novelty or additional value (though I**

**appreciate that *CAT has uses beyond TCAT).**

Thank you for this comment. Based on this, we have benchmarked our method against Azimuth

and ProjecTILs to illustrate the significantly improved performance (**Key revision 2, New**

**Figure 2B, S3B, above**). We have also further developed our website which can be seen at the

following link <https://immunogenomics.io/starcat>. It now hosts additional reference catalogs for

glioma-derived myeloid cells and bone-marrow derived CD34+ hematopoietic stem cells (**Key**

**revision 4, above**).

Most significantly, we have greatly expanded our analysis of cancer to show how TCAT reveals
features of CD4+ tumor-infiltrating lymphocytes that predict response to immune checkpoint
inhibitors (ICIs) in melanomas, colorectal cancer, and non-melanoma skin cancer (**Key revision**
**1 and New Figure 6, above**). Surprisingly, tumors with more proliferating and activated CD4 T-
cells were more likely to progress despite ICI. By contrast, tumors with more cells expressing
the CD4-naive cGEP (and consequently more *TCF7*) were associated with improved ICI
responsiveness. Thank you for raising these points. We believe the updated analysis is more
compelling and greatly increases the quality of the paper.

**R4.3**

**C. Data & methodology: validity of approach, quality of data, quality of presentation**

**As a general point the paper seems undecided as to whether it's a technical paper for a**
**specialist audience with a high background level of understanding, or a newly**
**generalisable method that might be of interest to less specialist readers who have single**
**cell sequencing data.**

We thank the reviewer for this comment. We agree that certain aspects of the paper are quite
technical depending on the background of the reader. Specifically, we envision three target
audiences: (1) T-cell biologists who might wish to use TCAT on their data (2) other biologists
who may use starCAT for other cell types of interest, and (3) single-cell methods developers
who may be interested in the general approach. Currently, the paper is most geared toward (1)
and (3) but may be harder for non-T-cell biologists and non-computational researchers to
understand. Although it is not possible to equally address all audiences, we have made edits to
better sign-post the potential utility of the method to non-T-cell biologists. Specifically, we have
added the following concluding sentence to the abstract:

*Our software package starCAT similarly enables reproducible annotation of other cell*
*types and tissues.*

We also note that the beginning of the discussion summarizes the general method of starCAT
rather than the specific T-cell implementation:

*Here, we introduced starCellAnnoTator (abbreviated starCAT) for annotating scRNA-*
*Seq data with predefined GEPs. starCAT exploits the observation that functionally*
*informative GEPs learned by cNMF are reproducible across datasets and contexts and*
*can thus aid interpretation of new datasets.*

In addition, we now specifically indicate some of the non T-cell references that are hosted on
our website:

*We demonstrated starCAT with T-cells but it is equally applicable to other cell types and*
*tissues. We make the starCAT software publicly available and have created a repository*
*to host cGEP catalogs, including a human glioma myeloid⁹ and human bone marrow*
*hematopoiesis reference¹⁰, enabling easy application to new datasets. Users studying*

other tissues and cell-types can contribute their own catalogs to the repository, akin to
 the molecular signatures database (MSigDB)^{17,18}, but hosting GEPs for annotation of
 scRNA-Seq data rather than gene-sets for enrichment testing. We hope starCAT will aid
 in comprehensive identification of GEPs underlying cell behavior across tissues and
 diseases.

**R4.4**

Given the journal is Nature Methods the figure legends are excessively terse which can
 make the data hard to understand: for example Fig 1A providing the overview of the
 whole approach is described in a single line of text. The figure itself could be expanded
 upon (or at least made larger) to help the reader grasp the core concept of this paper,
 while Fig1B is difficult to read and could be better linked to Fig1A_step2.

Compared to the previous NMF paper this paper has a more detailed methods section
 and a similar or more detailed supplementary data. The main figures are less crowded
 than the supplementals and have been prepared with care commensurate for a top tier
 journal.

Thank you for this comment. We have expanded the size of **Figure 1A** and made the included
 text more legible (**Figure 1A**, reproduced below).

**Revised Figure 1. Overview of star-CellAnnoTator (starCAT).** (A) Schematic of the
 starCAT pipeline. The discovery phase (top) identifies consensus GEPs (cGEPs) by
 running consensus NMF on multiple datasets and combining the results. The annotation
 phase (bottom) uses the cGEPs to annotate new query datasets and to compute
 additional scores and discrete classifiers.

**R4.5**

**D. Appropriate use of statistics and treatment of uncertainties**

I do not have the requisite background in statistics to comment on the mathematical
 suitability of the new normalisation techniques used to combine the 7 datasets. Some

**figures lack statistics, and/or the statistical tools to compare these types of highly**
**complex single cell datasets are not yet well established.**

**The authors have made the code underlying every figure available to allow readers to**
**repeat their analyses which is admirable and underscores their high level of technical**
**ability. If one were to criticise this might benefit from a little more curation - for example**
**Analysis/TCAT/Polarization_MarkerHeatmaps.ipynb contains a number of KeyErrors**
**which suggest that this might not be the final version used.**

Thank you for this comment. We have cleaned up extraneous notebook cells that were part of
our initial exploratory analysis but not part of the final paper. This removed additional KeyErrors
from the notebooks. We have also updated the documentation clarifying in which notebook each
figure was produced.

We have also added additional statistical analysis to figures where they were missing. For
example, **Figure 4F** which shows association of activation-associated cGEPs with lineages has
been re-made using pseudobulked analysis to enable more rigorous statistical hypothesis
testing. See the response to **R4.14** below.

**R4.6**

**E. Conclusions: robustness, validity, reliability**

**The authors demonstrate a number of use cases for *CAT using TCAT to analyse diverse**
**T cell repertoires to show that they can more accurately deconvolute T cell subsets.**

**F. Suggested improvements: experiments, data for possible revision**

**In general review figures and legends for clarity.**

**Fig S5B not mentioned in text**

Thank you for this comment. We have added the text reference for this figure.

**R4.7**

**Section 1**

**I would also suggest tying the results of single cell sequencing/TCAT more closely to**
**what the scientific audience know about T cells – for example in Fig1D how are known**
**markers like ENTPD1 and CXCL13 that happen not to be among the top genes per GEP**
**distributed? Some information here – such as that HLA-DRA and HLA-DRB are top genes**
**in the HLA cGEP – doesn't add much information.**

Thank you for this comment. One of the original goals for this figure was to emphasize the
specificity and ease of interpretability of the discovered cGEPs using some well-known example
programs with clear defining markers. However, we agree that including non T-cell specific
cGEPs such as the HLA adds less information. We have therefore replaced the HLA, ISG, IEG,
and cytoskeleton cGEPs with Exhaustion (which includes *ENTPD1*) and Tfh-1 (which includes
*CXCL13*) (**Revised Figure 1D**, below).

**Revised Figure 1D.** Marker genes for selected example cGEPs in cNMF gene score
units.

**R4.8**

**I was surprised to see CXCL13 as a top weighted gene in T follicular helped cells, as this**
**has been reported by many groups (Platten/Green, Wu, Rosenberg) as being a key**
**marker for tumor reactive T cells**

Thank you for this comment. We agree that *CXCL13* has been identified as an important marker
of tumor-reactivity. It has also been shown to be a key marker of T follicular helper cells^{19–21} and
T peripheral helper cells^{22,23}, which is consistent with the role of these cells in providing B-cell
help. We suspect that many chronically antigen stimulated cells within tumors and rheumatoid
arthritis synovia become Tph and Tfh. Thus this is not contradictory with *CXCL13* being
upregulated in tumor reactive T-cells that have become Tph and Tfh cells. We also have
identified a *CXCL13*-expressing activation cGEP, labeled OX40/EBI3, which we believe may be
associated with tumor-reactivity (**See R4.16**). We added the following text to the manuscript
noting the literature relating *CXCL13* to tumor-reactivity:

*The OX40/EBI3 cGEP was identified specifically in tissues and was most common in*
*tumor-infiltrating T-cells (Figure 1B, Extended Data Figure 7D). It includes CXCL13, a*
*marker of tumor-reactivity²⁴, as well as several activation-induced markers used to*
*define AIM-positivity, including TNFRSF4, encoding OX40, and IL2RA, encoding CD25.*

R4.9

Section 3

Fig 2B – Accuracy does not reflect false positive assessments, a metric like G-Mean might better show the improved performance of the TCAT multilabel approach vs the single cGEP especially for CD8 EM. Is this what the authors mean in Fig2C with “balanced accuracy”? Possibly this is stated in the methods, but would be easier to state this in the legend/results.

We thank the reviewer for their comment and we agree about the importance of choosing a metric that balances both sensitivity and specificity. Throughout this portion of the text, we used a balanced accuracy metric which works well when the data is imbalanced, which is the case for rare cell populations. In the binary case (as in the previous **Figure 2B**, now **Extended Data Figure 5A**), balanced accuracy is calculated as the arithmetic mean of the sensitivity and specificity for a prediction. It thus incorporates both false positive and false negative assessments. Balanced accuracy also generalizes to the multiple class prediction problem, in which case it is defined as the mean sensitivity across classes (used in new **Figure 2B** for simplicity, displaying a single accuracy value per method rather than one per lineage). This approach in effect reflects false positive rates, as a high false positive rate of one class will result in a low true positive rate (sensitivity) of others. While geometric mean would likely also work well, we chose arithmetic mean as we thought it may be more familiar for some readers. To help clarify why we used balanced accuracy, we have added the following text to the manuscript, which is in reference to the multi-class case:

To measure performance of a prediction, we quantified its balanced accuracy, calculated as the mean sensitivity across classes, an approach which gives equal weighting to each class and is appropriate for cases of class imbalance, such as in cases of rare cell populations.

We also added the following text to the legend of **Extended Data Figure 5A** (shown below) to clarify the calculation of balanced accuracy in the binary case:

R4.10

Section 4

New Extended Data 5A. Balanced accuracy comparisons of TCAT's prediction of manually gated subsets, as compared to Azimuth, Symphony, ProjectTILs, and clustering with multiple Leiden resolution parameters. Balanced accuracy is calculated as the mean sensitivity and specificity of a prediction, thus reflecting both true positive and negative rates and appropriate in cases of class imbalance.

While TCAT does deconvolute the proliferating T cell cluster well, many studies perform regression of cell cycle genes so this is more a limitation of the COMBAT Covid-19 dataset than a feature of TCAT (though it nicely shows how previous data benefits from mapping onto a common reference).

We thank the reviewer for highlighting this point. We agree that some researchers do address cell cycle as a confounder through regressing of cell cycle genes. However, cell cycle is just one example of a confounding non-lineage signal. We show similar effects of other signals such as interferon, cell quality, and cytotoxicity (**Figures S4B-C** reproduced below for convenience). In theory, these could all be regressed out, but we find our approach to be significantly more streamlined, and regressing out other signals is neither widespread nor validated. We like to keep the cell cycle example as a common, important, and easy to understand example of how lineage and non-lineage signals can get confounded. We also find it a biologically important example because scientists may frequently wish to know which lineages and conditions have the most proliferation. We agree that this could be characterized using alternative approaches (regress cell cycle to first cluster the cells, then use cell cycle gene set scoring to infer proliferating cells). However, our approach is significantly more streamlined and thus likely easier for many users. We have added the text below to clarify these points:

Next, we illustrate how TCAT reveals cellular heterogeneity not visible with clustering. As a first example, we consider cell cycle, a common signal that obscures other aspects of proliferating cells²⁵ in the COMBAT Covid-19 dataset. While it is possible to regress out cell cycle during preprocessing²⁶, this does not always work well, can remove correlated biological signals, and does not address analogous signals like cell quality, ISG, and cytotoxicity that can also obscure cell type. TCAT addresses these limitations by simultaneously modeling the GEPs underlying cell cycle, cell lineage, and more.

**Figure S4B-C. (B)** Usage of selected cGEPs (columns) in cells (rows) grouped by maximum
subset cGEP. Cells are drawn from subclusters with high usage of the ISG cGEP, indicated in
the colorbar. **(C)** Same as **(B)** but only showing cells from subclusters with high cytotoxicity
cGEP usage.

**R4.11**

**Looking at Fig S4 comparing the two covid databases COMBAT with UK-Covid, while**
**TCAT can deconvolute proliferating T cells into constituent subsets to show the**
**important role of the T peripheral helper cGEP in covid patients vs healthy, there is little**
**explanation of the large differences between these two covid datasets. Could the authors**
**comment on this – I believe this is important given that a core argument is that datasets**
**can (should?) be mapped to a common reference panel of GEPs. Dataset quality seems**
**to be part of the answer given that the authors later comment on the high incidence of a**
**cGEP indicative of poor cell quality in the COMBAT study. Why is the proportion of**
**CD8trm cGEP cells in the proliferative state so different between studies? I would**
**speculate that some cGEP are dependent on genes with low expression, such that poor**
**sample handling (e.g. long delay between sample acquisition and single cell capture)**
**depletes some cGEP more than others? The authors used an overdispersion metric to**
**identify key genes for the cNMF analysis, but does this come at a price of robustness? I**
**would like to see some discussion of these differences**

Thank you for this comment. While we are not certain about the precise reason for the
differences in rates of proliferation (and activation, as shown in **Figure 5G** and mentioned in
**R4.15**) between the two Covid-19 datasets, it is an important topic which we now specifically
address in the text. Following up on the reviewer’s suggestion, we observed a positive
correlation between both proliferation and antigen-specific activation (ASA) rates with several
signals that are proxies for sample quality, including usage of the Poor-Quality and
mitochondrial cGEPs, and the detected number of UMIs per cell. ASA and proliferation cGEPs
were positively correlated with PoorQuality and mitochondrial cGEP usage and negatively
correlated with average library size (**New Extended Data Figure 9A**). This is consistent with the
hypothesis that low quality samples have more activation. We hypothesize that differences in
cell processing pipelines may result in lower quality samples with higher rates of cell death and
non-specific activation and corresponding cell proliferation.

We have added a new paragraph and supplementary figure highlighting the differences in
proliferation and activation between the Covid-19 datasets and their potential explanations. As
all reference mapping algorithms can be influenced by batch artifacts and downstream effects of
processing methods, we also added text indicating the importance of careful interpretation of
cross-dataset comparisons, particularly regarding cGEPs which may be affected by processing
pipelines, such as artifact and activation-associated GEPs:

*We quantified higher rates of activation in both healthy and Covid-19 samples in UK-*

*Covid than in the COMBAT dataset. We hypothesized that this could reflect non-specific*

activation related to sample processing. Consistent with this, the two Covid-19 datasets greatly differed in sample quality, as marked by higher usage of the Poor-Quality and Mito cGEPs in the UK-Covid dataset. ASA and proliferation were both correlated with sample quality, as reflected by correlation with the Poor-Quality cGEP and library size (**Extended Data Figure 9**). Thus, it is important to carefully interpret cross-dataset differences in activation as these could reflect differences in sample quality.

New Extended Data Figure 9A. Per-sample scatterplots showing concordance between mean antigen-specific activation (ASA) score (top) or proliferation (bottom) and various quality metrics, across datasets and sites. Spearman correlation and P-values are shown.

The reviewer also highlighted a difference in proportions of proliferating CD8 TRMs between the two Covid-19 datasets. This is again an important point where we believe there may be a handful of factors at play. As both Covid-19 datasets are derived from blood, the number of CD8 TRMs, or cells with high usage of the CD8 TRM-Like cGEP, are relatively low. However, the presence of circulating TRM cells is somewhat surprising to see and may reflect memory cells migrating to the periphery. Different abundances of proliferating TRMs may reflect differences in clinical properties of the samples, treatments, or other dataset-specific features. Although the rates of proliferation of CD8 TRMs in Covid-19 samples is discordant between these two datasets, we did note a high degree of correlation in rates of proliferation across lineages

between the datasets ($R=0.80$ in Covid-19 samples, $R=0.56$ in healthy). We have added the
following text indicating the concordance of proliferation rates between the two datasets:

*Proliferation rates across lineages were largely concordant between both Covid-19*
*datasets ($R = 0.80$ and $P= 0.00021$ in Covid-19 samples; $R = 0.56$ and $P = 0.025$ in*
*healthy samples).*

**R4.12**

**When the authors state “Cells with high usage of the CD4-Naive cGEP expressed CD4**
**naive markers including CD45RA protein and SELL RNA, confirming that clustering**
**misclassified them as memory T-cells (Extended Data 4)” some memory T cells also**
**express CD45RA and SELL (e.g. TEMRA or Tcm)**

We thank the reviewer for this point. Our understanding is that TEMRAs are definitionally
CD45RA positive and *SELL* (L-selectin) negative, whereas Tcm are definitionally *SELL* positive
and CD45RA negative. The reviewer is correct that by looking at one marker at a time, rather
than both markers together, we may be incorrectly annotating some TEMRAs or Tcms as CD4-
Naive. To address this point, we have added a biaxial plot of AB_CD45RA vs. AB_CD62L for
the selected population, colored by the usage of CD4-Naive cGEP, which shows that a handful
of the cells in the memory cluster that use the CD4-Naive GEP in fact do express the
combination of markers that label them as naive.

***New Extended Data Figure 6E.*** Expression of CD45RA and CD62L colored by usage
*of the CD4-Naive cGEP. Protein expression is smoothed using MAGIC²⁷ and color*
*intensities are averaged over 20 nearest neighbors to reduce overplotting.*

**R4.13**

**Section 5 & 6**

**By definition naïve cells should not have encountered their cognate antigen. Clearly the**
**immune system is constantly generating new TCR clonotypes, so indeed some CEFX**
**reactive T cells could be naïve, however can the authors rule out a false positive result?**
**By sequencing TCR clonotypes (as in section 6) one would expect that clonotypes in the**
**naïve population should be less clonal/more diverse, and that T cells sharing the same**
**clonotypes should not be found in both groups. Some analysis of TCR clonality would**
**support this analysis and give additional confidence in the scoring (some of which is**
**described in Fig S6).**

We thank the reviewer for this point and have carried out the analyses they suggested resulting
in the new manuscript text and **New Extended Data Figure 7B**, reproduced below:

*Naive T-cell clones were significantly less likely to be clonally expanded (i.e. have 2+*
*cells with the same beta CDR3 sequence) than memory clones ($P=2.4 \times 10^{-40}$ and*
*$P=9.5 \times 10^{-257}$ for CD4 and CD8 T-cells, respectively). Furthermore, there were*
*significantly more expanded clones in AIM-positive than AIM-negative CD4 memory T-*
*cells and a trend in this direction for CD8 memory T-cells ($P=2.1 \times 10^{-7}$ and 0.14 for CD8*
*and CD8 memory T-cells, respectively, **Extended Data Figure 7B**). Overall, there was*
*significant concordance of AIM responsiveness within a clone: two cells with the same*
*beta CDR 3 chain had a 77% chance of being concordantly AIM positive or AIM*
*negative, compared to only a 51% probability based on chance alone (binomial test*
*$P < 1 \times 10^{-200}$).*

**New Extended Data Figure 7B.** *Fraction of clonally expanded T-cells stratified by*
*memory vs. naive and by treatment condition. Clonal expansion is defined as having 2+*
*T-cells in the same donor with the same CDR3 beta chain. Error bars reflect 95 percent*
*bootstrap confidence intervals.*

R4.14

While I would agree that proliferation is a core response of TCR activation, using it to validate cGEP choice seems less useful – I believe Fig 4F is one of the least convincing in the paper, with very large error bars and no stats. This should be better explained.

We thank the reviewer for this comment. Figure 4F attempts to demonstrate the lineage-specificity of some of the activation-associated cGEPs. We agree that it would be a significantly stronger figure with clear statistical hypothesis testing. To address this, instead of plotting GEP usage across cells from all samples combined, we now plot the average cGEP usage per pseudobulk sample for each gated lineage (**Revised Figure 4F**). This allows us to do more rigorous statistical hypothesis testing. This shows that while NME1/FABP and OX40/EBI3 are upregulated with antigen recognition in all lineages, TIMD4/TIM3 was specific to CD8_CM and MAIT and multi-cytokine is specific to CD8 memory, MAIT, and gDT.

We also agree with the reviewer that proliferation is just one response to TCR activation and thus is not sufficient validation. We hope the AIM-Seq assay, the correspondence of the cGEPs with activation marker proteins, the clonal abundance analysis, and the association between activation rates and disease states (e.g. activation being higher in Covid-19 than in healthy donors), provide additional validation for the activation-associated cGEPs.

**Revised Figure 4F. Identifying cGEPs associated with TCR-dependent activation.**
*Average usage of selected Aim-associated cGEPs in +, -, and U cells from different*
*gated subsets, per sample. Lines show the median. * indicates $P < .05$ and average*
*usage in the + cells of greater than 0.01.*

**R4.15**

**The ASA data is very interesting, but again large differences between COMBAT and UK-**
**covid data in terms of absolute percentage are not explained. I would like to see more on**
**this part of the paper – for example why are some cGEPs specific to particular cancers,**
**presumably due to cancer microenvironment factors or similar? Fig 5J starts to get at**
**this with a correlation between tumor mutational load and exhaustion, though seeing**
**how how ASA correlates with survival of individual patients would be more convincing**
**(though such datasets are only just becoming available).**

Thank you for this comment. We have discussed the difference between the COMBAT and UK-
Covid dataset per the comment above (**See R4.11**). We agree that further discussion of the link
between ASA and the tumor microenvironment would be impactful. To this end, we have added
a new analysis characterizing the link between ASA and individual cGEPs and response to
immune-checkpoint inhibitor (ICI) therapy (**Key revision 1 and New figure 6, above**). We find
this to be a more impressive analysis than **Figure S7B** (originally **Figure 6B**) which looks at
cGEP association with particular cancers and thus have substituted it in the revised manuscript.

**R4.16**

**A large number of recent papers have shown CXCL13 to be key marker for tumor-**
**reactive T cells (i.e. not just T peripheral helper cells), while virus reactive T cells seem to**
**show a slightly difference activation pattern. See previous points on CXCL13, I'd like to**
**see this explored more: could it be that the use of a viral peptide pool in AIMseq gives**
**slightly different cGEP enrichment than true tumor reactivity?**

We thank the reviewer for this comment. We agree that there are likely different patterns of
activation in different disease contexts and thus the CEFX stimulation may be biased in the
responses it elicits. To address this, we have added text describing this limitation to the
discussion section of the manuscript:

*A limitation of the AIM-Seq experiment is that it only uses a single costimulation signal,*
*CD28 and CD49d costimulation, and not the full diversity of what is possible in vivo.*
*Furthermore, it only uses a single set of peptides derived from common pathogens. It*
*therefore could bias the patterns of cGEP enrichments that are discovered. For example,*
*a tumor microenvironment may induce exhaustion more than an acute viral infection*
*microenvironment.*

However, we note that the cGEPs we found to be activation-associated and used to define the
ASA score were learned from both tissue and blood datasets from multiple disease contexts,
including cancer. To the reviewer's point, CXCL13 is included among the top associated genes
for the activation-associated OX40/EBI3 cGEP (**New Extended Data Figure 7C**). It is also high

in the Tph, Tfh-1, and Tfh-2 cGEPs. The reviewer raises an interesting point that we may be
 able to characterize different patterns of activation cGEP usages across different disease
 contexts. We note that the OX40/EBI3 cGEP was learned from tumor and normal tissue.
 Beyond this, we conducted an additional analysis showing that OX40/EBI3 usage is significantly
 enriched in tumor tissue with respect to healthy tissue or blood, or other disease contexts (**New**
 **Extended Data Figure 7D**). We describe this finding in the the new text reproduced below:

The OX40/EBI3 cGEP was identified specifically in tissues and was most common in tumor-infiltrating T-cells (Figure 1B, Extended Data Figure 7D). It includes CXCL13, a marker of tumor-reactivity⁴², as well as several activation-induced markers used to define AIM-positivity, including TNFRSF4, encoding OX40, and IL2RA, encoding CD25.

**New Extended Data Figure 7C-D. (C)** Gene weights for CXCL13 in various cGEPs,
 including Tfh-1, Tfh-2, Tph (positive controls) and all AIM-associated cGEPs. **(D)**
 Boxplots of mean per-sample usage of the AIM-associated cGEPs used to define ASA,
 by dataset and by disease. T-tests were performed for each cGEP to compare
 differences between tumor samples and other disease types (***) indicates $P < 0.001$, **
 indicates $P < 0.01$, * indicates $P < 0.05$). Boxes represent the interquartile range and
 whiskers represent 95% quantile range.

**R4.17**

**G. References: appropriate credit to previous work?**

**Yes**

**H. Clarity and context: lucidity of abstract/summary, appropriateness of abstract,**
**introduction and conclusions**

**The abstract, introduction and discussion sections are very well written.**

We thank the reviewer for these comments.

Bibliography

1. Parikh, R. B. *et al.* Uptake and Survival Outcomes Following Immune Checkpoint Inhibitor
Therapy Among Trial-Ineligible Patients With Advanced Solid Cancers. *JAMA Oncol* **7**,
1843–1850 (2021).

2. Sade-Feldman, M. *et al.* Defining T Cell States Associated with Response to Checkpoint
Immunotherapy in Melanoma. *Cell* **176**, 404 (2019).

3. Yost, K. E. *et al.* Clonal replacement of tumor-specific T cells following PD-1 blockade. *Nat*
*Med* **25**, 1251–1259 (2019).

4. Li, J. *et al.* Remodeling of the immune and stromal cell compartment by PD-1 blockade in
mismatch repair-deficient colorectal cancer. *Cancer Cell* **41**, 1152–1169.e7 (2023).

5. Hao, Y. *et al.* Integrated analysis of multimodal single-cell data. *Cell* **184**, 3573–3587.e29
(2021).

6. Andreatta, M. *et al.* Interpretation of T cell states from single-cell transcriptomics data using
reference atlases. *Nature Communications* **12**, 1–19 (2021).

7. Kang, J. B. *et al.* Efficient and precise single-cell reference atlas mapping with Symphony.
*Nat Commun* **12**, 5890 (2021).

8. Perkel, J. M. No installation required: how WebAssembly is changing scientific computing.
*Nature* **627**, 455–456 (2024).

9. Miller, T. E. *et al.* Programs, Origins, and Niches of Immunomodulatory Myeloid Cells in
Gliomas. *bioRxiv* (2023) doi:10.1101/2023.10.24.563466.

- 10. Li, H. *et al.* The dynamics of hematopoiesis over the human lifespan. *Nature Methods* 1–13
(2024).
- 11. Wagner, A., Regev, A. & Yosef, N. Revealing the vectors of cellular identity with single-cell
genomics. *Nat. Biotechnol.* **34**, 1145–1160 (2016).
- 12. Korsunsky, I. *et al.* Fast, sensitive and accurate integration of single-cell data with
Harmony. *Nat. Methods* **16**, 1289–1296 (2019).
- 13. Avila Cobos, F., Alquicira-Hernandez, J., Powell, J. E., Mestdagh, P. & De Preter, K.
Benchmarking of cell type deconvolution pipelines for transcriptomics data. *Nat Commun*
**11**, 5650 (2020).
- 14. Colino-Sanguino, Y. *et al.* Performance comparison of high throughput single-cell RNA-Seq
platforms in complex tissues. *Heliyon* **10**, e37185 (2024).
- 15. Cheng, C., Chen, W., Jin, H. & Chen, X. A Review of Single-Cell RNA-Seq Annotation,
Integration, and Cell-Cell Communication. *Cells* **12**, (2023).
- 16. Gavish, A. *et al.* Hallmarks of transcriptional intratumour heterogeneity across a thousand
tumours. *Nature* **618**, 598–606 (2023).
- 17. Subramanian, A. *et al.* Gene set enrichment analysis: a knowledge-based approach for
interpreting genome-wide expression profiles. *Proc. Natl. Acad. Sci. U. S. A.* **102**, 15545–
15550 (2005).
- 18. Liberzon, A. *et al.* Molecular signatures database (MSigDB) 3.0. *Bioinformatics* **27**, 1739–
1740 (2011).
- 19. Gu-Trantien, C. *et al.* CXCL13-producing TFH cells link immune suppression and adaptive
memory in human breast cancer. *JCI Insight* **2**, (2017).
- 20. Lin, X. *et al.* Follicular Helper T Cells Remodel the Immune Microenvironment of Pancreatic
Cancer via Secreting CXCL13 and IL-21. *Cancers* **13**, 3678 (2021).
- 21. Havenar-Daughton, C. *et al.* CXCL13 is a plasma biomarker of germinal center activity.
*Proc Natl Acad Sci U S A* **113**, 2702–2707 (2016).

- 22. Rao, D. A. *et al.* Pathologically expanded peripheral T helper cell subset drives B cells in
rheumatoid arthritis. *Nature* **542**, 110–114 (2017).
- 23. Kobayashi, S. *et al.* A distinct human CD4+ T cell subset that secretes CXCL13 in
rheumatoid synovium. *Arthritis Rheum* **65**, 3063–3072 (2013).
- 24. Tan, C. L. *et al.* Prediction of tumor-reactive T cell receptors from scRNA-seq data for
personalized T cell therapy. *Nature Biotechnology* 1–9 (2024).
- 25. Buettner, F. *et al.* Computational analysis of cell-to-cell heterogeneity in single-cell RNA-
sequencing data reveals hidden subpopulations of cells. *Nat. Biotechnol.* **33**, 155–160
(2015).
- 26. Butler, A., Hoffman, P., Smibert, P., Papalexi, E. & Satija, R. Integrating single-cell
transcriptomic data across different conditions, technologies, and species. *Nat Biotechnol*
**36**, 411–420 (2018).
- 27. van Dijk, D. *et al.* Recovering gene interactions from single-cell data using data diffusion.
*Cell* **174**, 716–729.e27 (2018).
